# Optimistic Rates for Multi-Task Representation Learning

**Austin Watkins**
Johns Hopkins University
Baltimore, MD 21218
awatki29@jhu.edu

**Enayat Ullah**
Johns Hopkins University
Baltimore, MD 21218
enayat@jhu.edu

**Thanh Nguyen-Tang**
Johns Hopkins University
Baltimore, MD 21218
nguyent@cs.jhu.edu

**Raman Arora**
Johns Hopkins University
Baltimore, MD 21218
arora@cs.jhu.edu

## Abstract

We study the problem of transfer learning via Multi-Task Representation Learning (MTRL), wherein multiple source tasks are used to learn a good common representation, and a predictor is trained on top of it for the target task. Under standard regularity assumptions on the loss function and task diversity, we provide new statistical rates on the excess risk of the target task, which demonstrate the benefit of representation learning. Importantly, our rates are optimistic, i.e., they interpolate between the standard $\mathcal{O}(m^{-1/2})$ rate and the fast $\mathcal{O}(m^{-1})$ rate, depending on the difficulty of the learning task, where $m$ is the number of samples for the target task. Besides the main result, we make several new contributions, including giving optimistic rates for excess risk of source tasks (Multi-Task Learning (MTL)), a local Rademacher complexity theorem for MTRL and MTL, as well as a chain rule for local Rademacher complexity for composite predictor classes.

## 1 Introduction

Transfer learning has emerged as a powerful tool in modern machine learning. The goal is to find a good predictor on a given *target* task with little data, by extracting knowledge from *source* task(s). The source tasks can either be explicitly given (supervised learning), or constructed from domain knowledge (unsupervised/self-supervised learning). An example of the supervised learning approach is a widely successful practice in deep learning, wherein a complex model is first trained on a source task with plentiful data, then applied to a target task by extracting *representations* of data by *freezing* the last layer of the complex model, and training a simple (linear) classifier on top of it [Bengio et al., 2013, Donahue et al., 2014]. Similarly, the unsupervised approaches of training a (large) neural network on a *general objective* has seen tremendous success, especially in natural language processing [Brown et al., 2020, Devlin et al., 2019] and computer vision. As before, a simple linear classifier trained on the output of the penultimate layer works well on a target task.

*Representation learning*, where the goal is to learn representations of data that are useful across multiple domains and tasks, is a common theme in the examples above. We study a popular formalization of this type of machine learning called *Multi-Task Representation Learning (MTRL)*. As the name suggests, we are given $t$ source tasks each with a small number of data points, say $n$. We seek to effectively pool the $nt$ data points to extract a common good representation useful for the target task. Therefore, if our tasks are sufficiently "diverse", we hope to effectively circumvent our data scarcity by sharing knowledge between tasks. As a by-product, since the shared representation

37th Conference on Neural Information Processing Systems (NeurIPS 2023).

can also be used for the source tasks themselves, this procedure is also effective for the related problem of *multi-task learning (MTL)*.

To study the transfer of knowledge via representations within the MTRL framework, we consider a **composite-learning model**. This model consists of $\mathcal{F}$ and $\mathcal{H}$, a class of hypotheses and representations respectively, where the final end-to-end predictors are given via the composition $f \circ h$, for all $f \in \mathcal{F}, h \in \mathcal{H}$. Since we must decide between compositions, we use a two-stage empirical risk minimization procedure. First, **the multi-task representation learning stage**, the procedure selects one common representation and $t$ predictors, one for each task, by minimizing the average empirical loss on the source tasks. Second, **the transfer learning stage**, the procedure selects a predictor for the target task that minimizes the empirical loss, w.r.t. $m$ samples, on top of the fixed representation from stage one.

We desire to learn a good predictor – i.e., one that has small excess risk – from $\mathcal{F} \circ \mathcal{H}$ for the target task. In this work, we consider the class $\mathcal{H}$ to be complex (e.g., deep neural networks), whereas the class $\mathcal{F}$ consists of simple functions (e.g., linear predictors). Therefore, the class $\mathcal{F} \circ \mathcal{H}$ is complex, so directly learning it would require a large number of samples. Yet, if we can effectively pool all our data from each task, the total number of examples could be sufficiently plentiful to successfully learn a complex representation. Further, a good representation can significantly reduce the statistical burden of learning a good predictor for the target task. This phenomenon is formalized by the prior work of Tripuraneni et al. [2020], which is the work most related to ours. Specifically, they show that if the loss function is Lipschitz and the tasks are sufficiently diverse (see Definition 2), the excess (transfer) risk for the target task is bounded as,

$$\text{Excess transfer risk } = \tilde{O}\left(\sqrt{\frac{C(\mathcal{H}) + tC(\mathcal{F})}{nt}} + \sqrt{\frac{C(\mathcal{F})}{m}}\right),$$

where $C(\mathcal{H})$ and $C(\mathcal{F})$ denote the complexity measures associated with learning $\mathcal{H}$ and $\mathcal{F}$ respectively (precise definition to follow in a later section).

While the above rate has the desirable properties described above, it is insufficient on many fronts, which our work seeks to address. First, it is akin to the *standard* rate of $n^{-1/2}$, albeit optimal in agnostic settings, for single-task PAC learning. However, it is well-known that a *fast* rate is achievable under realizability in the standard supervised learning. But, the above guarantee does not capture this possibility. Importantly, the work of Tripuraneni et al. [2020] operates under the assumption that the loss function is Lipschitz – in such settings, it is generally not possible to get fast rates, despite realizability, even for the special case of learning half-spaces [Srebro et al., 2010]. In the single task setting, the literature contains a rich theory of guarantees that interpolate between the slow and fast rate depending on the *level* of realizability, such guarantees are called "optimistic rates". We derive such optimistic rates for MTRL. Consequently, as an important application of our general theory, we show that the (transfer) risk of the MTRL approach on the target task is bounded as,

$$\tilde{O}\left(\boldsymbol{L}^*_{\text{target}} + \sqrt{\frac{\boldsymbol{L}^*_{\text{source}}\left(C(\mathcal{H}) + tC(\mathcal{F})\right)}{nt}} + \sqrt{\frac{\boldsymbol{L}^*_{\text{target}}C(\mathcal{F})}{m}} + \frac{C(\mathcal{H}) + tC(\mathcal{F})}{nt} + \frac{C(\mathcal{F})}{m}\right) \quad (1)$$

when using non-negative smooth losses (such as squared loss in linear regression or smoothed ramp loss for classification [Cortes et al., 2021, Srebro et al., 2010]). The values $\boldsymbol{L}^*_{\text{source}}$ and $\boldsymbol{L}^*_{\text{target}}$ represent the minimum average risk over the source tasks and target task respectively, hence (1) yields a fast rate under *mild* realizability, i.e. when they are small.

In the course of proving the above, we extend a number of foundational results in the single-task setting to the multi-task setting. We list them in our contributions below.

1. We give a general local Rademacher complexity result for transfer learning via multitask representation learning – see Theorem 1.
2. We also provide a local Rademacher complexity result for multitask learning – see Theorem 3. Our result has two benefits compared to prior work [Yousefi et al., 2018]. First, we perform MTL via MTRL that leads to improved bounds which show the benefit of pooling data as opposed to learning each task separately. Second, we operate under the special, yet fairly natural setting, of hypothesis classes with product-space structure – this simplifies the proof significantly as well as recovers the result for single-task setting with the exact same constants (unlike rates in Yousefi et al. [2018]).

3. For non-negative smooth losses, we derive optimistic rates (such as Eqn. (1)) for transfer learning and MTL via MTRL. Notably, when we restrict our rates to the single task setting, our proof is significantly simpler and our obtained bound on local Rademacher complexity is smaller compared to the prior work of Srebro et al. [2010].

4. Finally, we present a chain rule for local Rademacher complexity for composite settings, which allows us to decouple the (local) complexities of representation and predictor classes.

## 1.1 Our techniques

In this section, we expand on our contributions and provide an overview of techniques and challenges overcome in the context of prior art.

**Local Rademacher complexity for transfer learning via MTRL.** We build on the work of Tripuraneni et al. [2020], which introduced the notion of task diversity. They showed that task diversity leads to improved rates on excess transfer risk via MTRL, compared to training only on the target task. Their work however yields the standard $O(m^{-1/2})$ rate on excess risk, akin to what is obtained in agnostic settings. We provide a local Rademacher complexity result for transfer learning via MTRL, yielding optimistic rates depending on distributional properties. Our methods are based on tools and techniques from the seminal local Rademacher complexity paper of Bartlett et al. [2005]. A crucial component of our results is relative deviation Bennett concentration inequalities established via log-Sobolev inequalities together with the entropy method [Bousquet, 2002]. We make the crucial observation that the above concentration inequalities, though established for the single-task setting, are general enough to be used to extend the local Rademacher complexity framework to the multi-task and transfer learning setting.

**Local Rademacher complexity for multi-task learning via MTRL.** Intermediate to our above contribution, we provide a local Rademacher complexity result for MTL via MTRL, yielding optimistic rates. We note that the prior work of Yousefi et al. [2018] already provided a local Rademacher complexity result for MTL, though not via MTRL; importantly, our work establishes the benefits of pooling data compared to learning each task separately. However, in contrast to Yousefi et al. [2018][1], our setting is limited to hypothesis classes with a product space structure rather than a general vector-valued hypothesis. This simply means that the tasks are permutation-invariant in our setting, which we argue is a fairly reasonable assumption in the multi-task context. This assumption makes the analysis much simpler as we can reuse concentration inequalities for single task setting whereas Yousefi et al. [2018] had to extend log-Sobolev concentration inequalities to the multi-task setting. Additionally, this yields bounds with better constants than Yousefi et al. [2018] – in the special case of a single task, this recovers the known result with the exact same constants.

**Optimistic rates for non-negative smooth losses.** An important application of our theorems is providing optimistic rates when using non-negative smooth losses, which is known to enable distribution-free optimistic rates in a single task setting [Cortes et al., 2021, Srebro et al., 2010]. This result shows the provable benefits of representation learning. Further, it can be combined with the chain rule for Gaussian complexity of Tripuraneni et al. [2020]. This allows us to separate the composite class $\mathcal{F} \circ \mathcal{H}$ into the complexities of representation class $\mathcal{H}$ and predictor class $\mathcal{F}$ and also reuse existing complexity bounds in the literature. While we borrow some of the tools for optimistic rates theory in the single task setting from Srebro et al. [2010], the prior proof technique does not directly extend to the multi-task setting. Consequently, we modify the proof, simplifying it significantly as well as getting an improved bound, even in the single task setting. In particular, Srebro et al. [2010] bounds the local Rademacher complexity of the loss class by an $\ell_2$-covering number via Dudley's entropy integral formula, which is then bound by $\ell_\infty$-covering number of hypothesis class using smoothness, which in turn is bound by the fat-shattering dimension, which eventually is bound by the (global) Rademacher complexity of the hypothesis class. Some of the steps seemingly do not generalize in the multi-task setting because there are no (well-studied) multi-task analogues, e.g. the fat-shattering dimension. Our proof starts with the above bound on the local Rademacher complexity of the loss class by an $\ell_2$-covering number via Dudley's entropy integral formula but then applies the well-known Sudakov minoration inequality that yields a bound in terms of Gaussian width; this is in turn bound by Rademacher complexity by standard known relations between the two quantities.

**Chain rule for local Rademacher complexities.** In the application to non-negative smooth losses, the local Rademacher complexity of the loss class is bounded by a non-trivial function of the

---

[1]Yousefi et al. [2018] additionally considers Bernstein classes, which we do not. However, we think that our results can be extended to that setting.

(global) Gaussian complexity of the hypothesis class which is further bounded using the chain rule of Tripuraneni et al. [2020]. For general loss functions, there may not be any such non-trivial function. To this end, for Lipschitz losses, we develop a chain rule of local Rademacher complexity, Theorem 5, that similarly allows separating the *local complexities* of the representation and predictor class.

## 1.2 Related work

As mentioned before, our work is most related to, and builds on, Tripuraneni et al. [2020]. Other related work includes an early work of Baxter [2000] that gives guarantees on excess risk of transfer learning and multi-task learning via MTRL under a certain generative model for tasks. This was subsequently improved by Maurer et al. [2016], Pontil and Maurer [2013]. These early works give rates of the form $\mathcal{O}\left(t^{-1/2} + m^{-1/2}\right)$ and do not capture the advantage of having a large number of source samples per task. Prior work has extensively studied the special case of linear representations and/or linear predictors, partly in the context of meta learning, for instance see Du et al. [2021], Tripuraneni et al. [2021], Xu and Tewari [2021]. Further, multitask learning has been explored under various notions of task relatedness [Ben-David and Borbely, 2008, Cavallanti et al., 2010].

Regarding optimistic rates, the seminal work of Vapnik and Cervonenkis [1971] provided the first optimistic rates, via a normalized uniform convergence analysis, but was limited to the PAC learning setting. Much later, a line of work of Bousquet et al. [2002], Koltchinskii and Panchenko [2000] derived optimistic rates in more general settings using the technique of *localization*. In this area, an important result is the local Rademacher complexity theorem of Bartlett et al. [2005], which many works have leveraged to yield improved rates for various problems [Blanchard et al., 2007, Cortes et al., 2013, Ullah et al., 2018]. Importantly, this theorem applies to learning with non-negative smooth losses [Srebro et al., 2010] where the local Rademacher complexity-based technique can be used to give distribution-free optimistic rates. Further, optimistic rates have seen renewed interest owing to the interpolation/benign overfitting phenomenon in deep learning. In certain problems, predictors may interpolate the data (zero training error), yet have small risk. A line of work [Zhou et al., 2020, 2021] shows that in certain linear regression problems, optimistic rates can be used to derive risk bounds that explain the phenomenon, where the usual tool of uniform convergence provably fails.

There has been some work on optimistic rates in the multi-task setting. The work of Yousefi et al. [2018] established a local Rademacher complexity theorem for multi-task learning (see Section "Our techniques" for detailed comparison). Besides that, the work of Reeve and Kaban [2020] proved optimistic rates for vector-valued hypothesis classes for self-bounding loss functions (which includes non-negative smooth losses).

**Notation.** Denote $\|\cdot\|_2$ be the Euclidean norm and $\|\cdot\|_\infty$ be the infinity norm. We use the convention that $[m] = [1, \ldots, m]$ is the list of contiguous integers starting at 1 and ending at $m$. For $\boldsymbol{f} = (f_1, \ldots, f_t)$ let $\boldsymbol{f}^2 = \left(f_1^2, \ldots, f_t^2\right)$ and denote

$$P\boldsymbol{f} := \frac{1}{t}\sum_{j=1}^t P_j f_j = \frac{1}{t}\sum_{j=1}^t \mathbb{E}\left(f_j(X_j)\right), \quad \hat{P}^n \boldsymbol{f} := \frac{1}{t}\sum_{j=1}^t \hat{P}_j^n f_j = \frac{1}{nt}\sum_{j=1}^t \sum_{i=1}^n f\left(X_j^i\right).$$

## 2 Problem setup and preliminaries

Let $\mathcal{X} \subseteq \mathbb{R}^d$ and $\mathcal{Y} \subseteq \mathbb{R}$ denote the input feature space and the output label space respectively. The source tasks and the target task are represented using probability distributions $\{P_j\}_{j=1}^t$ and $P_0$, respectively. Let $\mathcal{F}$ and $\mathcal{F}_0$ be a class of predictor functions from $\mathbb{R}^k$ to $\mathcal{Y}$ for the source tasks and target task respectively, and $\mathcal{H}$ a class of representation functions from $\mathbb{R}^d$ to $\mathbb{R}^k$.

Following Tripuraneni et al. [2020], we assume that marginal distribution of $P_j$ over $\mathcal{X}$ is the same for all the $j$ from 0 to $t$, and that there exists a common representation $h^* \in \mathcal{H}$ and task-specific predictors $f_j^* \in \mathcal{F}$ for $j \in [t]$ and $f_0 \in \mathcal{F}_0$ such that $P_j$ can be decomposed as $P_j(x, y) = P_x(x)P_{y|x}(y|f_j^* \circ h^*(x))$. Note that this decomposition does not assume that $f_j^* \circ h^*$ is the optimal in-class predictor; however, we will assume it is to make our optimistic rates more meaningful (although this is not strictly necessary). Also the decomposition implicitly assumes that $y$ depends on $x$ only via $f_j^* \circ h^*(x)$, and thus any additional noise in $y$ is independent of $x$. Note that like Tripuraneni et al. [2020], we allow the predictor class for the target task $\mathcal{F}_0$ to be different than on the source tasks $\mathcal{F}$. For example, when performing logistic regression with a linear representation

$B \in \mathbb{R}^{k \times d}$ and linear predictor $w \in \mathbb{R}^k$, we have $P_{y|x}(y = 1 | f_j^* \circ h^*(x)) = \sigma(w^T B^T x)$, where $\sigma$ is the sigmoid function.

Given a loss function $\ell : \mathbb{R} \times \mathcal{Y} \to \mathbb{R}$, the goal is to find a good predictor $\hat{f} \circ \hat{h}$ from the composite-class $\mathcal{F}_0 \circ \mathcal{H}$ with small transfer risk,

$$R_{\text{target}}(\hat{f}_0, \hat{h}) = \mathbb{E}_{(x,y) \sim P_0}[\ell(\hat{f}_0 \circ \hat{h}(x), y)].$$

Yet, as standard, we do not have access to the joint distribution for all tasks, but instead $n$ independent and identically distributed (i.i.d.) samples from each source task and $m$ i.i.d. samples for the target task. Let $(x_j^i, y_j^i)$ be the $i^{\text{th}}$ sample for the $j^{\text{th}}$ task. To accomplish the above, we use the source data to find a representation, and the target data to find a predictor for the target task. Specifically, in this MTRL procedure, first, we train on all source task data, then freeze the resulting representation. Second, we train over this frozen representation to find a good predictor on the target samples.

We formalize the above as the following two-stage Empirical Risk Minimization (ERM) procedure,

$$(\hat{\boldsymbol{f}}, \hat{h}) \in \underset{\boldsymbol{f} \in \mathcal{F}^{\otimes t}, h \in \mathcal{H}}{\arg\min} \frac{1}{nt} \sum_{j=1}^t \sum_{i=1}^n \ell(f_j \circ h(x_j^i), y_j^i), \quad \text{(Multi-task (representation) learning)} \quad (2)$$

$$\hat{f}_0 \in \underset{f \in \mathcal{F}_0}{\arg\min} \frac{1}{m} \sum_{i=1}^m \ell(f \circ \hat{h}(x_0^i), y_0^i). \quad \text{(Transfer learning)} \quad (3)$$

Besides bounding the transfer risk, we seek to understand if the above procedure also yields improved bounds for the source tasks (the problem of MTL).

$$R_{\text{source}}(\hat{\boldsymbol{f}}, \hat{h}) = \frac{1}{t} \sum_{j=1}^t \mathbb{E}_{(x,y) \sim P_j}[\ell(\hat{f}_j \circ \hat{h}(x), y)]$$

We make the following regularity assumptions.

**Assumption 1.**

- **A:** *The loss function $\ell$ is nonnegative and $b$-bounded, i.e., $0 \le \ell(y', y) \le b < \infty$ for all $y', y \in \mathcal{Y}$.*

- **B:** *All functions in $\mathcal{F}$ are $L$-Lipschitz for some $0 < L < \infty$ w.r.t. $\|\cdot\|_2$, i.e., $\|f(z_1) - f(z_2)\|_2 \le L \|z_1 - z_2\|_2$ for all $z_1, z_2 \in Dom(f)$ and $f \in \mathcal{F}$.*

- **C:** *Any $f \circ h \in \mathcal{F} \circ \mathcal{H}$ is $D$-bounded over $\mathcal{X}$ w.r.t. $\|\cdot\|_\infty$, i.e., $\sup_{x \in \mathcal{X}} |f \circ h(x)| \le D < \infty$ for all $f \in \mathcal{F}$ and $h \in \mathcal{H}$.*

Unlike Tripuraneni et al. [2020], we assume for some of our theorems that our function is smooth instead of Lipschitz, as leveraging smoothness is key to achieving fast rates [Srebro et al., 2010].

**Definition 1** ($H$-smoothness). *The loss function $\ell : \mathbb{R} \times \mathcal{Y} \to \mathbb{R}$ is $H$-smooth if $\left| \ell_y'(y_1, y) - \ell_y'(y_2, y) \right| \le H |y_1 - y_2|$ for all $y_1, y_2 \in \mathbb{R}, y \in \mathcal{Y}$.*

**Task diversity [Tripuraneni et al., 2020].** We benefit from learning a representation for a new target task, so long as there must be information encoded for that task by the source tasks. A typical way to quantify this encoded information is to control the difference between the learned representation and the underlying representation. To this end, we use the framework introduced by Tripuraneni et al. [2020] that quantifies a measure of task diversity. As mentioned in their work, this framework recovers task diversity assumptions in Du et al. [2021] exactly. In our setting, the definitions given in Tripuraneni et al. [2020] simplify considerably.

**Definition 2.** *The tasks $\{P_j\}_{i=1}^t$ are $(\nu, \epsilon)$-diverse over $P_0$, if for the corresponding $\boldsymbol{f}^* \in \mathcal{F}^{\otimes t}, f_0^* \in \mathcal{F}_0$ and representation $h^* \in \mathcal{H}$, we have that for all $h' \in \mathcal{H}$*

$$\inf_{f' \in \mathcal{F}_0} R_{\text{target}}(f', h') - R_{\text{target}}(f_0^*, h^*) \le \frac{1}{\nu} \left( \inf_{\boldsymbol{f}' \in \mathcal{F}^{\otimes t}} R_{\text{source}}(\boldsymbol{f}', h') - R_{\text{source}}(\boldsymbol{f}^*, h^*) \right) + \varepsilon.$$

Parameters $\nu$ and $\varepsilon$ quantify the similarity between learning the source tasks and the target task. For a detailed analysis of these parameters and the framework introduced by Tripuraneni et al. [2020], see Appendix B.

**Local Rademacher Complexity.** We now define the measures of complexity that are used in our work. Let $u, p, n \in \mathbb{N}$, an input space $\mathcal{Z}$, a class of vector-valued functions $\mathcal{Q} : \mathcal{Z} \to \mathbb{R}^u$, and a dataset $\mathbf{Z} = (z_j^i)_{j \in [p], i \in [n]}$ where $z_j^i \in \mathcal{Z}$. Define the *data-dependent* Rademacher width, $\tilde{\mathfrak{R}}_{\mathbf{Z}}(\cdot)$, as

$$\tilde{\mathfrak{R}}_{\mathbf{Z}}(\mathcal{Q}^{\otimes p}) = \mathbb{E}_{\sigma_{i,j,k}}\left[\sup_{\boldsymbol{q} \in \mathcal{Q}^{\otimes p}} \frac{1}{np} \sum_{i,j,k=1}^{n,p,u} \sigma_{i,j,k}\left(q_j(z_j^i)\right)_k\right],$$

where $\sigma_{i,j,k}$ are i.i.d. Rademacher random variables. Analogously, define the *data-dependent* Gaussian width, $\tilde{\mathfrak{G}}_{\mathbf{Z}}(\cdot)$, as

$$\tilde{\mathfrak{G}}_{\mathbf{Z}}(\mathcal{Q}^{\otimes p}) = \mathbb{E}_{g_{i,j,k}}\left[\sup_{\boldsymbol{q} \in \mathcal{Q}^{\otimes p}} \frac{1}{np} \sum_{i,j,k=1}^{n,p,u} g_{i,j,k}\left(q_j(z_j^i)\right)_k\right],$$

where $g_{i,j,k}$ are i.i.d. $\mathcal{N}(0,1)$ random variables. We define the *worst-case* Rademacher width as $\tilde{\mathfrak{R}}_n(\mathcal{Q}^{\otimes p}) = \sup_{\mathbf{Z} \in \mathcal{Z}^{pn}} \tilde{\mathfrak{R}}_{\mathbf{Z}}(\mathcal{Q}^{\otimes p})$ and, analogously, the *worst-case* Gaussian width as $\tilde{\mathfrak{G}}_n(\mathcal{Q}^{\otimes p}) = \sup_{\mathbf{Z} \in \mathcal{Z}^{pn}} \tilde{\mathfrak{G}}_{\mathbf{Z}}(\mathcal{Q}^{\otimes p})$. The above definitions generalize the standard Rademacher and Gaussian width for real-valued functions and can be derived as a special case of the more general *set-based* definitions [Wainwright, 2019] (see Appendix A).

We now define *local Rademacher width* as $\tilde{\mathfrak{R}}_{\mathbf{Z}}(\mathcal{Q}^{\otimes p}, r) = \tilde{\mathfrak{R}}_{\mathbf{Z}}(\{\boldsymbol{q} \in \mathcal{Q}^{\otimes p} : V(\boldsymbol{q}) \leq r\})$, where $V : \mathcal{Q}^p \to \mathbb{R}$. That is, the *local Rademacher width* is simply the Rademacher width restricted by a functional. In our applications, we consider any $V$ satisfying $V(\boldsymbol{q}) \leq b \frac{1}{np} \sum_{i,j,k=1}(q_j(z_j^i))_k$, where $b$ is the uniform bound on the range of $q$ in $\ell_2$-norm. Note this recovers the classical local Rademacher width[2] for real-valued functions and non-product spaces ($p = u = 1$). We are mostly interested in the local Rademacher width of the loss applied to the pairwise composition between our vector-valued hypothesis class and our representation class. Specifically, if we define such a class as $\mathcal{L}_\ell(\mathcal{F}^{\otimes t}(\mathcal{H})) = \{(\ell \circ \boldsymbol{f}_1 \circ h, \ldots, \ell \circ \boldsymbol{f}_t \circ h) \mid h \in \mathcal{H}, \boldsymbol{f} \in \mathcal{F}^{\otimes t}\}$, then we seek to bound $\tilde{\mathfrak{R}}_n(\mathcal{L}_\ell(\mathcal{F}^{\otimes t}(\mathcal{H})), r)$. Crucially, our bound will be a *sub-root function* in $r$, where we define this type of function below.

**Definition 3** (Sub-root Function). *A function* $\psi : [0, \infty) \to [0, \infty)$ *is sub-root if it is nonnegative, nondecreasing, and if* $r \mapsto \psi(r)/\sqrt{r}$ *is nonincreasing for* $r > 0$.

Sub-root functions have the desirable properties of always being continuous and having a unique fixed point, i.e., $r^* = \psi(r^*)$ is only satisfied by some $r^*$; see Lemma 8 in Appendix A.

## 3 Main Results

In this section, we detail our local Rademacher complexity results for transfer learning (TL) and MTL via MTRL, a bound on the local Rademacher complexity of smooth bounded non-negative losses, and the application of this bound to both TL and MTRL.

Our first result is a local Rademacher complexity result for TL via MTRL.

**Theorem 1.** *Let* $\hat{h}$ *and* $\hat{f}_0$ *be the learned representation and target predictor, as described in Eqns.* (2) *and* (3). *Under Assumption 1.A, if* $\boldsymbol{f}^*$ *is* $(\nu, \epsilon)$*-diverse over* $\mathcal{F}_0$ *w.r.t.* $h^*$ *and let* $\psi_1$ *and* $\psi_2$ *be sub-root functions such that* $\psi_1(r) \geq \mathfrak{R}_n(\ell \circ \mathcal{F}^{\otimes t} \circ \mathcal{H}, r)$ *and* $\psi_2(r) \geq \mathfrak{R}_m(\ell \circ \mathcal{F}_0, r)$, *then with probability at least* $1 - 4e^{-\delta}$, *the transfer learning risk is upper-bounded by*

$$R_{target}(\hat{f}_0, \hat{h}) \leq R_{target}(f_0^*, h^*) + c_1\left(\sqrt{R_{\text{target}}(f_0^*, h^*)}\left(\sqrt{\frac{b\delta}{m}} + \sqrt{\frac{r_1^*}{b}}\right) + \frac{b\delta}{m} + \frac{r_1^*}{b}\right)$$

$$+ \frac{1}{\nu}\left(c_2\left(\sqrt{R_{\text{source}}(\boldsymbol{f}^*, h^*)}\left(\sqrt{\frac{b\delta}{nt}} + \sqrt{\frac{r_2^*}{b}}\right) + \frac{b\delta}{nt} + \frac{r_2^*}{b}\right)\right) + \varepsilon, \quad (4)$$

*where* $r_1^*$ *and* $r_2^*$ *are the fixed points of* $\psi_1(r)$ *and* $\psi_2(r)$, *respectively, and* $c_1$ *and* $c_2$ *are absolute constants.*[3]

---

[2]Bartlett et al. [2005] uses "complexity", but we use "width", as common in stochastic processes.

[3]For exact constants for each term, please refer to Appendix D.

Inequality (4) bounds the transfer risk in terms of task diversity parameters $\nu$ and $\epsilon$; local Rademacher complexity parameters, fixed points $r_1^*$ and $r_1^*$; the minimum risk for the target task; and the minimum average risk for the source tasks.

As discussed in Xu and Tewari [2021], there exist settings where the task diversity parameters $\nu$ and $\epsilon$ are favorable, i.e., $\epsilon = 0$ and $\nu = \Theta(1)$, so we will disregard them in the subsequent discussion. The bound is an optimistic rate because it interpolates between $\sqrt{r_1^*} + \sqrt{r_2^*}$ and $r_1^* + r_2^*$, depending on both $R_{\text{target}}(f_0^*, h^*)$ and $R_{\text{source}}(\boldsymbol{f}^*, h^*)$.

The fixed point $r_1^*$ is a function of $\mathcal{F}^{\otimes t}(\mathcal{H})$ and encodes its complexity measured with respect to the number of samples for the source tasks (among other class-specific parameters). The same reasoning holds for $r_2^*$ w.r.t. $\mathcal{F}_0$.

We consider $\mathcal{H}$ to be *complex* and both $\mathcal{F}$ and $\mathcal{F}_0$ to be *simple*. So, if we instead learned the target task directly, then we would pay the complexity of $\mathcal{F}_0(\mathcal{H})$ against $m$ samples. In contrast, in our bound, we only pay the complexity of $\mathcal{F}_0$ for $m$ samples and the complexity of $\mathcal{F}^{\otimes t}(\mathcal{H})$ for $nt$ samples. Since in many settings $nt$ is much larger than $m$, we are successfully leveraging our representation to learn the target task.

The dependence on minimum risk is useful in various settings. An instructive regime is when $nt \gg m$ with $R_{\text{target}}(f_0^*, h^*) = 0$ and $R_{\text{source}}(\boldsymbol{f}^*, h^*) > 0$; in this case, we achieve a bound of $\sqrt{r_1^* + r_2^*}$. This demonstrates that data-abundant environments can be leveraged to learn representations that work well in small sample regimes.

The next result uses a decoupling of function classes, the Gaussian chain rule of Tripuraneni et al. [2020], and smoothness to bound $r_1^*$ and $r_2^*$. We see that under this additional assumption that $\ell$ is $H$-smooth, $r_1^*$ and $r_2^*$ decay quickly.

**Theorem 2.** *Under the setting of Theorem 1 along with $\ell$ being $H$-smooth and Assumptions 1.B and 1.C, with probability at least $1 - 6e^{-\delta}$, the fixed points in Inequality (4),*

$$r_1^* \leq c_3 b \left( H \tilde{\mathfrak{G}}_m^2(\mathcal{F}_0 \circ \hat{h}) \log^2(m) + (1 + \delta)\frac{b}{m} \right), and$$

$$r_2^* \leq c_4 b \left( \left( L^2 \tilde{\mathfrak{G}}_{nt}^2(\mathcal{H}) + \tilde{\mathfrak{G}}_n^2(\mathcal{F}) \right) H \log(nt)^4 + \frac{D^2 H \log^2(nt) + b(1 + \delta)}{nt} \right),$$

*where $c_3$ and $c_4$ are absolute constants.*

**Remark 1.** *Note that in Theorem 2 it is not necessary to apply the chain rule. If the chain rule is not used, Assumptions 1.B and 1.C is not needed. The fixed point $r_2^*$ would be bounded as a function of $\mathcal{F}^{\otimes t}(\mathcal{H})$. In the following, we use the above decomposed form for interpretability.*

Note that for constant $L, D, H, b$, and $\delta$, the terms on the right are *always* fast, so the rate of decay depends crucially on the behavior of the Gaussian widths for each class. For many function classes used in machine learning, it is common that $\mathfrak{G}_{nt}(\mathcal{H}) \approx \sqrt{\frac{C(\mathcal{H})}{nt}}$ and $\mathfrak{G}_n(\mathcal{F}) \approx \sqrt{\frac{C(\mathcal{F})}{n}}$, where $C(\cdot)$ is some notion of complexity of the function class, which could be, for instance, the VC dimension, pseudo-dimension, fat-shattering dimension, etc. Therefore, it is common that

$$L^2 \tilde{\mathfrak{G}}_{nt}^2(\mathcal{H}) + \tilde{\mathfrak{G}}_n^2(\mathcal{F}) \approx L^2 \frac{C(\mathcal{H})}{nt} + \frac{C(\mathcal{F})}{n} \ \text{ and } \ \tilde{\mathfrak{G}}_n^2(\mathcal{F}_0 \circ \hat{h}) \approx \frac{C(\mathcal{F}_0)}{m}.$$

Recall, from our discussion following Theorem 1, that these fixed points control the order of the rate. Therefore, this theorem interpolates between a rate of $1/\sqrt{m} + 1/\sqrt{nt}$ and $1/m + 1/nt$, where the fast rate is achieved in a realizable setting.

**Linear classes.** As an example let us consider a linear projection onto a lower dimensional space along with linear regressors and squared loss. Let $\mathcal{X} = \mathbb{R}^d, \mathcal{Y} = \mathbb{R}$, and

$$\mathcal{F} = \left\{ f \mid f(\mathbf{z}) = \boldsymbol{\alpha}^\top \mathbf{z}, \boldsymbol{\alpha} \in \mathbb{R}^k, \|\boldsymbol{\alpha}\| \leq c \right\},$$

$$\mathcal{H} = \left\{ \mathbf{h} \mid \mathbf{h}(\mathbf{x}) = \mathbf{B}^\top \mathbf{x}, \mathbf{B} \in \mathbb{R}^{d \times k}, \mathbf{B} \text{ is a matrix with orthonormal columns } \right\}.$$

If we assume that $P_x$ is sub-Gaussian, then standard arguments (see Tripuraneni et al. [2020]) show that $\tilde{\mathfrak{G}}_m^2(\mathcal{F}_0 \circ \hat{h}) \leq \mathcal{O}(\frac{k}{m})$, $\tilde{\mathfrak{G}}_n^2(\mathcal{F}) \leq \mathcal{O}(\frac{k}{n})$, and $\tilde{\mathfrak{G}}_{nt}^2(\mathcal{H}) \leq \mathcal{O}(\frac{k^2 d}{nt})$. To simplify the bounds let $L = 1, H = 1, D = 1, b = 1$, and $\delta = 0.05$. Thus, under the assumptions of Theorem 2, the fixed points are bounded by

$$r_1^* \lesssim \left( \frac{k}{m} + \frac{1}{m} \right), \text{ and } r_2^* \lesssim \left( \frac{k^2 d}{nt} + \frac{k}{n} + \frac{1}{nt} \right),$$

which gives

$$R_{\text{target}}(\hat{f}_0, \hat{h}) - R_{\text{target}}(f_0^*, h^*) \lesssim \left( \sqrt{R_{\text{target}}(f_0^*, h^*)} \left( \sqrt{\frac{k}{m}} \right) + \frac{k}{m} \right)$$
$$+ \frac{1}{\nu} \left( \left( \sqrt{R_{\text{source}}(\boldsymbol{f}^*, h^*)} \left( \sqrt{\frac{k^2 d}{nt} + \frac{k}{n}} \right) + \frac{k^2 d}{nt} + \frac{k}{n} \right) \right) + \varepsilon.$$

The following result is such a bound on the local Rademacher complexity for a non-negative $H$-smooth loss and any function class $\mathcal{H}$.

**Proposition 1** (Smooth non-negative local Rademacher complexity bound). *Under the setting of Theorem 1 along with $\ell$ being $H$-smooth and Assumptions 1.B and 1.C, there exists an absolute constants $c_3$ and $c_4$ such that*

$$\mathfrak{R}_{nt}(\mathcal{L}_\ell(\mathcal{F}^{\otimes t}(\mathcal{H})), r) \le c_3 \sqrt{r} \Big( G\left(\mathcal{F}^{\otimes t}(\mathcal{H})\right) \sqrt{H} \log(n)^2 + \frac{D\sqrt{H}\log(n)}{\sqrt{n}} + \sqrt{\frac{b}{nt}} \Big), \quad (5)$$

$$\mathfrak{R}_{nt}(\mathcal{L}_\ell(\mathcal{F}^{\otimes t}(\mathcal{H})), r) \le c_4 \sqrt{r} \Big( \Pi(\mathcal{F}^{\otimes t}(H)) \log\left( \frac{e\sqrt{b}}{\Pi(\mathcal{F}^{\otimes t}(H))} \right) + \sqrt{\frac{b}{nt}} \Big), \quad (6)$$

*where $G(\mathcal{F}^{\otimes t}(\mathcal{H})) = L\tilde{\mathfrak{G}}_{nt}(\mathcal{H}) + \tilde{\mathfrak{G}}_n(\mathcal{F})$ and $\Pi(\mathcal{F}^{\otimes t}(H)) = \sqrt{H} G\left(\mathcal{F}^{\otimes t}(\mathcal{H})\right) \log\left( \frac{eD}{G(\mathcal{F}^{\otimes t}(\mathcal{H}))} \right).$*

When we specialize the above bound to the standard single-task non-compositional setting and compare with Srebro et al. [2010], then

$$\tilde{\mathfrak{R}}_n(\mathcal{L}_\ell(\mathcal{F}, r)) \le c \tilde{\mathfrak{G}}_n(\mathcal{F}) \sqrt{Hr} \log\left( \frac{e\sqrt{b}}{\tilde{\mathfrak{G}}_n(\mathcal{F})\sqrt{H}} \right)$$

$$\le c \tilde{\mathfrak{R}}_n(\mathcal{F}) \sqrt{Hr} \sqrt{\log(n)} \log\left( \frac{e\sqrt{b}}{\tilde{\mathfrak{R}}_n(\mathcal{F})\sqrt{H\log(n)}} \right)$$

$$\le c' \tilde{\mathfrak{R}}_n(\mathcal{F}) \sqrt{Hr} \sqrt{\log(n)} \log\left( \frac{eb}{H\log(n)} \frac{n}{B^2} \right).$$

The second inequality uses the relation $\tilde{\mathfrak{G}}_n(\mathcal{F}) \le 2\sqrt{\log(n)}\tilde{\mathfrak{R}}_n(\mathcal{F})$ [Wainwright, 2019, p. 155] and that $x \mapsto x \log eC/x$ is increasing in $x$ until $x = C$. The third inequality uses Khintchine's inequality, $\tilde{\mathfrak{R}}_n(\mathcal{F}) \ge \frac{\sqrt{2}B}{\sqrt{n}}$, where $B$ is the $\ell_2$-bound on the range of $\mathcal{F}$. Thus, the upper bound above is always better than the bound in Srebro et al. [2010][4] for large enough $n$. Furthermore, under the additional though well-studied assumption that $\ell(0) \le HB^2$ [Arora et al., 2022, Shamir, 2015], ours is better for $n = \Omega(1)$. Finally, the Gaussian and Rademacher widths are of the same order for the class of linear predictors, bounded in norm, by a strongly convex regularizer, in general non-Euclidean settings. Hence, in this setting, our bound is smaller by a factor of $\sqrt{\log n}$ in Kakade et al. [2008] – see Appendix E for a detailed comparison.

### 3.1 Multi-Task Learning

Next, we detail some theorems foundational to the results above. For those results, we were able to link MTL with MTRL by using the task diversity assumption. Therefore, we naturally have the following local Rademacher result for MTL.

**Theorem 3.** *Let $(\hat{\boldsymbol{f}}, \hat{h})$ be an empirical risk minimizer of $\hat{R}_{source}(\cdot, \cdot)$ as given in Eqn. (2). Under Assumption 1.A, let $\psi$ be a sub-root function such that $\psi(r) \ge \mathfrak{R}_n(\mathcal{F}^{\otimes t}(\mathcal{H}), r)$ with $r^*$ the fixed point of $\psi(r)$, then with probability $1 - 2e^{-\delta}$,*

$$R_{source}(\hat{\boldsymbol{f}}, \hat{h}) \le R_{source}(\boldsymbol{f}^*, h^*) + c\Big( \sqrt{R_{source}(\boldsymbol{f}^*, h^*)}\Big( \sqrt{\frac{b\delta}{nt}} + \sqrt{\frac{r^*}{b}} \Big) + \frac{b\delta}{nt} + \frac{r^*}{b} \Big), \quad (7)$$

*where $c$ is an absolute constant.*

---

[4]We note that Srebro et al. [2010] incur an additional $\sqrt{rb/n}$ term when converting from complexity to width

As above, we can use our bound on the local Rademacher complexity of a $H$-smooth loss class to get the following result. This result is similar to Srebro et al. [2010], Theorem 1, yet ours is in an MTL via MTRL setting.

**Theorem 4.** *Under the setting of Theorem 3 along with $\ell$ being $H$-smooth and Assumptions 1.B and 1.C, with probability at least $1 - 3e^{-\delta}$, $r^*$ of $\psi(r)$ is bounded by*

$$cb\Big( \Big( L^2 \tilde{\mathfrak{G}}_{nt}^2(\mathcal{H}) + \tilde{\mathfrak{G}}_n^2(\mathcal{F}) \Big) H \log(nt)^4 + \frac{D^2 H \log(nt)^2}{nt} + \frac{b(1+\delta)}{nt} \Big),$$

*where $c$ is an absolute constant.*

## 3.2 Local Rademacher complexity chain rule

Using the chain rule of Tripuraneni et al. [2021], as we did above, to decouple the complexities of the representation and prediction classes is desirable. Yet, our general local Rademacher complexity Theorems 1 and 3, as stated, do not seem to have this property. Consequently, we develop a *local chain rule*, which aims to separate the *local complexities* of learning the representation and predictor. The main result is the following.

**Theorem 5.** *Suppose the loss function $\ell$ is $L_\ell$-Lipschitz. Define the restricted representation and predictor classes as follows,*

$$\ell \circ \mathcal{F}_{\mathbf{X}}(r) := \big\{ \ell \circ \boldsymbol{f} \in \ell \circ \mathcal{F}^{\otimes t} : \exists h \in \mathcal{H} : V(\ell \circ \boldsymbol{f} \circ h) \leq r \big\}$$

$$\mathcal{H}_{\mathbf{X}}(r) := \big\{ h \in \mathcal{H} : \exists \boldsymbol{f} \in \mathcal{F}^{\otimes t} : V(\ell \circ \boldsymbol{f} \circ h) \leq r \big\},$$

*where $V$ is the functional in the local Rademacher complexity description. Under Assumptions 1.B and 1.C and that the worst-case Gaussian width of the above is bounded by the sub-root functions $\psi_\mathcal{F}$ and $\psi_\mathcal{H}$, respectively, there exists an absolute constant $c$ such that*

$$\tilde{\mathfrak{G}}_n(\mathcal{L}_\ell(\mathcal{F}^{\otimes t}(\mathcal{H}), r)) \leq c\Big( (L L_\ell \psi_\mathcal{F}(r) + \psi_\mathcal{H}(r)) \log(nt) + \frac{D}{(nt)^2} \Big).$$

# 4 Proof Techniques

In this section, we give details on some of the tools we use by giving proof sketches for the results in Section 3. In the next section, we cover some theorems at the root of all our results.

**Proof sketch for Theorem 1.** Let $\tilde{f}_0 = \arg\min_{f \in \mathcal{F}} R_{\text{target}}(f, \hat{h})$. By a risk decomposition, we can bound the excess transfer risk by $R_{\text{target}}(\hat{f}_0, \hat{h}) - R_{\text{target}}(\tilde{f}_0, \hat{h}) + \sup_{f_0 \in \mathcal{F}_0} \inf_{f' \in \mathcal{F}} \{ R_{\text{target}}(f', \hat{h}) - R_{\text{target}}(f_0, h^*) \}$. We can now use Theorem 3, with $t = 1$, to bound $R_{\text{target}}(\hat{f}_0, \hat{h})$. Note that $R_{\text{target}}(f_0^*, \hat{h}) - R_{\text{target}}(\tilde{f}_0, \hat{h}) \leq 0$. Now $\inf_{f' \in \mathcal{F}} R_{\text{target}}(f', \hat{h}) - R_{\text{target}}(f_0^*, h^*)$ is less than $\frac{1}{\nu}(\inf_{\boldsymbol{f}' \in \mathcal{F}^{\otimes t}} R_{\text{source}}(\boldsymbol{f}', \hat{h}) - R_{\text{source}}(\boldsymbol{f}^*, h^*) + \varepsilon$, by the task-diversity assumption. As shown in Tripuraneni et al. [2020], we can bound $\inf_{\boldsymbol{f}' \in \mathcal{F}^{\otimes t}} \{ R_{\text{source}}(\boldsymbol{f}', \hat{h}) - R_{\text{source}}(\boldsymbol{f}^*, h^*) \}$ with $R_{\text{source}}(\hat{\boldsymbol{f}}, \hat{h})$. We can then apply Theorem 3 to the remaining $R_{\text{source}}(\hat{\boldsymbol{f}}, \hat{h})$ term. By leveraging task diversity again we can bound the remaining $R_{\text{target}}(f_0^*, \hat{h})$ as a function of $R_{\text{target}}(f_0^*, h^*)$.

**Proof sketch for Proposition 1.** We achieve the following bound on the local Rademacher complexity of the loss class by using a truncated version of Dudley's integral [Srebro et al., 2010, see. Lemma A.3].

$$\mathfrak{R}_{nt}(\mathcal{L}_\ell(\mathcal{F}^{\otimes t}(\mathcal{H})), r) \leq \inf_{0 \leq \alpha \leq \sqrt{br}} \Big\{ 4\alpha + 10(nt)^{-1/2} \int_\alpha^{\sqrt{br}} \sqrt{\log \mathcal{N}_2(\mathcal{L}_\ell(\mathcal{F}^{\otimes t}(\mathcal{H})), r), \varepsilon, nt)} \Big\}$$

We observe that we can use smoothness to move the dependence on the loss to an appropriate scaling, which gives that $\mathcal{N}_2(\mathcal{L}_\ell(\mathcal{F}^{\otimes t}(\mathcal{H})), r), \varepsilon, nt) \leq \mathcal{N}_2(\mathcal{F}^{\otimes t}(\mathcal{H}), \varepsilon/\sqrt{12Hrnt}, nt)$ (Lemma 18 in the appendix). Next, we apply Sudakov minoration to bound the log covering number by the Gaussian width of our function class: $\sqrt{\log \mathcal{N}_2(\mathcal{F}^{\otimes t}(\mathcal{H}), \varepsilon/\sqrt{12Hrnt}, nt)} \leq \frac{\sqrt{12Hrnt}}{\epsilon} \tilde{\mathcal{G}}_n(\mathcal{F}^{\otimes t}(\mathcal{H}))$. With these simplifications, the parameter $\alpha$ from Dudley's integral can either be solved for exactly (as in Inequality 6), so long as $\alpha \leq \sqrt{br}$, or set in such a way that it always holds (like in Inequality 5). Finally, we can optionally apply the Gaussian chain rule from Tripuraneni et al. [2020] to bound $\tilde{\mathcal{G}}_n(\mathcal{F}^{\otimes t}(\mathcal{H}))$ with $L \tilde{\mathfrak{G}}_{nt}(\mathcal{H}) + \tilde{\mathfrak{G}}_n(\mathcal{F})$.

**Proof sketch for Theorem 2.** The right hand side of Inequality 5 is a sub-root function in $r$, because it is an affine transformation of $\sqrt{r}$. Therefore, we just solve for the fixed point of this sub-root function, i.e., solve for $r^*$ w.r.t. the equation below.

$$c\sqrt{r^*}\Big(G\left(\mathcal{F}^{\otimes t}(\mathcal{H})\right)\sqrt{H}\log(n)^2 + \frac{D\sqrt{H}\log(n)}{\sqrt{n}} + \sqrt{\frac{b}{nt}}\Big) = r^*. \tag{8}$$

All that remains of the proof is basic algebraic simplifications. The proof for Inequality 6 is similar.

## 5 Discussion

The proofs of the above results hinge on the following local Rademacher complexity result.

**Theorem 6.** *Let $\mathcal{F}$ be a class of non-negative $b$ bounded functions. Assume that $X_j^1, \ldots, X_j^n$ are $n$ draws from $P_j$ for $j \in [t]$ and all $nt$ samples are independent. Suppose that $\mathrm{Var}\big[\boldsymbol{f}\big] \leq V(\boldsymbol{f}) \leq BP\boldsymbol{f}$, where $V$ is a functional from $\mathcal{F}^{\otimes t}$ to $\mathbb{R}$ and $B > 0$.*

*Let $\psi$ be a sub-root function with fixed point $r^*$ such that $B\Re(\mathcal{F}^{\otimes t}, r) \leq \psi(r)$ for all $r \geq r^*$, then for $K > 1$ and $\delta > 1$, with probability at least $1 - e^{-\delta}$,*

$$\forall \boldsymbol{f} \in \mathcal{F}^{\otimes t} \quad P\boldsymbol{f} \leq \frac{K}{K-1}\hat{P}^n\boldsymbol{f} + 200(1+\alpha)^2\frac{Kr^*}{B} + \frac{5}{4}\frac{BK\delta}{nt} + 2\Big(\frac{1}{\alpha} + \frac{1}{3}\Big)\frac{b\delta}{nt}.$$

Yousefi et al. [2018], Theorem 9, showed a result of this form in a more general setting of Bernstein classes and a function class that is not necessarily a product space. In contrast, for simplicity, we do not consider Bernstein classes, though we believe such a generalization is possible, and our function classes are product spaces. By doing so, we achieve better constants than Yousefi et al. [2018] by using the next theorem.

**Theorem 7** (Concentration with $t$ functions). *Let $\mathcal{F}$ be a class of functions from $\mathcal{X}$ to $\mathbb{R}$ and assume that all functions $f$ in $\mathcal{F}$ are $P_j$-measurable for all $j \in [t]$, square-integrable, and satisfy $\mathbb{E}f = 0$. Let $\sup_{f \in \mathcal{F}} \mathrm{ess\,sup}\, f \leq 1$ and $Z = \sup_{\boldsymbol{f} \in \mathcal{F}^{\otimes t}} \sum_{j=1}^{t} \sum_{i=1}^{n} f_j(X_j^i)$. Let $\sigma > 0$ such that $\sigma^2 \geq \sup_{\boldsymbol{f} \in \mathcal{F}^{\otimes t}} \frac{1}{t}\sum_{j=1}^{t} \mathrm{Var}\big[f_j(X_j^1)\big]$ almost surely. Then, for all $\delta \geq 0$, we have that*

$$\mathbb{P}\{Z \leq \mathbb{E}Z + \sqrt{2\delta(tn\sigma^2 + 2\mathbb{E}Z)} + \frac{\delta}{3})\} \leq e^{-\delta}.$$

*This result also holds for $\sup_{\boldsymbol{f} \in \mathcal{F}^{\otimes t}} \Big|\sum_{j=1}^{t}\sum_{i=1}^{n} f_j(X_j^i)\Big|$ under the condition that $\sup_{f \in \mathcal{F}} \|f\|_\infty \leq 1$.*

Yousefi et al. [2018] also proves something similar to the above from scratch using "Logarithmic Sobolev" inequalities. We make the observation that within our setting a proof similar to the original proof given by Bousquet [2002] still holds. To clarify this further, Yousefi et al. [2018] state:

> Note that the difference between the constants in (2) [Bousquet, 2002, Theorem 2.3] and (3) [Yousefi et al., 2018, Theorem 1] is due to the fact that we were unable to directly apply Bousquet's version of Talagrand's inequality (like it was done in Bartlett et al. [2005] for scalar-valued functions) to the class of vector-valued functions.

We show that within the product space structure, which we consider to be the most natural for MTL, it is possible to achieve the same constants. Concretely, when $t = 1$, we realize the single function version of Bousquet [2002] exactly.

The proof of Theorem 6 is similar to the one given in Bartlett et al. [2005], Sec. 3, and generalized to MTL by Yousefi et al. [2018], Appendix B. Indeed, we follow the same steps within these proofs but use our inequality above.

## 6 Conclusion

We provide the first optimistic rates for transfer learning and multi-task learning via multi-task representation learning. This is achieved by establishing a local Rademacher complexity result for multi-task representation learning. We provide distribution-free optimistic rates for smooth non-negative losses by bounding the local Rademacher complexity for multi-task loss classes. Besides our contributions to multi-task representation learning and multi-task learning, our work provides several foundational results and improved rates in the standard single-task setting. Directions for future work include: exploring adversarial robustness, active learning, and fine-tuning of the representation within multi-task representation learning.

## Acknowledgements

This research was supported, in part, by DARPA GARD award HR00112020004, NSF CAREER award IIS-1943251, funding from the Institute for Assured Autonomy (IAA) at JHU, and the Spring'22 workshop on "Learning and Games" at the Simons Institute for the Theory of Computing.

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

# Appendices

## Contents

# A  Additional preliminaries on Gaussian processes and complexities

## A.1  Preliminaries

We recall basic preliminaries on (sub) Gaussian processes [Vershynin, 2018, Wainwright, 2019]. Given a set $\Theta \subseteq \mathbb{R}^k$, a random process is a collection of random variables $\{Z_\theta\}_{\theta \in \Theta}$ defined on the same probability space, indexed by elements of set $\Theta$. The random process induces the following canonical psuedometric on the (abstract) set $\Theta$, defined as $d(\theta, \theta') = \|Z_\theta - Z_{\theta'}\|_{L_2} = \sqrt{\mathbb{E}\left(Z_\theta - Z_{\theta'}\right)^2}$, (pseudo)metrizing the set $T$. This allows us to define the covering and packing number of the set $\Theta$ with respect to the metric $d$, denoted as $\mathcal{N}(\Theta, d, \epsilon)$ and $\mathcal{M}(\Theta, d, \epsilon)$ respectively.

**Definition 4** (Covering number). *An $\epsilon$-cover of a set $\Theta$ with respect to a metric $d$, is a set $C \subseteq \Theta$ such that for any $\theta \in \Theta$, there exists a $c \in C$, such that $d(\theta, c) \leq \epsilon$. The $\epsilon$-covering number, $\mathcal{N}(\Theta, d, \epsilon)$ is the cardinality of the smallest $\epsilon$-cover.*

**Definition 5** (Packing number). *An $\epsilon$-packing of a set $\Theta$ with respect to a metric $d$, is a set $P \subseteq \Theta$ such that for all $p, q \in P$, $p \neq q$, $d(p, q) > \epsilon$. The $\epsilon$-packing number, $\mathcal{M}(\Theta, d, \epsilon)$ is the cardinality of the largest $\epsilon$-packing.*

These are related as follows.

**Lemma 1** (Wainwright [2019]). *For any totally-bounded set $\Theta$ and $\epsilon > 0$,*

$$\mathcal{M}(\Theta, d, 2\epsilon) \leq \mathcal{N}(\Theta, d, \epsilon) \leq \mathcal{M}(\Theta, d, \epsilon)$$

A well-studied problem is to control the process uniformly, $\mathbb{E}[\sup_{\theta \in \Theta} Z_\theta]$ in terms of the geometric properties of the set $\Theta$.

A random process is a Gaussian process, if for all finite sets $\Theta_0 \subseteq \Theta$, the distribution of the random vector $\{Z_\theta\}_{\theta \in \Theta_0}$ is Gaussian. A well-known result regarding controlling $\mathbb{E}[\sup_{\theta \in \Theta} Z_\theta]$ is the Sudakov minoration inequality.

**Lemma 2** (Sudakov minoration [Wainwright, 2019]). *Let $\{Z_\theta, \theta \in \Theta\}$ be a zero-mean Gaussian process. Then*

$$\mathbb{E}\left[\sup_{\theta \in \Theta} Z_\theta\right] \geq \sup_{\alpha > 0} \frac{\alpha}{2} \sqrt{\log \mathcal{M}(\Theta, d, \alpha)},$$

An important example is the so-called *canonical Gaussian process*, $Z_\theta = \langle \theta, g \rangle$, where $g \sim \mathcal{N}(0, \mathbb{I}_k)$ is the standard Normal vector. In this case, the canonical metric simply becomes the $\ell_2$-norm, $d(\theta, \theta') = \|\theta - \theta'\|_2$. The uniform control of the random process becomes a standard object, known as **Gaussian width** of the set $\Theta$.

$$\tilde{\mathfrak{G}}(\Theta) = \mathbb{E}\left[\sup_{\theta \in \Theta} Z_\theta\right] = \mathbb{E}_g\left[\sup_{z \in \Theta} \langle g, \theta \rangle\right].$$

The celebrated Fernique-Talagrand theory of majorizing measures establishes a characterization of the Gaussian width of a set in terms of its metric geometry – see Vershynin [2018] for more details.

A related concept is that of **Gaussian complexity**, where the goal is to control the process **absolutely** and uniformly, defined as,

$$\mathfrak{G}(\Theta) = \mathbb{E}\left[\sup_{z \in \Theta} |Z_\theta|\right] = \mathbb{E}_g\left[\sup_{z \in T} |\langle g, \theta \rangle|\right].$$

These two are related as follows.

**Lemma 3** (Vershynin [2018]). *The following are true about the relationship between Gaussian width and complexity.*

1. *$\tilde{\mathfrak{G}}(\Theta) \leq \mathfrak{G}(\Theta)$.*

2. *For any $\theta \in \Theta$, we have $\frac{1}{3}\left(\tilde{\mathfrak{G}}(\Theta) + \|\theta\|\right) \leq \mathfrak{G}(\Theta) \leq 2\left(\tilde{\mathfrak{G}}(\Theta) + \|\theta\|\right)$.*

An important category, generalizing Gaussian process, is that of sub-Gaussian processes, in which for all $\theta, \theta' \in \Theta$, the random variable $Z_\theta - Z_{\theta'}$ is $d(\theta, \theta')$-sub-Gaussian,

$$\mathbb{E} \exp\left(\lambda(Z_\theta - Z_{\theta'})^2\right) \leq \exp\left(\frac{\lambda^2 d(\theta, \theta')^2}{2}\right) \quad \text{for all } \lambda.$$

The supremum of sub-Gaussian process is upper bound by Dudley's entropy integral formula, as follows.

**Lemma 4** (Refined Dudley's entropy integral formula Wainwright [2019]). *Let $\{Z_\theta\}_{\theta \in \Theta}$ be a zero-mean sub-Gaussian process indexed on set $\Theta$ with diameter $B$, then*

$$\mathbb{E}[\sup_{\theta \in \Theta} Z_\theta] \leq \mathbb{E}[\sup_{\theta, \theta' \in \Theta} Z_\theta - Z_{\theta'}] \leq \inf_{0 \leq \alpha \leq B} \left\{ 4\alpha + 10 \int_\alpha^B \sqrt{\log \mathcal{N}(\Theta, d, \alpha)} d\epsilon \right\}.$$

An important example of sub-Gaussian process is the so-called Bernoulli/Rademacher process, where $Z_\theta = \langle \sigma, \theta \rangle$, where $\sigma \in \{-1, 1\}^k$, whose co-ordinates are i.i.d. Rademacher random variables. This leads to the notions of Rademacher width $\tilde{\mathfrak{R}}(\Theta)$ and Rademacher complexity, $\mathfrak{R}(\Theta)$, defined as follows.

$$\tilde{\mathfrak{R}}(\Theta) = \mathbb{E}_\sigma \left[ \sup_{\theta \in \Theta} \langle \sigma, \theta \rangle \right] \qquad \tilde{\mathfrak{R}}(\Theta) = \mathbb{E}_\sigma \left[ \sup_{\theta \in \Theta} |\langle \sigma, \theta \rangle| \right].$$

The Rademacher width and complexity are related in the same way as Gaussian width and complexity (akin to Lemma 3).

**Lemma 5** (Vershynin [2018]). *The following is true about the relationship between Rademacher width and complexity.*

1. $\tilde{\mathfrak{R}}(\Theta) \leq \mathfrak{R}(\Theta)$

2. *For any $\theta \in \Theta$, we have* $\frac{1}{3} \left( \tilde{\mathfrak{R}}(\Theta) + \|\theta\| \right) \leq \mathfrak{R}(\Theta) \leq 2 \left( \tilde{\mathfrak{R}}(\Theta) + \|\theta\| \right)$.

Further, the Gaussian and Rademacher widths are related as follows.

**Lemma 6** (Wainwright [2019]). *For a set $\Theta \subset \mathbb{R}^k$,*

$$\sqrt{\frac{2}{\pi}} \tilde{\mathfrak{R}}(\Theta) \leq \tilde{\mathfrak{S}}(\Theta) \leq 2\tilde{\mathfrak{R}}(\Theta)\sqrt{\log k}.$$

Akin to the Fernique-Talagrand majorizing measures theory for Gaussian processes, a complete characterization of the expected suprema for Bernoulli/Rademacher processes in terms of metric and co-ordinate geometry is obtained via (the now proved) Talagrand's Bernoulli conjecture Bednorz and Latała [2014], Ledoux and Talagrand [1991].

**Complexity of real-valued function classes.** Relatedly, another well-studied concept is Gaussian/Rademacher width/complexity of a set of **real-valued functions** on a fixed set of inputs. We first define it and then argue how this is simply a special case of the above "set"-view of these. Consider a class of functions $\mathcal{Q}$ which map an abstract set $\mathcal{Z}$ to $\mathbb{R}$. Given input $\mathbf{Z}$ consisting of $n$ points in the domain of $q \in \mathcal{Q}$, the Gaussian/Rademacher complexity and width of $\mathcal{Q}$, on $\mathbf{Z}$ are defined as follows.

$$\tilde{\mathfrak{R}}_\mathbf{Z}(\mathcal{Q}) = \mathbb{E}\left[ \sup_{q \in \mathcal{Q}} \frac{1}{n} \sum_{i=1}^n \sigma_i q(z_i) \right], \qquad \mathfrak{R}_\mathbf{Z}(\mathcal{Q}) = \mathbb{E}\left[ \sup_{q \in \mathcal{Q}} \left| \frac{1}{n} \sum_{i=1}^n \sigma_i q(z_i) \right| \right],$$

$$\tilde{\mathfrak{S}}_\mathbf{Z}(\mathcal{Q}) = \mathbb{E}\left[ \sup_{q \in \mathcal{Q}} \frac{1}{n} \sum_{i=1}^n g_i q(z_i) \right], \qquad \mathfrak{S}_\mathbf{Z}(\mathcal{Q}) = \mathbb{E}\left[ \sup_{q \in \mathcal{Q}} \left| \frac{1}{n} \sum_{i=1}^n g_i q(z_i) \right| \right],$$

where $\sigma_i$ and $g_i$ are i.i.d. Rademacher and standard Normal random variables, respectively. The metric induced on the set $\mathcal{Q}$ is the the $L_2$ distance with respect to the empirical measure on input $\mathbf{Z}$,

$$d_\mathbf{Z}(q, q') = \|q - q'\|_{L^2(\mathbf{Z})} = \sqrt{\frac{1}{n} \sum_{i=1}^n (q(z_i) - q(z_i))^2}.$$

The corresponding covering and packing numbers of $\mathcal{Q}$ are denoted as $\mathcal{N}(\mathcal{Q}, \epsilon, \mathbf{Z})$ and $\mathcal{M}(\mathcal{Q}, \epsilon, \mathbf{Z})$.

To reduce the above from the set-definition, consider the set

$$\mathcal{Q}(\mathbf{Z}) = \left\{ \frac{1}{n}(q(z_1), q(z_2), \ldots, q(z_n)) : q \in \mathcal{Q} \right\}.$$

In that case, $\mathfrak{R}_{\mathbf{Z}}(\mathcal{Q}) = \mathfrak{R}(\mathcal{Q}(\mathbf{Z}))$. Further, $d_{\mathbf{Z}}(q, q') = \sqrt{n} \, d(\theta_q, \theta_{q'})$ where $\theta_q = \frac{1}{n}(q(z_1), q(z_2), \ldots, q(z_n))$ and $\theta_{q'} = \frac{1}{n}(q'(z_1), q'(z_2), \ldots, q'(z_n))$ and thus lie in $\mathcal{Q}(\mathbf{Z})$.

**Complexity of function classes in the multi-task learning setting.** We now generalize the above to functions encountered in multi-task representation learning. For input space $\mathcal{Z}$, $p, q, n \in \mathbb{N}$, consider a class of vector-valued functions $\mathcal{Q} : \mathcal{Z} \to \mathbb{R}^q$, and a dataset $\mathbf{Z} = (z_j^i)_{j \in [p], i \in [n]}$, where $z_j^i \in \mathcal{Z}$. For $\mathcal{Q}^{\otimes p}$, the $p$-fold Cartesian product of $\mathcal{Q}$, we define its *data-dependent* Rademacher width, $\tilde{\mathfrak{R}}_{\mathbf{Z}}(\cdot)$, *data-dependent* Gaussian width, $\tilde{\mathfrak{S}}_{\mathbf{Z}}(\cdot)$, *data-dependent* Rademacher complexity, $\mathfrak{R}_{\mathbf{Z}}(\cdot)$, *data-dependent* Gaussian complexity, $\mathfrak{S}_{\mathbf{Z}}$, with respect to input $\mathbf{Z}$ as,

$$\tilde{\mathfrak{R}}_{\mathbf{Z}}(\mathcal{Q}^{\otimes p}) = \mathbb{E}_{\sigma_{i,j,k}} \left[ \sup_{\boldsymbol{q} \in \mathcal{Q}^{\otimes p}} \frac{1}{np} \sum_{i,jk=1}^{n,p,q} \sigma_{ijk} \left( q_j(z_j^i) \right)_k \right], \qquad \text{(Rademacher width)}$$

$$\tilde{\mathfrak{S}}_{\mathbf{Z}}(\mathcal{Q}^{\otimes p}) = \mathbb{E}_{g_{i,j,k}} \left[ \sup_{\boldsymbol{q} \in \mathcal{Q}^{\otimes p}} \frac{1}{np} \sum_{i,j,k=1}^{n,p,q} g_{i,j,k} \left( q_j(z_j^i) \right)_k \right], \qquad \text{(Gaussian width)}$$

$$\mathfrak{R}_{\mathbf{Z}}(\mathcal{Q}^{\otimes p}) = \mathbb{E}_{\sigma_{i,j,k}} \left[ \sup_{\boldsymbol{q} \in \mathcal{Q}^{\otimes p}} \left| \frac{1}{np} \sum_{i,jk=1}^{n,p,q} \sigma_{ijk} \left( q_j(z_j^i) \right)_k \right| \right], \qquad \text{(Rademacher complexity)}$$

$$\mathfrak{S}_{\mathbf{Z}}(\mathcal{Q}^{\otimes p}) = \mathbb{E}_{g_{i,j,k}} \left[ \sup_{\boldsymbol{q} \in \mathcal{Q}^{\otimes p}} \left| \frac{1}{np} \sum_{i,j,k=1}^{n,p,q} g_{i,j,k} \left( q_j(z_j^i) \right)_k \right| \right], \qquad \text{(Gaussian complexity)}$$

where $\sigma_{i,j,k}$ and $g_{i,j,k}$ are i.i.d. Rademacher and standard Normal random variables, respectively.

To reduce the above from the set-based definition, consider the set

$$\mathcal{Q}^{\otimes p}(\mathbf{Z}) : \left\{ \frac{1}{np}(((q_j(z_j^i))_k)_j)_i : \boldsymbol{q} \in \mathcal{Q}^{\otimes p} \right\}.$$

In the above, the notation $\frac{1}{np}(((q_j(z_j^i))_k)_j)_i$ denotes a vector in $\mathbb{R}^{npq}$ obtained via enumerating $i, j$ and $k$.

The corresponding metric analogously is,

$$d_{\mathbf{Z}}(\boldsymbol{q}, \boldsymbol{q}') = \|\boldsymbol{q} - \boldsymbol{q}'\|_{L^2(\mathbf{Z})} = \sqrt{\frac{1}{np} \sum_{i,j,k} \left( (q_j(z_j^i))_k - (q_j(z_j^i))_k \right)^2} = \sqrt{np} d_{\mathbf{Z}}(\theta_{\boldsymbol{q}}, \theta_{\boldsymbol{q}'}),$$

where $\theta_{\boldsymbol{q}} = \frac{1}{np}(((q_j(z_j^i))_k)_j)_i$ and similarly $\theta_{\boldsymbol{q}'}$, which lie in $\mathcal{Q}^{\otimes p}(\mathbf{Z})$.

The above can be used to define covering and packing numbers of $\mathcal{Q}^{\otimes p}$, denoted as $\mathcal{N}(\mathcal{Q}^{\otimes p}, \epsilon, \mathbf{Z})$ and $\mathcal{M}(\mathcal{Q}^{\otimes p}, \epsilon, \mathbf{Z})$ respectively.

The concepts of Dudley's entropy integral formula and Sudakov minoration thus generalize accordingly. A result of the following form, for real-valued functions, appears in Srebro et al. [2010].

**Lemma 7** (Refined Dudley's entropy integral). *Consider a class of vector-valued functions $\mathcal{Q}$ and input set $\mathbf{Z} = (z_j^i)_{j \in [p], i \in [n]}$. Define $B := \sup_{\boldsymbol{q} \in \mathcal{Q}^{\otimes p}} \|\boldsymbol{q}\|_{L^2(\mathbf{Z})} = \sqrt{np} \sup_{\boldsymbol{q} \in \mathcal{Q}^{\otimes p}} \left\| \frac{1}{np}(((q_j(z_j^i))_k)_j)_i \right\|$. Then,*

$$\mathfrak{R}_{\mathbf{Z}}(\mathcal{F}) \leq \inf_{0 \leq \alpha \leq B/\sqrt{np}} \left\{ 4\alpha + 10 \int_{\alpha}^{B/\sqrt{np}} \sqrt{\log \mathcal{N}\left(\mathcal{Q}^{\otimes p}, \epsilon, \mathbf{Z}\right)} d\epsilon \right\}$$

$$= \inf_{0 \leq \alpha \leq B} \left\{ 4\alpha + 10 \int_{\alpha}^{B} \sqrt{\frac{\log \mathcal{N}\left(\mathcal{Q}^{\otimes p}, \epsilon, \mathbf{Z}\right)}{np}} d\epsilon \right\}.$$

The equality above follows from the change of variables.

**Worst-case complexities of functions.** Often, we take the sup over datasets of a specified size, in the above definitions of Rademacher/ Gaussian width/complexity, yielding their **worst-case** counterparts, denoted as,

$$\tilde{\mathfrak{G}}_{np}(\mathcal{Q}^{\otimes p}) = \sup_{\mathbf{Z} \in \mathcal{Z}^{np}} \tilde{\mathfrak{G}}_{\mathbf{Z}}(\mathcal{Q}^{\otimes p}), \qquad \mathfrak{G}_{np}(\mathcal{Q}^{\otimes p}) = \sup_{\mathbf{Z} \in \mathcal{Z}^{np}} \mathfrak{G}_{\mathbf{Z}}(\mathcal{Q}^{\otimes p}),$$

$$\tilde{\mathfrak{R}}_{np}(\mathcal{Q}^{\otimes p}) = \sup_{\mathbf{Z} \in \mathcal{Z}^{np}} \tilde{\mathfrak{R}}_{\mathbf{Z}}(\mathcal{Q}^{\otimes p}), \qquad \mathfrak{R}_{np}(\mathcal{Q}^{\otimes p}) = \sup_{\mathbf{Z} \in \mathcal{Z}^{np}} \mathfrak{R}_{\mathbf{Z}}(\mathcal{Q}^{\otimes p}).$$

The corresponding metric similarly is

$$d_{np}(\boldsymbol{q}, \boldsymbol{q}') = \sup_{\mathbf{Z} \in \mathcal{Z}^{np}} d_{\mathbf{Z}}(\boldsymbol{q}, \boldsymbol{q}'),$$

which can then be used to define covering and packing numbers of $\mathcal{Q}^{\otimes p}$, denoted as $\mathcal{N}(\mathcal{Q}^{\otimes}, \epsilon, np)$ and $\mathcal{M}(\mathcal{Q}^{\otimes}, \epsilon, np)$, respectively.

As before, Dudley's entropy integral formula and Sudakov minoration generalize analogously.

### A.2 Sub-root functions and local Rademacher Complexity

We present a key property about sub-root functions from [Bartlett et al., 2005] below.

**Lemma 8** (Lemma 3.2. [Bartlett et al., 2005])**.** *If $\psi : [0, \infty) \to [0, \infty)$ is a nontrivial sub-root function, then it is continuous on $[0, \infty)$ and the equation $\psi(r) = r$ has a unique positive solution. Moreover, if we denote the solution by $r^*$, then for all $r > 0, r \geq \psi(r)$ if and only if $r^* \leq r$.*

## B  Task diversity digression

In this section we make a series of observations regarding the task-diversity definition introduced by Tripuraneni et al. [2020]. Within this work, they introduce two definitions that are used to relate average source tasks performance to target task performance: task-averaged representation difference and worst-case representation difference.

**Definition 6** (The task-averaged representation difference w.r.t $h \in \mathcal{H}$)**.**

$$\bar{d}_{\mathcal{F}, \boldsymbol{f}}(h'; h) = \frac{1}{t} \sum_{j=1}^{t} \inf_{f' \in \mathcal{F}} \mathbb{E}_{x_j, y_j \sim P_j} \left\{ \ell\left(f' \circ h'(x_j), y_j\right) - \ell\left(f_j \circ h(x_j), y_j\right) \right\}$$

Importantly, like the work which introduces these concepts, it is possible to statistically control $\bar{d}_{\mathcal{F}, \boldsymbol{f}}\left(\hat{h}; h^*\right)$, where $\hat{h}$ results from the first step of the two–tage ERM procedure, see the variational definition Eqn. (2).

The other quantity they introduce is the worst-case representation difference.

**Definition 7** (The worst-case representation difference w.r.t $h, h' \in \mathcal{H}$)**.**

$$d_{\mathcal{F}, \mathcal{F}_0}(h'; h) = \sup_{f_0 \in \mathcal{F}} \inf_{f' \in \mathcal{F}} \mathbb{E}_{x, y \sim P_0} \left\{ \ell(f' \circ h'(x), y) - \ell(f_0 \circ h(x), y) \right\},$$

Finally, they introduce the following assumption on the two quantities defined above. It has a multiplicative parameter $\nu$ and additive parameter $\varepsilon$.

**Definition 8** (($\nu, \varepsilon$)-task diversity w.r.t $h \in \mathcal{H}$ [Tripuraneni et al., 2020])**.** *For a function class $\mathcal{F}$, we say $\boldsymbol{f} \in \mathcal{F}^{\otimes t}$ is $(\nu, \epsilon)$-diverse over $\mathcal{F}_0$ for a representation h, if uniformly for all $h' \in \mathcal{H}$,*

$$d_{\mathcal{F}, \mathcal{F}_0}(h'; h) \leq \bar{d}_{\mathcal{F}, \mathbf{f}}(h'; h) / \nu + \epsilon$$

We will make a series of observations about these quantities which provide substantial simplification in our setting.

First, observe in the setting of product space,

$$\bar{d}_{\mathcal{F}, \boldsymbol{f}}(h'; h) = \inf_{\boldsymbol{f}' \in \mathcal{F}^{\otimes t}} \left\{ R_{\text{source}}(\boldsymbol{f}', h') - R_{\text{source}}(\boldsymbol{f}, h) \right\}$$

and

$$d_{\mathcal{F},\mathcal{F}_0}(h';h) = \sup_{f_0\in\mathcal{F}_0} \inf_{f'\in\mathcal{F}} \left\{ R_{\text{target}}(f',h') - R_{\text{target}}(f_0,h) \right\}$$

Note, here we are again leveraging product space structure to simplify prior work. Recall in Section 5 we observed that this structure allows us to recover existing single function theorems exactly, unlike Yousefi et al. [2018]. Specifically in the proof of Theorem 9 we "commute" suprema and summations.

Here we are "commuting" infima and summations to simplify the framework specified in Tripuraneni et al. [2020] within our setting. Note they work with product spaces but do not make the simplifying observation above. Nonetheless, the definitions they provide are suitable for the more general function classes considered in Yousefi et al. [2018], which are not necessarily product spaces.

Now consider the setting detailed in the preliminaries applied to our approximation $\hat{h}, h^* \in \mathcal{H}$ and $\boldsymbol{f}^* \in \mathcal{F}^{\otimes t}$ with the following ratio

$$
\begin{aligned}
\frac{d_{\mathcal{F},\mathcal{F}_0}(\hat{h};h^*)}{\bar{d}_{\mathcal{F},\boldsymbol{f}^*}(\hat{h};h^*)} &= \frac{\sup_{f_0\in\mathcal{F}_0}\inf_{f'\in\mathcal{F}}\left\{R_{\text{target}}(f',\hat{h}) - R_{\text{target}}(f_0,h^*)\right\}}{\inf_{\boldsymbol{f}'\in\mathcal{F}^{\otimes t}}\left\{R_{\text{source}}(\boldsymbol{f}',\hat{h}) - R_{\text{source}}(\boldsymbol{f}^*,h^*)\right\}} \\
&= \frac{\inf_{f'\in\mathcal{F}}R_{\text{target}}(f',\hat{h}) - \inf_{f_0\in\mathcal{F}_0}R_{\text{target}}(f_0,h^*)}{\inf_{\boldsymbol{f}'\in\mathcal{F}^{\otimes t}}\left\{R_{\text{source}}(\boldsymbol{f}',\hat{h}) - R_{\text{source}}(\boldsymbol{f}^*,h^*)\right\}} \\
&= \frac{\inf_{f'\in\mathcal{F}}R_{\text{target}}(f',\hat{h}) - R_{\text{target}}(f_0^*,h^*)}{\inf_{\boldsymbol{f}'\in\mathcal{F}^{\otimes t}}\left\{R_{\text{source}}(\boldsymbol{f}',\hat{h}) - R_{\text{source}}(\boldsymbol{f}^*,h^*)\right\}}.
\end{aligned}
\tag{9}
$$

Note every $\sup$ and $\inf$ will realize a function that minimizes their respective risk.

This ratio is a measure of how similar the performance we can expect when we use our representation learned from the source tasks to and apply it the target task. That is if we seek to control this quantity we do not care per se if the risks are small nor do we care that we are getting good performance in comparison to the ground truth, but how comparable is our performance between source and target.

We seek to control this ratio with some $\nu$

$$\frac{\inf_{f'\in\mathcal{F}}R_{\text{target}}(f',\hat{h}) - R_{\text{target}}(f_0^*,h^*)}{\inf_{\boldsymbol{f}'\in\mathcal{F}^{\otimes t}}\left\{R_{\text{source}}(\boldsymbol{f}',\hat{h}) - R_{\text{source}}(\boldsymbol{f}^*,h^*)\right\}} \le \frac{1}{\nu}.$$

If the risk on our source tasks and target task is similar w.r.t. to the ground truth for all tasks then $\nu \approx 1$. That is low (high) average risk on the source tasks leads to low (high) risk on the target task.

This is even more apparent in the realizable setting.

$$\frac{\inf_{f'\in\mathcal{F}}R_{\text{target}}(f',\hat{h})}{\inf_{\boldsymbol{f}'\in\mathcal{F}^{\otimes t}}R_{\text{source}}(\boldsymbol{f}',\hat{h})} \le \frac{1}{\nu}.$$

First, imagine a setting in which there are several tasks and on all but a few source tasks, we perform well.

$$\frac{\inf_{f'\in\mathcal{F}}R_{\text{target}}(f',\hat{h})}{\inf_{\boldsymbol{f}'\in\mathcal{F}^{\otimes t}}R_{\text{source}}(\boldsymbol{f}',\hat{h})} \approx \frac{\text{relatively small}}{\text{relatively big}}.$$

Then $\frac{1}{\nu}$ will be small, which is okay if we do not care about performance across all $t$ tasks, as long as the target task performance is good.

Alternatively, imagine that the task does relatively poorly on the target task but is still small. This could lead to a very large $\frac{1}{\nu}$ which is not desirable.

$$\frac{\inf_{f'\in\mathcal{F}}R_{\text{target}}(f',\hat{h})}{\inf_{\boldsymbol{f}'\in\mathcal{F}^{\otimes t}}R_{\text{source}}(\boldsymbol{f}',\hat{h})} \approx \frac{\text{relatively big}}{\text{relatively small}}.$$

Therefore we allow for an additive $\varepsilon$ to capture this quantity when we get risk on the source task is *very small*, but we are okay with *relatively* poor performance on the target task.

That is we care about the following inequality

$$\inf_{f' \in \mathcal{F}} R_{\text{target}}(f', \hat{h}) - R_{\text{target}}(f_0^*, h^*) \leq \frac{1}{\nu} \inf_{f' \in \mathcal{F}^{\otimes t}} \left\{ R_{\text{source}}(f', \hat{h}) - R_{\text{source}}(f^*, h^*) \right\} + \varepsilon, \quad (10)$$

where $\varepsilon$ encodes how much tolerance we have between the performance on the source tasks and the target task. Hence, it is Inequality (10) which we care to control over all functions in $\mathcal{H}$. Therefore, we will assume uniformly for all $h' \in \mathcal{H}$ that:

$$\inf_{f' \in \mathcal{F}} R_{\text{target}}(f', \hat{h}) - R_{\text{target}}(f_0^*, h^*) \leq \frac{1}{\nu} \inf_{f' \in \mathcal{F}^{\otimes t}} \left\{ R_{\text{source}}(f', \hat{h}) - R_{\text{source}}(f^*, h^*) \right\} + \varepsilon.$$

Although we observe this simplification, to be succinct we will still use that

$$d_{\mathcal{F}, \mathcal{F}_0}(\hat{h}; h^*) = \inf_{f' \in \mathcal{F}} R_{\text{target}}(f', \hat{h}) - R_{\text{target}}(f_0^*, h^*) \quad (11)$$

and

$$\bar{d}_{\mathcal{F}, f^*}(\hat{h}; h^*) = \inf_{f' \in \mathcal{F}^{\otimes t}} \left\{ R_{\text{source}}(f', \hat{h}) - R_{\text{source}}(f^*, h^*) \right\} \quad (12)$$

and use this notation in the proofs of Appendix D.3.

# C   Technical Miscellanea

In this section, we present technical miscellanea that we shall later refer to in obtaining our proofs of the main results in Appendix D.

## C.1   Martingale Concentration Inequality

For the sake of notational succinctness, we forgo the compositional model by overloading the notation to just consider a generic function class $\mathcal{F} : \mathcal{X} \to \mathbb{R}$.

**Lemma 9** (Theorem 2.1 in Bousquet [2002]). *Let $W_1, \ldots, W_n$ be independent random variables in a Polish space $\mathcal{W}$.*

*Let*

$$\mathcal{A} = \sigma(W_1, \ldots, W_n)$$
$$\forall k \in [n] \quad \mathcal{A}_k = \sigma(W_1, , \ldots, W_{k-1}, W_{k+1}, \ldots, W_n)$$

*be $\sigma$-algebras generated by these random variables.*

*We denote by $\mathbb{E}_k[\cdot]$ the expectation taken conditionally on $\mathcal{A}_k$. Let $\varphi(x) = (1 + x) \log(1 + x) - x$ and $\psi(x) = e^{-x} - 1 + x$.*

*Let $(Z, Z_1', \ldots, Z_n')$ be a sequence of $\mathcal{A}$-measurable random variables and let $(Z_k)_{k \in [n]}$ be a sequence of random variables that are $\mathcal{A}_k$ measurable, respectively. Assume that there exists $u > 0$ such that for all $k \in [n]$ the following inequalities are satisfied*

$$Z_k' \leq Z - Z_k \leq 1 \quad a.s., \quad \mathbb{E}_k[Z_k'] \geq 0 \quad and \quad Z_k' \leq u \quad a.s.$$

*Let $\sigma \in \mathbb{R}$ be a real number satisfying $\sigma^2 \geq \frac{1}{n} \sum_{k=1}^n \mathbb{E}_k\left[(Z_k')^2\right]$ almost surely and let $v = (1 + u)\mathbb{E}[Z] + n\sigma^2$. If the following condition holds*

$$\sum_{k=1}^n Z - Z_k \leq Z \quad a.s.,$$

*we obtain, for all $\lambda \geq 0$,*

$$\log \mathbb{E}\left[e^{\lambda(Z - \mathbb{E}[Z])}\right] \leq \psi(-\lambda)v,$$

*which gives the following bounds for all $\delta > 0$,*

$$\mathbb{P}\{Z \geq \mathbb{E}[Z] + \delta\} \leq \exp\left(-v\varphi\left(\frac{\delta}{v}\right)\right) \quad and \quad \mathbb{P}\{Z \geq \mathbb{E}[Z] + \sqrt{2v\delta} + \frac{\delta}{3}\} \leq e^{-\delta}.$$

The following corollary is immediate as independence is sufficient Bousquet [2003] and we can express the summation over $nt$ random variables as "double summation" where we first sum over $q \in [n]$ and then over $j \in [t]$.

**Corollary 1** (Lemma 9 in block structure). *Assume that $W_j^1, \ldots, W_j^n$ are $n$ independent draws from $P_j$ for $j \in [t]$ with random variables in some polish space, where all $nt$ samples are independent.*

*Let $\mathcal{A}$ be the sigma algebra generated by $(W_j^i)_{j \in [t], i \in [n]}$ and $\mathcal{A}_k^q$ be the sigma algebras generated by $(W_j^i)_{\substack{j \in [t], i \in [n] \\ (j,i) \neq (k,q)}}$. We denote by $\mathbb{E}_k^q[\cdot]$ the expectation taken conditionally on $\mathcal{A}_k^q$.*

*Let $Z$ be a $\mathcal{A}$-measurable r.v., $(\tilde{Z}_k^q)_{k \in [t], q \in [n]}$ be a sequence of $\mathcal{A}$-measurable random variables, and $(Z_k^q)_{k \in [t], q \in [n]}$ be a sequence of random variables that are $\mathcal{A}_k^q$ measurable, respectively. Assume that there exists $u > 0$ such that for all $k \in [t]$ and $q \in [n]$ the following inequalities are satisfied*

$$\tilde{Z}_k^q \leq Z - Z_k^q \leq 1 \quad a.s., \tag{13}$$

$$\mathbb{E}_k^q\left[\tilde{Z}_k^q\right] \geq 0, \tag{14}$$

$$and \qquad \tilde{Z}_k^q \leq u \quad a.s. \tag{15}$$

*Let $\sigma \in \mathbb{R}$ be a real number satisfying*

$$\sigma^2 \geq \frac{1}{nt} \sum_{k=1}^{t} \sum_{q=1}^{n} \mathbb{E}_k^q\left[\left(\tilde{Z}_k^q\right)^2\right] \tag{16}$$

*almost surely and let $v = (1+u)\mathbb{E}[Z] + n\sigma^2$. If the following condition holds*

$$\sum_{k=1}^{t} \sum_{q=1}^{n} Z - Z_k^q \leq Z \quad a.s., \tag{17}$$

*we obtain, for all $\lambda \geq 0$,*

$$\log \mathbb{E}\left[e^{\lambda(Z - \mathbb{E}[Z])}\right] \leq \psi(-\lambda) v,$$

*which gives the following bounds for all $\delta > 0$,*

$$\mathbb{P}\{Z \geq \mathbb{E}[Z] + \delta\} \leq \exp\left(-v\varphi\left(\frac{\delta}{v}\right)\right) \quad and \quad \mathbb{P}\{Z \geq \mathbb{E}[Z] + \sqrt{2v\delta} + \frac{\delta}{3}\} \leq e^{-\delta}.$$

### C.2 Empirical Process Lemmas

The following result is standard symmetrization.

**Lemma 10.** *Let $\mathcal{F}$ be a class of functions from $\mathcal{X}$ to $\mathbb{R}$ and let $\mathbf{X}$ denote the set of inputs $\left\{X_j^i\right\}_{i,j=1}^{n,t}$. Assume that all functions $f$ in $\mathcal{F}$ are $P_j$-measurable for all $j \in [t]$. Then,*

$$\mathbb{E}\left[\sup_{\boldsymbol{f} \in \mathcal{F}^{\otimes t}} \frac{1}{nt} \sum_{j=1}^{t} \sum_{i=1}^{n} f_j(X_j^i) - \mathbb{E}\frac{1}{nt} \sum_{j=1}^{t} \sum_{i=1}^{n} f_j(X_j^i)\right] \leq 2\tilde{\mathfrak{R}}_{\mathbf{X}}(\mathcal{F}^{\otimes t})$$

*where $\{\sigma_j^i\}$ is a sequence of $nt$ independent Rademacher variables.*

The result follows by standard symmetrization proof extended to the multi-function setting.

*Proof.* Let $\tilde{X}_j^1, \ldots, \tilde{X}_j^n$ be independent copies of $X_j^1, \ldots, X_j^n$ on probability space $P_j$ for all $j \in [t]$. Let $\mathbb{E}_{\tilde{\mathbf{X}}}$ be the expectation with respect to $\tilde{X}_j^1, \ldots, \tilde{X}_j^n$ for all $j \in [t]$ and $\mathbb{E}_{\mathbf{X}}$ be the expectation with respect to $X_j^1, \ldots, X_j^n$ for all $j \in [t]$. Let $\sigma_j^i$ be i.i.d. Rademacher random variables for all $j \in [t]$

and $i \in [n]$. Fixing $X_j^1, \ldots, X_j^n$ for all $j \in [t]$, and using the copied variables we have that

$$\mathbb{E} \sup_{\boldsymbol{f} \in \mathcal{F}^{\otimes t}} \left( \frac{1}{nt} \sum_{j=1}^{t} \sum_{i=1}^{n} f_j(X_j^i) - \mathbb{E} \frac{1}{nt} \sum_{j=1}^{t} \sum_{i=1}^{n} f_j(X_j^i) \right)$$

$$= \mathbb{E}_{\mathbf{X}} \sup_{\boldsymbol{f} \in \mathcal{F}^{\otimes t}} \left( \frac{1}{nt} \sum_{j=1}^{t} \sum_{i=1}^{n} f_j(X_j^i) - \mathbb{E}_{\tilde{X}} \frac{1}{nt} \sum_{j=1}^{t} \sum_{i=1}^{n} f_j(\tilde{X}_j^i) \right).$$

By Jensen's inequality, we have

$$\mathbb{E} \sup_{\boldsymbol{f} \in \mathcal{F}^{\otimes t}} \left( \frac{1}{nt} \sum_{j=1}^{t} \sum_{i=1}^{n} f_j(X_j^i) - \mathbb{E} \frac{1}{nt} \sum_{j=1}^{t} \sum_{i=1}^{n} f_j(X_j^i) \right)$$

$$\leq \mathbb{E}_{\mathbf{X}, \tilde{\mathbf{X}}} \sup_{\boldsymbol{f} \in \mathcal{F}^{\otimes t}} \left( \frac{1}{nt} \sum_{j=1}^{t} \sum_{i=1}^{n} f_j(X_j^i) - \frac{1}{nt} \sum_{j=1}^{t} \sum_{i=1}^{n} f_j(\tilde{X}_j^i) \right)$$

$$= \frac{1}{nt} \mathbb{E}_{\mathbf{X}, \tilde{\mathbf{X}}} \sup_{\boldsymbol{f} \in \mathcal{F}^{\otimes t}} \left( \sum_{j=1}^{t} \sum_{i=1}^{n} \left( f_j(\tilde{X}_j^i) - f_j(X_j^i) \right) \right)$$

If $\tilde{X}_j^i$ and $X_j^i$ have the same distribution, we could exchange $f_j(\tilde{X}_j^i) - f_j(X_j^i)$ for $f_j(X_j^i) - f_j(\tilde{X}_j^i)$ and the equation would be equivalent. We can do this with Rademacher random variables, by randomly flipping the sign with probability 1/2. This yields the following

$$\mathbb{E} \sup_{\boldsymbol{f} \in \mathcal{F}^{\otimes t}} \left( \frac{1}{nt} \sum_{j=1}^{t} \sum_{i=1}^{n} f_j(X_j^i) - \mathbb{E} \frac{1}{nt} \sum_{j=1}^{t} \sum_{i=1}^{n} f_j(X_j^i) \right)$$

$$= \frac{1}{nt} \mathbb{E}_{\sigma, \mathbf{X}, \tilde{\mathbf{X}}} \sup_{\boldsymbol{f} \in \mathcal{F}^{\otimes t}} \left( \sum_{j=1}^{t} \sum_{i=1}^{n} \left( \sigma_j^i f_j(\tilde{X}_j^i) - \sigma_j^i f_j(X_j^i) \right) \right)$$

$$\leq \frac{1}{nt} \mathbb{E}_{\sigma, \tilde{\mathbf{X}}} \sup_{\boldsymbol{f} \in \mathcal{F}^{\otimes t}} \left( \sum_{j=1}^{t} \sum_{i=1}^{n} \left( \sigma_j^i f_j(\tilde{X}_j^i) \right) \right)$$

$$+ \frac{1}{nt} \mathbb{E}_{\sigma, \mathbf{X}} \sup_{\boldsymbol{f} \in \mathcal{F}^{\otimes t}} \left( \sum_{j=1}^{t} \sum_{i=1}^{n} \left( \sigma_j^i f_j(X_j^i) \right) \right)$$

$$= 2 \mathbb{E} \tilde{\mathfrak{R}}_{\mathbf{X}} (\mathcal{F}^{\otimes t}).$$

where the last inequality follows from triangle inequality. $\qquad \square$

The next result, a Gaussian chain rule given in Tripuraneni et al. [2020], is crucial to decoupling the complexity of the representation function class and the hypothesis class.

**Theorem 8** (Theorem 7 from Tripuraneni et al. [2020])**.** *Let the function class $\mathcal{F}$ consist of functions that satisfy Assumption 1. Then the (empirical) Gaussian width of the function class $\mathcal{F}^{\otimes t}(\mathcal{H})$ satisfies,*

$$\tilde{\mathfrak{G}}_{\boldsymbol{Z}} \left( \mathcal{F}^{\otimes t}(\mathcal{H}) \right) \leq \inf_{D \geq \alpha > 0} \left\{ 4\alpha + 64 G \left( \mathcal{F}^{\otimes t}(\mathcal{H}) \right) \log \left( \frac{D}{\alpha} \right) \right\}.$$

*where $G(\mathcal{F}^{\otimes t}(\mathcal{H})) = L \tilde{\mathfrak{G}}_{nt}(\mathcal{H}) + \tilde{\mathfrak{G}}_n(\mathcal{F})$. Further, if $G \left( \mathcal{F}^{\otimes t}(\mathcal{H}) \right) \leq D$ then by computing the exact infima of the expression,*

$$\tilde{\mathfrak{G}}_{\mathbf{X}} \left( \mathcal{F}^{\otimes t}(\mathcal{H}) \right) \leq 64 G \left( \mathcal{F}^{\otimes t}(\mathcal{H}) \right) \log \left( \frac{eD}{G \left( \mathcal{F}^{\otimes t}(\mathcal{H}) \right)} \right). \tag{18}$$

**Lemma 11** (Lemma A.4 in Bartlett et al. [2005]). *Let $\mathcal{F}$ be a class of real-valued functions with range in $[a,b]$. Given a set $\mathbf{Z}$ of $nt$ i.i.d. points, with probability at least $1 - e^{-\delta}$,*

$$\mathbb{E}\tilde{\mathfrak{R}}_{\mathbf{Z}}(\mathcal{F}) \leq \inf_{\alpha \in (0,1)} \left( \frac{\tilde{\mathfrak{R}}_{\mathbf{Z}}(\mathcal{F})}{1-\alpha} + \frac{(b-a)\delta}{4n\alpha(1-\alpha)} \right).$$

## C.3  Analytic Inequalities

**Lemma 12.** *If $x - \sqrt{x}A - B = 0$ for $A, B > 0$ then $x \leq A^2 + 2B$.*

*Proof.* As the discriminant of the quadratic in $\sqrt{x}$ is positive we are guaranteed two real roots $x_1, x_2$. Taking the larger of the two solutions we have $\sqrt{x_2} = \frac{A}{2} + \frac{\sqrt{A^2+4B}}{2}$. So $x_2 = \frac{A^2}{2} + B + A\frac{\sqrt{A^2+4B}}{2}$. By AM-GM, $x_2 \leq \frac{A^2}{2} + B + \frac{A^2}{4} + \frac{A^2+4B}{4} = A^2 + 2B$ □

**Lemma 13** (Lemma B.1. from Srebro et al. [2010]). *For any $H$-smooth non-negative function $\ell : \mathbb{R} \mapsto \mathbb{R}$ and any $t, r \in \mathbb{R}$ we have that*

$$(\ell(t) - \ell(r))^2 \leq 6H(\ell(t) + \ell(r))(t-r)^2.$$

**Lemma 14.** *For $k \in [t], q \in [n]$,*

$$\left( \left| \sum_{j,i=1}^{t,n} a_{ji} \right| - \left| \sum_{\substack{j,i=1 \\ (j,i) \neq (k,q)}}^{t,n} a_{ji} \right| \right)^2 \leq a_{kq}^2.$$

*Proof.* First note that for any $k \in [n]$,

$$\left( \left| \sum_{i=1}^{n} a_i \right| - \left| \sum_{\substack{i=1 \\ i \neq k}}^{n} a_i \right| \right)^2 \leq a_k^2.$$

This follows from applying reverse triangle inequality

$$\left| \left| \sum_{i=1}^{n} a_i \right| - \left| \sum_{\substack{i=1 \\ i \neq k}}^{n} a_i \right| \right| \leq \left| \sum_{i=1}^{n} a_i - \sum_{\substack{i=1 \\ i \neq k}}^{n} a_i \right| = |a_k|.$$

Finally, we square both sides and rearrange the single summation into a double summation, which completes the proof. □

**Lemma 15.** *Consider the function $f(x) = x + a \ln(b/x)$ for $a, b \geq 0$.*

1. *The minimizer of the function over $0 < x \leq b$ is $x^* = \min(a, b)$.*

2. *The corresponding minimum value is $a \ln \left( \frac{eb}{\min(a,b)} \right)$*

3. *For any $c$ such that $a \leq c \leq b$, $a \ln(eb/a) \leq c \ln(eb/c)$.*

*Proof.* The first and second claims follow from the fact that the function $f$ decreases from $x = 0$ to $a$, and then increases. The third claim follows since the function $x \mapsto x \ln(eb/x)$ increases till $x = b$. □

**Lemma 16.** *Let $Z$ be a random variable taking values in $\mathbb{R}$ and let $A \in \mathbb{R}$ be such that $\mathbb{P}[Z \geq A] > 0$, then $\sup |Z| \geq \operatorname{ess\,sup} |Z| = \|Z\|_\infty \geq A$.*

*Proof.* The first inequality is standard. The second simply follows from the definition of $\operatorname{ess\,sup}$:

$$\operatorname{ess\,sup} |Z| = \inf \{a \in \mathbb{R} : \mathbb{P}[Z \geq a] = 0\}.$$

□

# D   Proofs of main results

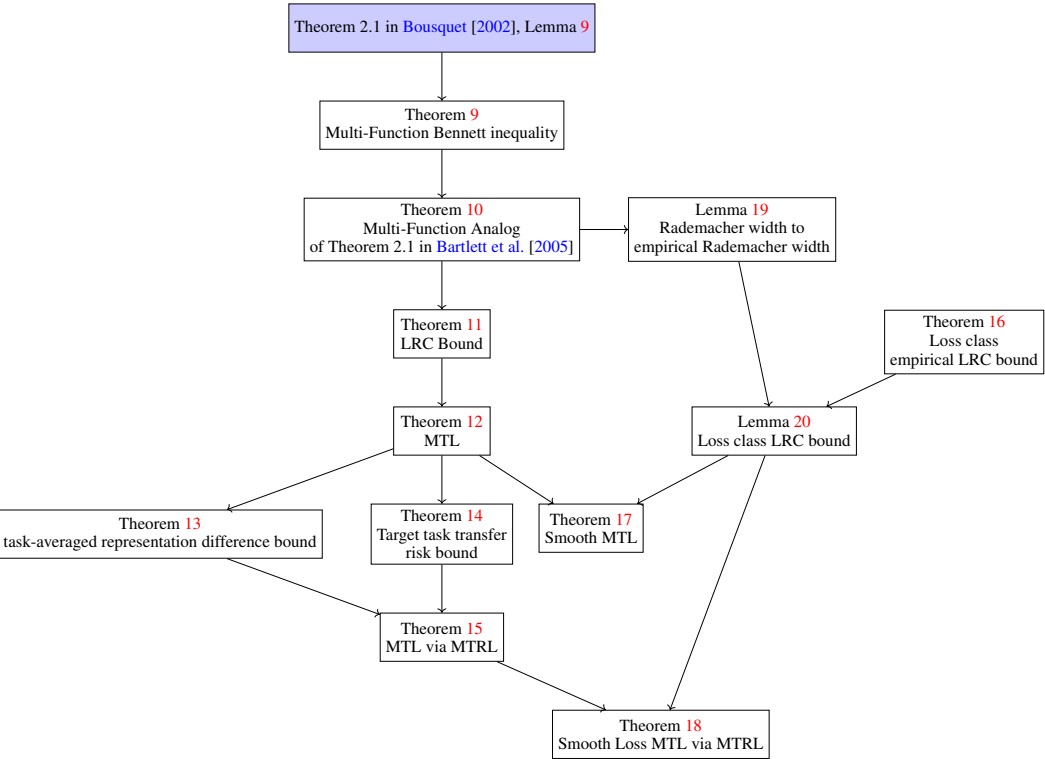

Figure 1: A graph demonstrating the dependency structure of our results. The top blue node is the concentration inequality given in Bousquet [2002]. We abbreviate "local Rademacher complexity" with "LRC".

In this section, we restate the main theorems in our main paper (with precise constants) and give a detailed proof of our results. A graph in Appendix D shows the dependency between our theorems.

Note the relationship between the theorems in the main paper and the theorems in the appendix are as follows:

- Theorem 1 becomes Theorem 15
- Theorem 2 becomes Theorem 18
- Proposition 1 becomes Theorem 16
- Theorem 3 becomes Theorem 12
- Theorem 4 becomes Theorem 17
- Theorem 5 becomes Theorem 19
- Theorem 6 becomes Theorem 11
- Theorem 7 becomes Theorem 9

In Appendix D.1, we present Theorems 9 to 11 which are generalizations of prior work and also recover the single function setting. In particular, we give a concentration inequality and two risk bounds. In Section D.2, we present Theorem 12 our result for multitask learning with bounded non-negative loss classes. Next, Appendix D.3 contains Theorems 13 to 15 the MTL via MTRL learning setting.

The remaining theorems add the additional assumption of smoothness. We start, in section Appendix D.4, by giving Theorem 16, our bound on the empirical local Rademacher complexity of smooth bounded non-negative loss classes. However, in order to use the above result with our prior theorems we need to bound the local Rademacher complexity constrained on the risk to one

constrained by the empirical risk. So, in Appendix D.5, we introduce Lemmas 19 and 20 to make this conversion. In Appendix D.6 we conclude by invoking our bound on the local Rademacher complexity of smooth bounded non-negative loss classes within our MTL and MTRL theorems with Theorems 17 and 18.

## D.1  Concentration Inequalities and Risk Bounds

In this section we forgo the compositional model and will consider a generic function class $\mathcal{F}$.

**Theorem 9** (Concentration with $t$ functions). *Let $\mathcal{F}$ be a class of functions from $\mathcal{X}$ to $\mathbb{R}$ and assume that all functions $f$ in $\mathcal{F}$ are $P_j$-measurable for all $j \in [t]$, square-integrable, and satisfy $\mathbb{E}f = 0$. Let $\sup_{f \in \mathcal{F}} \|f\|_\infty \leq 1$ and*

$$Z = \sup_{\boldsymbol{f} \in \mathcal{F}^{\otimes t}} \left| \sum_{j=1}^{t} \sum_{i=1}^{n} f_j(X_j^i) \right| \tag{19}$$

*or let $\sup_{f \in \mathcal{F}} \operatorname{ess\,sup} f \leq 1$ and*

$$Z = \sup_{\boldsymbol{f} \in \mathcal{F}^{\otimes t}} \sum_{j=1}^{t} \sum_{i=1}^{n} f_j(X_j^i). \tag{20}$$

*Let $\sigma > 0$ such that $\sigma^2 \geq \sup_{\boldsymbol{f} \in \mathcal{F}^{\otimes t}} \frac{1}{t} \sum_{j=1}^{t} \operatorname{Var}\left[ f_j(X_j^1) \right]$ almost surely, then for all $\delta \geq 0$, we have*

$$\mathbb{P}\{ Z \leq \mathbb{E}Z + \sqrt{2\delta(tn\sigma^2 + 2\mathbb{E}Z)} + \frac{\delta}{3} ) \} \leq e^{-\delta}.$$

*Proof of Theorem 9.* We prove the result for $Z$ as defined in Equation (19); the proof for the definition of $X$ in Equation (20) is similar. Let

$$Z = \sup_{\boldsymbol{f} \in \mathcal{F}^{\otimes t}} \left| \sum_{j,i=1}^{t,n} f_j(X_j^i) \right|.$$

and let $f_1^{(0)}, \ldots, f_t^{(0)}$ be the functions that achieve the supremum w.r.t. $Z$.

$$Z = \left| \sum_{j,i=1}^{t,n} f_j^{(0)}(X_j^i) \right|.$$

Next, for each $k \in [t]$ and $q \in [n]$ let

$$Z_k^q = \sup_{\boldsymbol{f} \in \mathcal{F}^{\otimes t}} \left| \sum_{\substack{j,i=1 \\ (j,i) \neq (k,q)}}^{t,n} f_j(X_j^i) \right| = \sup_{\boldsymbol{f} \in \mathcal{F}^{\otimes t}} \left| -f_k(X_k^q) + \sum_{j,i=1}^{t,n} f_j(X_j^i) \right|.$$

The random variable $Z_k^q$ is essentially removing a $f_k(X_k^q)$ term from $Z$. We define $f_1^{(k,q)}, \ldots, f_t^{(k,q)}$ as the functions which achieve the supremum w.r.t $Z_k^q$. That is,

$$Z_k^q = \left| \sum_{\substack{j,i=1 \\ (j,i) \neq (k,q)}}^{t,n} f_j^{(k,q)}(X_j^i) \right|.$$

Finally define,

$$\tilde{Z}_k^q = \left| \sum_{j,i=1}^{t,n} f_j^{(k,q)}(X_j^i) \right| - Z_k^q.$$

We now prove that the conditions in Corollary 1 are satisfied.

$$\tilde{Z}_k^q = \left| \sum_{j,i=1}^{t,n} f_j^{(k,q)}(X_j^i) \right| - Z_k^q$$

$$\leq \left| \sum_{j,i=1}^{t,n} f_j^{(0)}(X_j^i) \right| - \left| \sum_{\substack{j,i=1 \\ (j,i) \neq (k,q)}}^{t,n} f_j^{(k,q)}(X_j^i) \right| \tag{21}$$

$$\leq \left| \sum_{j,i=1}^{t,n} f_j^{(0)}(X_j^i) \right| - \left| \sum_{\substack{j,i=1 \\ (j,i) \neq (k,q)}}^{t,n} f_j^{(0)}(X_j^i) \right| \tag{22}$$

$$\leq \left| \sum_{\substack{j,i=1 \\ (j,i) \neq (k,q)}}^{t,n} f_j^{(0)}(X_j^i) \right| + \left| f_k^{(0)}(X_k^q) \right| - \left| \sum_{\substack{j,i=1 \\ (j,i) \neq (k,q)}}^{t,n} f_j^{(0)}(X_j^i) \right| \quad \text{(by triangle inequality)}$$

$$= \left| f_k^{(0)}(X_k^q) \right|$$

$$\leq 1 \quad \text{a.s} \quad \text{(by assumption)}$$

where Inequality 21 follows from $f_1^{(0)}, \ldots, f_t^{(0)}$ being the functions which achieve the supremum w.r.t $Z$ and Inequality 22 follows from $f_1^{(k,q)}, \ldots, f_t^{(k,q)}$ achieving the supremum w.r.t $Z_k^q$.

In addition, we show that $\mathbb{E}_k^q \left[ \tilde{Z}_k^q \right]$ is non-negative. Recall that $\mathbb{E}_k^q$ is conditioned on $\mathcal{A}_k^q$.

$$\mathbb{E}_k^q \left[ \tilde{Z}_k^q \right] = \mathbb{E}_k^q \left[ \left| \sum_{j,i=1}^{t,n} f_j^{(k,q)}(X_j^i) \right| - Z_k^q \right]$$

$$= \mathbb{E}_k^q \left[ \left| \sum_{j,i=1}^{t,n} f_j^{(k,q)}(X_j^i) \right| \right] - Z_k^q \quad \text{(since } Z_k^q \text{ is a constant w.r.t } \mathbb{E}_k^q\text{)}$$

$$\geq \left| \mathbb{E}_k^q \left[ \sum_{j,i=1}^{t,n} f_j^{(k,q)}(X_j^i) \right] \right| - Z_k^q \quad \text{(using Jensen's inequality)}.$$

Finally, each $f_j^{(k,q)}$ is a constant w.r.t $\mathbb{E}_k^q$ besides when $j = k$ and $i = q$ so the above reduces to:

$$\mathbb{E}_k^q \left[ \tilde{Z}_k^q \right] \geq \left| \sum_{\substack{j,i=1 \\ (j,i) \neq (k,q)}}^{t,n} f_j^{(k,q)}(X_j^i) \right| - Z_k^q$$

$$= 0.$$

Also, we have

$$(nt-1)Z = \left| (nt-1)\sum_{j=1}^{t}\sum_{i=1}^{n} f_j^{(0)}(X_j^i) \right|$$

$$= \left| \sum_{k,q=1}^{t,n} \left[ \sum_{\substack{j,i=1 \\ (j,i)\neq(k,q)}}^{t,n} f_j^{(0)}(X_j^i) \right] \right|$$

$$\leq \sum_{k,q=1}^{t,n} \left| \sum_{\substack{j,i=1 \\ (j,i)\neq(k,q)}}^{t,n} f_j^{(0)}(X_j^i) \right|$$

$$\leq \sum_{k,q=1}^{t,n} \left| \sum_{\substack{j,i=1 \\ (j,i)\neq(k,q)}}^{t,n} f_j^{(k,q)}(X_j^i) \right|$$

$$= \sum_{k,q=1}^{t,n} Z_k^q.$$

Therefore, $\sum_{j=1}^{t}\sum_{i=1}^{n}(Z - Z_k^q) \leq Z$.

Finally,

$$\sum_{k,q=1}^{t,n} \mathbb{E}_k^q\left[\left(\tilde{Z}_k^q\right)^2\right] = \sum_{k,q=1}^{t,n} \mathbb{E}_k^q\left[\left( \left| \sum_{j,i=1}^{t,n} f_j^{(k,q)}(X_j^i) \right| - \left| \sum_{\substack{j,i=1 \\ (j,i)\neq(k,q)}}^{t,n} f_j^{(k,q)}(X_j^i) \right| \right)^2\right]$$

$$\leq \sum_{k,q=1}^{t,n} \mathbb{E}_k^q\left[\left( f_k^{(k,q)}(X_k^q)\right)^2\right] \qquad \text{(by Corollary 14)}$$

As the samples for the same task are being drawn from the same distribution,

$$\sum_{k,q=1}^{t,n} \mathbb{E}_k^q\left[\left(\tilde{Z}_k^q\right)^2\right] = \sum_{k,q=1}^{t,n} \mathbb{E}_k^1\left[\left( f_k^{(k,1)}(X_k^1)\right)^2\right]$$

$$= n\sum_{k=1}^{t} \mathbb{E}_k^1\left[\left( f_k^{(k,1)}(X_k^1)\right)^2\right].$$

Note that $f_k^{(k,1)}$ is a fixed function w.r.t. $X_k^1$, therefore $f_k^{(k,1)}$ is a fixed function as we integrate over $X_k^1$. So , $\mathbb{E}_k^1\left[\left( f_k^{(k,1)}(X_k^1)\right)^2\right] \leq \sup_{f_j\in\mathcal{F}} \mathbb{E}_k^1\left[\left( f_j(X_k^1)\right)^2\right]$. Applying this reasoning to the $t$ functions gives the desired bound.

$$n\sum_{k=1}^{t} \mathbb{E}_k^1\left[\left( f_k^{(k,1)}(X_k^1)\right)^2\right] \leq n\sum_{k=1}^{t} \sup_{f_k\in\mathcal{F}} \mathbb{E}_k^1\left[\left( f_k(X_k^1)\right)^2\right]$$

$$= n\sup_{\boldsymbol{f}\in\mathcal{F}^{\otimes t}} \sum_{k=1}^{t} \mathbb{E}_k^1\left[\left( f_k(X_k^1)\right)^2\right].$$

Note, crucially, that the product space structure of our function class $\mathcal{F}^{\otimes t}$ allows us to move the suprema outside the summation. Dividing by $nt$ gives the desired bound.

$$\frac{1}{nt}\sum_{k,q=1}^{t,n} \mathbb{E}_k^q\left[\left(\tilde{Z}_k^q\right)^2\right] \leq \sup_{\boldsymbol{f}\in\mathcal{F}^{\otimes t}} \frac{1}{t}\sum_{k=1}^{t} \mathbb{E}_k^q\left[\left( f_k(X_k^1)\right)^2\right] = \sup_{\boldsymbol{f}\in\mathcal{F}^{\otimes t}} \frac{1}{t}\sum_{k=1}^{t} \mathrm{Var}\left[f_k(X_k^1)\right].$$

The right-hand side is bounded by $\sigma^2$ by hypothesis.

Finally, now we apply Corollary 1 as all the conditions are satisfied. $\qquad\square$

**Theorem 10** (Analog of Theorem 2.1 in Bartlett et al. [2005]). *Let $\mathcal{F}$ be a class of functions that map $\mathcal{X}$ into $[a,b]$. Given an input $\mathbf{X} = \left\{X_j^i\right\}_{i,j=1}^{t,n}$, assume that there is some $r > 0$ such that for all $f_1, \ldots, f_t$ that $\frac{1}{t}\sum_{j=1}^t \mathrm{Var}\big[f_j(X_j^1)\big] \leq r$. Then, for every $\delta > 0$, with probability $1 - e^{-\delta}$,*

$$\sup_{\boldsymbol{f} \in \mathcal{F}^{\otimes t}} (P\boldsymbol{f} - \hat{P}^n \boldsymbol{f}) \leq \inf_{\alpha} \left\{ 2(1+\alpha)\mathbb{E}\mathfrak{R}_{\mathbf{X}}(\mathcal{F}^{\otimes t}) + \sqrt{\frac{2\delta r}{tn}} + (b-a)\left(\frac{1}{\alpha} + \frac{1}{3}\right)\frac{\delta}{nt} \right\}.$$

*Proof of Theorem 10.* Let $V^+ = \sup_{\boldsymbol{f} \in \mathcal{F}^{\otimes t}}(P\boldsymbol{f} - \hat{P}^n \boldsymbol{f})$ where $\hat{P}^n \boldsymbol{f} = \frac{1}{nt}\sum_{j=1}^t \sum_{i=1}^n f_j(X_j^i)$ and $P\boldsymbol{f} = \mathbb{E}\hat{P}^n \boldsymbol{f}$. Let $\tilde{\mathcal{F}} = \left\{ \frac{f - \mathbb{E}f}{(b-a)} \mid f \in \mathcal{F} \right\}$ be a set of scaled functions. For each $\tilde{f} \in \tilde{\mathcal{F}}$ we have that $\mathbb{E}\tilde{f} = 0$ and $\left|\tilde{f}\right| \leq 1$. In addition, for all $\tilde{\boldsymbol{f}} \in \tilde{\mathcal{F}}^{\otimes t}$ we have that $\frac{1}{t}\sum_{j=1}^t \mathrm{Var}\big[\tilde{f}_j(X_j^1)\big] = \frac{1}{(b-a)^2}\frac{1}{t}\sum_{j=1}^t \mathrm{Var}\big[f_j(X_j^1)\big] \leq \frac{r}{(b-a)^2}$. Therefore, the conditions for Theorem 9 are satisfied.

Letting $Z = \sup_{\tilde{\boldsymbol{f}} \in \tilde{\mathcal{F}}^{\otimes t}} \sum_{j=1}^t \sum_{i=1}^n \tilde{f}_j(X_j^i) = \frac{nt}{(b-a)}V^+$ as in Equation (20), we thus have with probability $1 - e^{-\delta}$:

$$V^+ \leq \mathbb{E}V^+ + \sqrt{\frac{2\delta r}{tn} + \frac{4\delta\mathbb{E}V^+(b-a)}{nt} + \frac{(b-a)\delta}{nt3}}$$

$$\leq \mathbb{E}V^+ + \sqrt{\frac{2\delta r}{tn}} + \sqrt{\frac{4\delta\mathbb{E}V^+(b-a)}{nt}} + \frac{(b-a)\delta}{nt3} \tag{23}$$

$$\leq \mathbb{E}V^+ + \sqrt{\frac{2\delta r}{tn}} + \alpha\mathbb{E}V^+ + \frac{\delta(b-a)}{\alpha nt} + \frac{(b-a)\delta}{nt3} \tag{24}$$

$$\leq 2(1+\alpha)\mathbb{E}\tilde{\mathfrak{R}}_{\mathbf{X}}(\mathcal{F}^{\otimes t}) + \sqrt{\frac{2\delta r}{tn}} + (b-a)\left(\frac{1}{\alpha} + \frac{1}{3}\right)\frac{\delta}{nt} \tag{25}$$

where Inequality 23 is by subadditivity of the square root, Inequality 24 is by application of the AM-GM inequality, and Inequality 25 is by Lemma 10. $\qquad\square$

**Theorem 11.** *Let $\mathcal{F}$ be a class non-negative $b$ bounded functions. Assume that $X_j^1, \ldots, X_j^n$ are $n$ draws from $P_j$ for $j \in [t]$ and all $nt$ samples are independent. Suppose that*

$$\mathrm{Var}\big[\boldsymbol{f}\big] \leq V(\boldsymbol{f}) \leq BP\boldsymbol{f}$$

*where $V$ is a functional from $\mathcal{F}^{\otimes t}$ to $\mathbb{R}$ and $B > 0$. Given an input $\mathbf{X} = \left\{X_j^i\right\}_{i,j=1}^{t,n}$, let $\psi$ be a sub-root function with fixed point $r^*$ such that $B\mathbb{E}\tilde{\mathfrak{R}}_{\mathbf{X}}(\mathcal{F}^{\otimes t}, r) \leq \psi(r)$ for all $r \geq r^*$. Then, the following holds: For $K > 1$, $\alpha > 0$, and $x > 1$, with probability at least $1 - e^{-x}$.*

$$\forall \boldsymbol{f} \in \mathcal{F}^{\otimes t} \quad P\boldsymbol{f} \leq \frac{K}{K-1}\hat{P}^n \boldsymbol{f} + 200(1+\alpha)^2\frac{Kr^*}{B} + \frac{5}{4}\frac{BKx}{nt} + 2\left(\frac{1}{\alpha} + \frac{1}{3}\right)\frac{bx}{nt}.$$

**Remark 2** (Comparison to Yousefi et al. [2018]). *A Theorem of this form was shown by Yousefi et al. [2018], which generalized results from Bartlett et al. [2005] to vector valued functions. The proof of Theorem 11 follows the same steps as the proof of Theorem Yousefi et al. [2018]. Yet, we will use our Theorem 10 which has better constants than Theorem 1 within Yousefi et al. [2018] for when the class that the suprema is indexing over is has product space structure.*

The following lemma, and its proof, is a modified version of work in [Yousefi et al., 2018, Lemma B.2 ], which is a generalized version of work within [Bartlett et al., 2005, Lemma 3.8.]. We remove the Bernstein class structure, make $B > 0$, and only consider non-negative bounded functions [5].

---

[5] The reason we restrict to non-negative functions instead of functions that map to $[a,b]$ as done in prior work is because we failed to reproduce a step within Bartlett et al. [2005] and Yousefi et al. [2018]. Consider the last sentence within the first paragraph of the proof of Lemma 3.8. in Bartlett et al. [2005]. It is shown that $Pf \leq \hat{P}^n f + r/(\lambda BK)$. Now $\hat{P}^n f$ may be negative for $f$ which maps to $[a,b]$ with $a < 0$. Therefore, we cannot multiply this quantity by a value greater than one. In this case $K/(K-1)$ for $K > 1$.

The theorem can be generalized to Bernstien classes like Theorem B.3 from Yousefi et al. [2018] with worse dependence on $B$.

**Lemma 17.** *Let $K > 1, r > 0$ and $B > 0$. Suppose that for all $\boldsymbol{f} \in \mathcal{F}^{\otimes t}$ we have $V(\boldsymbol{f}) \leq BP\boldsymbol{f}$. Define the re-scaled version of $\mathcal{F}^{\otimes t}$ as*

$$\mathcal{F}_r^{\otimes t} = \left\{ \boldsymbol{f}' = (\boldsymbol{f}'_1, \ldots, \boldsymbol{f}'_t) \mid f'_j = \frac{rf_j}{\max(r, V(\boldsymbol{f}))}, \boldsymbol{f} = (\boldsymbol{f}_1, \ldots, \boldsymbol{f}_t) \in \mathcal{F}^{\otimes t} \right\}. \qquad (26)$$

*If $V_r^+ = \sup_{\boldsymbol{f}' \in \mathcal{F}_r^{\otimes t}} \left[ P\boldsymbol{f}' - \hat{P}^n \boldsymbol{f}' \right] \leq \frac{r}{BK}$, then*

$$\forall \boldsymbol{f} \in \mathcal{F}^{\otimes t}, \quad P\boldsymbol{f} \leq \frac{K}{K-1} \hat{P}^n \boldsymbol{f} + \frac{r}{BK}.$$

*Proof of Lemma 17.* Let $\boldsymbol{f}$ be any element in $\mathcal{F}^{\otimes t}$.

**Case 1.** If $V(\boldsymbol{f}) \leq r$, then $\boldsymbol{f}' = \boldsymbol{f}$ and the inequality $V_r^+ \leq \frac{r}{BK}$ leads to

$$P(\boldsymbol{f}) \leq \hat{P}^n(\boldsymbol{f}) + \frac{r}{BK}.$$

If all component functions of $\boldsymbol{f}$ are positive then $\hat{P}^n(\boldsymbol{f})$ is positive and as $K/(K-1) > 1$ for all $K > 1$ we have

$$P(\boldsymbol{f}) \leq \frac{K}{K-1} \hat{P}^n(\boldsymbol{f}) + \frac{r}{BK}.$$

**Case 2.** If $V(\boldsymbol{f}) \geq r$, then $\boldsymbol{f}' = r\boldsymbol{f}/V(\boldsymbol{f})$ and the inequality $V_r^+ \leq \frac{r}{BK}$ yields

$$P(\boldsymbol{f}) \leq \hat{P}^n(\boldsymbol{f}) + \frac{V(\boldsymbol{f})}{BK} \leq \hat{P}^n(\boldsymbol{f}) + \frac{P(\boldsymbol{f})}{K}.$$

Solving for $P(\boldsymbol{f})$ gives

$$P(\boldsymbol{f}) \leq \frac{K}{K-1} \hat{P}^n(\boldsymbol{f}) + \leq \frac{K}{K-1} \hat{P}^n(\boldsymbol{f}) + \frac{r}{BK}.$$

which completes the proof. $\qquad \square$

*Proof of Theorem 11.* We modify the proof within Yousefi et al. [2018], which generalises the proof within Bartlett et al. [2005] by extending it to multi-function and Bernstein classes. At a high level we apply the same steps as Yousefi et al. [2018], yet we use our concentration inequality and do not need to consider the more complicated inequality [Yousefi et al., 2018, Lemma B.1] as we do not have the additional Bernstein structure. The proof closely follows from Yousefi et al. [2018] with minor modifications as described and presented for completeness.

Let $r \geq r^*$ be a fixed real number.

**Verifying Conditions of Theorem 10.** We seek to use Theorem 10 on $\mathcal{F}_r^{\otimes t}$, see Equation (26), therefore we will validate the conditions of the theorem.

Fix $\boldsymbol{f}' \in \mathcal{F}_r^{\otimes t}$. Thus, $\boldsymbol{f}' = r\boldsymbol{f}/\max(r, V(\boldsymbol{f}))$ for some $\boldsymbol{f} \in \mathcal{F}$. If $\max(r, V(\boldsymbol{f})) = r$, then $\boldsymbol{f}' = r\boldsymbol{f}/\max(r, V(\boldsymbol{f})) = \boldsymbol{f}$. Thus, $\mathrm{Var}[\boldsymbol{f}'] = \mathrm{Var}[\boldsymbol{f}] \leq V(\boldsymbol{f}) \leq r$. If $\max(r, V(\boldsymbol{f})) = V(\boldsymbol{f})$, then $\mathrm{Var}[\boldsymbol{f}'] = \frac{r^2}{V^2(\boldsymbol{f})} \mathrm{Var}[\boldsymbol{f}] \leq \frac{r^2}{V(\boldsymbol{f})} \leq r$. Therefore, $\frac{1}{t} \sup_{\boldsymbol{f}' \in \mathcal{F}_r^{\otimes t}} \sum_{j=1}^t \mathbb{E}\left[f'_j(X_j)\right]^2 \leq r$. Finally, recall, functions in $\mathcal{F}$ are positive and $b$-bounded.

Thus, by Theorem 10, with probability at least $1 - e^{-\delta}$ and $\forall \delta > 0$

$$\sup_{\boldsymbol{f} \in \mathcal{F}^{\otimes t}} (P\boldsymbol{f} - \hat{P}^n \boldsymbol{f}) \leq \inf_{\alpha} \left\{ \leq 2(1+\alpha)\mathbb{E}\tilde{\mathfrak{R}}_{\mathbf{X}}(\mathcal{F}_r^{\otimes t}) + \sqrt{\frac{2\delta r}{tn}} + b\left(\frac{1}{\alpha} + \frac{1}{3}\right)\frac{\delta}{nt} \right\}.$$

**Bounding $\mathfrak{R}_\mathbf{X}(\mathcal{F}_r^{\otimes t})$.** Now we must control the Rademacher complexity within the above inequality. This part of the proof is the same for Bartlett et al. [2005], Yousefi et al. [2018].

Let $\mathcal{F}^{\otimes t}(u,v) := \{\boldsymbol{f} \in \mathcal{F}^{\otimes t} : u \leq V(\boldsymbol{f}) \leq v\}$ for all $0 \leq u \leq v$, and let

$$\mathcal{R}_\mathbf{X}(\boldsymbol{f}') = \frac{1}{nt} \sum_{j=1}^{t} \sum_{i=1}^{n} \sigma_j^i f_j'\left(X_j^i\right),$$

$$\mathcal{R}_\mathbf{X}\left(\mathcal{F}_r^{\otimes t}\right) = \sup_{f' \in \mathcal{F}_r^{\otimes t}} \left[\mathcal{R}_\mathbf{X}(\boldsymbol{f}')\right],$$

Observe that $\tilde{\mathfrak{R}}_\mathbf{X}\left(\mathcal{F}_r^{\otimes t}\right) = \mathbb{E}\mathcal{R}_\mathbf{X}\left(\mathcal{F}_r^{\otimes t}\right)$. By assumption we have that $V(\boldsymbol{f}) \leq B(P\boldsymbol{f}) \leq Bb, \forall \boldsymbol{f} \in \mathcal{F}^{\otimes t}$. Fix some $\lambda > 1$ and define $k$ as the smallest integer such that $r\lambda^{k+1} \geq Bb$. By leveraging the principle that combining function classes on a uniform set of inputs is equivalent to the Minkowski sum of the created image, and utilizing the trait of Rademacher width breakdown under Minkowski sum as detailed by Vershynin [2018], we have,

$$\tilde{\mathfrak{R}}_\mathbf{X}\left(\mathcal{G}_1 \cup \mathcal{G}_2\right) \leq \tilde{\mathfrak{R}}_\mathbf{X}\left(\mathcal{G}_1\right) + \tilde{\mathfrak{R}}_\mathbf{X}\left(\mathcal{G}_2\right).$$

We thus the following inequalities

$$\tilde{\mathfrak{R}}_\mathbf{X}\left(\mathcal{F}_r^{\otimes t}\right) = \mathbb{E}\left[\sup_{f' \in \mathcal{F}_r^{\otimes t}} \mathcal{R}_\mathbf{X}(\boldsymbol{f}')\right] = \mathbb{E}\left[\sup_{\boldsymbol{f} \in \mathcal{F}^{\otimes t}} \frac{1}{nt} \sum_{j=1}^{t} \sum_{i=1}^{n} \frac{r}{\max(r, V(\boldsymbol{f}))} \sigma_j^i f_j\left(X_j^i\right)\right]$$

$$\leq \mathbb{E}\left[\sup_{\boldsymbol{f} \in \mathcal{F}^{\otimes t}(0,r)} \frac{1}{nt} \sum_{j=1}^{t} \sum_{i=1}^{n} \sigma_j^i f_j\left(X_j^i\right)\right] + \mathbb{E}\left[\sup_{\boldsymbol{f} \in \mathcal{F}^{\otimes t}(r,Bb)} \frac{1}{nt} \sum_{j=1}^{t} \sum_{i=1}^{n} \frac{r}{V(\boldsymbol{f})} \sigma_j^i f_j\left(X_j^i\right)\right]$$

$$\leq \mathbb{E}\left[\sup_{\boldsymbol{f} \in \mathcal{F}^{\otimes t}(0,r)} \frac{1}{nt} \sum_{j=1}^{t} \sum_{i=1}^{n} \sigma_j^i f_j\left(X_j^i\right)\right] + \sum_{j=0}^{k} \lambda^{-j} \mathbb{E}\left[\sup_{\boldsymbol{f} \in \mathcal{F}^{\otimes t}(r\lambda^j, r\lambda^{j+1})} \mathcal{R}_\mathbf{X}(\boldsymbol{f})\right]$$

$$\leq \tilde{\mathfrak{R}}_\mathbf{X}(\mathcal{F}^{\otimes t}, r) + \sum_{j=0}^{k} \lambda^{-j} \tilde{\mathfrak{R}}_\mathbf{X}\left(\mathcal{F}^{\otimes t}, r\lambda^{j+1}\right)$$

As $r \geq r^*$, we can bound the local Rademacher width with the sub-root function.

$$\leq \frac{\psi(r)}{B} + \frac{1}{B} \sum_{j=0}^{k} \lambda^{-j} \psi\left(r\lambda^{j+1}\right).$$

Now the sub-root property of $\psi$ gives us that that $\psi(\xi r) \leq \xi^{\frac{1}{2}} \psi(r)$ for any $\xi \geq 1$ and, hence,

$$\tilde{\mathfrak{R}}_\mathbf{X}\left(\mathcal{F}_r^{\otimes t}\right) \leq \frac{\psi(r)}{B}\left(1 + \sqrt{\lambda} \sum_{j=0}^{k} \lambda^{-\frac{j}{2}}\right) \leq \frac{\psi(r)}{B}\left(1 + \frac{\lambda}{\sqrt{\lambda}-1}\right).$$

Pick $\lambda = 4$ in the above inequality, which gives $\tilde{\mathfrak{R}}_\mathbf{X}\left(\mathcal{F}_r^{\otimes t}\right) \leq 5\psi(r)/B$. Also, $r^*$ is the fixed point of $\psi$, we have for all $r \geq r^*$ that $\psi(r) \leq \sqrt{r/r^*}\psi(r^*) = \sqrt{rr^*}$. Taken together, we have

$$\tilde{\mathfrak{R}}_\mathbf{X}\left(\mathcal{F}_r^{\otimes t}\right) \leq \frac{5}{B}\sqrt{rr^*}, \quad \forall r \geq r^*.$$

**Applying Lemma 17.** The remainder of the proof is simple manipulations and applying Lemma 17. Here our proof deviates from [Yousefi et al., 2018]. Using our above bound on the Rademacher complexity gives for any $r \geq r^*$ and $\delta > 0$, we have with probability at least $1 - e^{-\delta}$,

$$\sup_{\boldsymbol{f} \in \mathcal{F}^{\otimes t}} (P\boldsymbol{f} - \hat{P}^n \boldsymbol{f}) \inf_\alpha \left\{(1+\alpha)\frac{10}{B}\sqrt{rr^*} + \sqrt{\frac{2\delta r}{tn}} + b\left(\frac{1}{\alpha} + \frac{1}{3}\right)\frac{\delta}{nt}\right\}$$

Now, letting $A = 10(1+\alpha)\sqrt{r^*}/B + \sqrt{2\delta/nt}$ and $C = b\left(\frac{1}{\alpha} + \frac{1}{3}\right)\frac{\delta}{nt}$,

$$\sup_{f' \in \mathcal{F}_r^{\otimes t}} \left[P\boldsymbol{f}' - \hat{P}^n \boldsymbol{f}'\right] \leq A\sqrt{r} + C.$$

Setting $A\sqrt{r} + C = \frac{r}{BK}$ gives a quadratic, which has both roots bounded by $(ABK)^2 + 2BKC$ by Lemma 12 now applying Lemma 17 we have that

$$P\boldsymbol{f} \leq \frac{K}{K-1}\hat{P}^n\boldsymbol{f} + BKA^2 + 2C.$$

Plugging in our values for $A$ and $C$ gives

$$P\boldsymbol{f} \leq \frac{K}{K-1}\hat{P}^n\boldsymbol{f} + 100\,(1+\alpha)^2\,\frac{Kr^*}{B} + 2\frac{BK\delta}{nt} + 20\sqrt{2}\,(1+\alpha)\,K\sqrt{r^*}\sqrt{\frac{\delta}{nt}} + 2\left(\frac{1}{3}+\frac{1}{\alpha}\right)\frac{b\delta}{nt}.$$

Applying AM-GM inequality,

$$P\boldsymbol{f} \leq \frac{K}{K-1}\hat{P}^n\boldsymbol{f} + 101\,(1+\alpha)^2\,\frac{Kr^*}{B} + 802\frac{BK\delta}{nt} + 2\left(\frac{1}{3}+\frac{1}{\alpha}\right)\frac{b\delta}{nt},$$

which completes the proof. $\hfill\square$

### D.2 Multi-Task Learning

The following theorem is a vector-valued generalization of Theorem 5.2 within Bartlett et al. [2005] with several differences: we do not require $\frac{\delta}{n} \leq r^*$, $b = 1$, and we solve for constants.

**Theorem 12.** *Let $(\hat{\boldsymbol{f}}, \hat{h})$ be an empirical risk minimizer as given in Eqn. (2). Let $\psi(r) \geq b\mathbb{E}\tilde{\mathfrak{R}}_{\mathbf{X}}(\mathcal{F}^{\otimes t}(\mathcal{H}), r)$ with $r^*$ the fixed point of $\psi(r)$. Then, under Assumption 1.A with probability $1 - 2e^{-\delta}$,*

$$R_{\text{source}}(\hat{\boldsymbol{f}}, \hat{h}) \leq R_{\text{source}}(\boldsymbol{f}^*, h^*) + \sqrt{R_{\text{source}}(\boldsymbol{f}^*, h^*)}\left(6\sqrt{\frac{b\delta}{nt}} + 146\sqrt{\frac{r^*}{b}}\right) + \frac{102b\delta}{nt} + \frac{217r^*}{b}.$$
(27)

*Proof of Theorem 12.* To simplify the notation within the proof let $\boldsymbol{L}^* = R_{\text{source}}(\boldsymbol{f}^*, h^*)$.

We follow steps similar to the proof of Theorem 5.2 within Bartlett et al. [2005]. Assume $f_j^* = \arg\min_{f \in \mathcal{F}} P_j\ell_f$ exists. (As mentioned in Bartlett et al. [2005] if it does not exist we can consider a sequence converging to the infimum.) Note that $\{f_j^*(X_j^i)\}_{i \in [n], j \in [t]}$ are $nt$ independent r.v.. Note that

$$\frac{1}{nt}\sum_{j=1}^{t}\sum_{j=1}^{n}\text{Var}\left[f_j^*(X_j^i)\right] \leq \frac{1}{nt}\sum_{j=1}^{t}\sum_{j=1}^{n}\mathbb{E}\left(f_j^*(X_j^i)\right)^2 \leq \frac{b}{nt}\sum_{j=1}^{t}\sum_{j=1}^{n}\mathbb{E}f_j^*(X_j^i) = b\boldsymbol{L}^*.$$

By Bernstein inequality, with probability $1 - e^{-\delta}$

$$\hat{P}^n\hat{\boldsymbol{f}} \leq \hat{P}^n\boldsymbol{f}^* \leq \boldsymbol{L}^* + \sqrt{\frac{2\boldsymbol{L}^*b\delta}{tn}} + \frac{2}{3}\frac{b\delta}{tn}.$$

By Theorem 11, setting $\alpha = \frac{1}{11}$ and $B = b$, and union bound, with probability $1 - 2e^{-\delta}$,

$$P\ell_{\hat{f}} \leq \frac{K}{K-1}\left(\sqrt{2}\sqrt{\frac{\boldsymbol{L}^*b\delta}{nt}} + \boldsymbol{L}^* + \frac{2b\delta}{3nt}\right) + \frac{1789bK\delta}{882nt} + \frac{68b\delta}{3nt} + \frac{144Kr^*}{b}.$$

By change of variables we consider $K > 0$ by adding 1 to $K$:

$$\frac{1789b\delta\,(K+1)}{882nt} + \frac{68b\delta}{3nt} + \frac{144r^*\,(K+1)}{b} + \frac{(K+1)\left(\sqrt{2}\sqrt{\frac{\boldsymbol{L}^*b\delta}{nt}} + \boldsymbol{L}^* + \frac{2b\delta}{3nt}\right)}{K}$$

$$= \frac{1789Kb\delta}{882nt} + \frac{144Kr^*}{b} + \sqrt{2}\sqrt{\frac{\boldsymbol{L}^*b\delta}{nt}} + \boldsymbol{L}^* + \frac{22369b\delta}{882nt} + \frac{144r^*}{b} + \frac{\sqrt{2}}{K}\sqrt{\frac{\boldsymbol{L}^*b\delta}{nt}} + \frac{\boldsymbol{L}^*}{K} + \frac{2b\delta}{3Knt}.$$

Applying AM-GM to $\frac{\sqrt{2}}{K}\sqrt{\frac{L^*b\delta}{nt}}$,

$$L^* + \frac{1789Kb\delta}{882nt} + \frac{144Kr^*}{b} + \sqrt{2}\sqrt{\frac{L^*b\delta}{nt}} + \frac{22369b\delta}{882nt} + \frac{144r^*}{b} + \frac{2L^*}{K} + \frac{5b\delta}{3Knt}.$$

These are the terms without $K$:

$$L^* + \sqrt{2}\sqrt{\frac{L^*b\delta}{nt}} + \frac{22369b\delta}{882nt} + \frac{144r^*}{b}. \tag{28}$$

Considering only those terms a function of $K$:

$$\frac{1789Kb\delta}{882nt} + \frac{144Kr^*}{b} + \frac{2L^*}{K} + \frac{5b\delta}{3Knt}.$$

We set $K = \frac{\max\left(\sqrt{L^*},\sqrt{\frac{b\delta}{nt}}\right)}{\max\left(\sqrt{\frac{r^*}{b}},\sqrt{\frac{b\delta}{nt}}\right)}$. Terms with $K$ in the numerator are upper-bounded by

$$\frac{1789b\delta\left(\sqrt{L^*}+\sqrt{\frac{b\delta}{nt}}\right)}{882nt\max\left(\sqrt{\frac{r^*}{b}},\sqrt{\frac{b\delta}{nt}}\right)} + \frac{144r^*\left(\sqrt{L^*}+\sqrt{\frac{b\delta}{nt}}\right)}{b\max\left(\sqrt{\frac{r^*}{b}},\sqrt{\frac{b\delta}{nt}}\right)}.$$

Terms with $K$ in the denominator are upper-bounded by

$$\frac{2L^*\left(\sqrt{\frac{b\delta}{nt}}+\sqrt{\frac{r^*}{b}}\right)}{\max\left(\sqrt{L^*},\sqrt{\frac{b\delta}{nt}}\right)} + \frac{5b\delta\left(\sqrt{\frac{b\delta}{nt}}+\sqrt{\frac{r^*}{b}}\right)}{3nt\max\left(\sqrt{L^*},\sqrt{\frac{b\delta}{nt}}\right)}.$$

Expanding both expressions gives eight terms where we will choose an appropriate value to maximize the quantity:

$$\frac{2\boldsymbol{L}^*\sqrt{r^*}}{\sqrt{b}\max\left(\sqrt{\boldsymbol{L}^*},\sqrt{\frac{b\delta}{nt}}\right)} \leq 2\sqrt{\frac{\boldsymbol{L}^*r^*}{b}}$$

$$\frac{144\sqrt{\boldsymbol{L}^*}r^*}{b\max\left(\sqrt{\frac{r^*}{b}},\sqrt{\frac{b\delta}{nt}}\right)} \leq 144\sqrt{\frac{\boldsymbol{L}^*r^*}{b}}$$

$$\frac{5b^{\frac{3}{2}}\delta^{\frac{3}{2}}}{3n^{\frac{3}{2}}t^{\frac{3}{2}}\max\left(\sqrt{\boldsymbol{L}^*},\sqrt{\frac{b\delta}{nt}}\right)} \leq \frac{5b\delta}{3nt}$$

$$\frac{1789b^{\frac{3}{2}}\delta^{\frac{3}{2}}}{882n^{\frac{3}{2}}t^{\frac{3}{2}}\max\left(\sqrt{\frac{r^*}{b}},\sqrt{\frac{b\delta}{nt}}\right)} \leq \frac{1789b\delta}{882nt}$$

$$\frac{2\boldsymbol{L}^*\sqrt{b}\sqrt{\delta}}{\sqrt{n}\sqrt{t}\max\left(\sqrt{\boldsymbol{L}^*},\sqrt{\frac{b\delta}{nt}}\right)} \leq 2\sqrt{\frac{\boldsymbol{L}^*b\delta}{nt}}$$

$$\frac{144r^*\sqrt{\delta}}{\sqrt{b}\sqrt{n}\sqrt{t}\max\left(\sqrt{\frac{r^*}{b}},\sqrt{\frac{b\delta}{nt}}\right)} \leq 144\sqrt{\frac{r^*\delta}{nt}}$$

$$\frac{5\sqrt{b}\sqrt{r^*}\delta}{3nt\max\left(\sqrt{\boldsymbol{L}^*},\sqrt{\frac{b\delta}{nt}}\right)} \leq \frac{5}{3}\sqrt{\frac{r^*\delta}{nt}}$$

$$\frac{1789\sqrt{\boldsymbol{L}^*}b\delta}{882nt\max\left(\sqrt{\frac{r^*}{b}},\sqrt{\frac{b\delta}{nt}}\right)} \leq \frac{1789}{882}\sqrt{\frac{\boldsymbol{L}^*b\delta}{nt}}.$$

Adding all of the right-hand side expressions together gives:

$$\frac{3553}{882}\sqrt{\frac{\boldsymbol{L}^*b\delta}{nt}} + 146\sqrt{\frac{\boldsymbol{L}^*r^*}{b}} + \frac{3259b\delta}{882nt} + \frac{437}{3}\sqrt{\frac{r^*\delta}{nt}}.$$

Adding the terms without $K$, Equation (28) gives:

$$\sqrt{2}\sqrt{\frac{\boldsymbol{L}^*b\delta}{nt}} + \frac{3553}{882}\sqrt{\frac{\boldsymbol{L}^*b\delta}{nt}} + 146\sqrt{\frac{\boldsymbol{L}^*r^*}{b}} + \boldsymbol{L}^* + \frac{12814b\delta}{441nt} + \frac{437}{3}\sqrt{\frac{r^*\delta}{nt}} + \frac{144r^*}{b}.$$

Applying AM-GM again

$$\sqrt{\boldsymbol{L}^*}\left(\sqrt{2}\sqrt{\frac{b\delta}{nt}} + \frac{3553}{882}\sqrt{\frac{b\delta}{nt}} + 146\sqrt{\frac{r^*}{b}}\right) + \boldsymbol{L}^* + \frac{89867b\delta}{882nt} + \frac{1301r^*}{6b}.$$

Rounding up to the nearest integers:

$$\sqrt{\boldsymbol{L}^*}\left(6\sqrt{\frac{b\delta}{nt}} + 146\sqrt{\frac{r^*}{b}}\right) + \boldsymbol{L}^* + \frac{102b\delta}{nt} + \frac{217r^*}{b}.$$

$\square$

## D.3 Multi-Task Learning via Representation Learning

**Theorem 13.** *Let $\hat{h}$ be an empirical risk minimizer as in (2). Let $\psi(r) \geq b\mathbb{E}\tilde{\mathfrak{R}}_{\mathbf{Z}}(\ell \circ \mathcal{F}^{\otimes t} \circ \mathcal{H}, r)$ with $r^*$ the fixed point of $\psi(r)$. Then, if Assumption 1.A holds. Then, with probability at least $1 - e^{-\delta}$,*

$$\bar{d}_{\mathcal{F},f^*}(\hat{h}, h^*) \leq \sqrt{R_{\text{source}}(\boldsymbol{f}^*, h^*)} \left( 6\sqrt{\frac{b\delta}{nt}} + 146\sqrt{\frac{r^*}{b}} \right) + \frac{102b\delta}{nt} + \frac{217r^*}{b},$$

*where recall that $\bar{d}_{\mathcal{F},f^*}$ is defined in Equation (12).*

*Proof of Theorem 13.*

$$\begin{aligned}
\bar{d}_{\mathcal{F},f^*}(\hat{h}, h^*) &= \inf_{\boldsymbol{f}' \in \mathcal{F}^{\otimes t}} \left\{ R_{\text{source}}(\boldsymbol{f}', \hat{h}) - R_{\text{source}}(\boldsymbol{f}^*, h^*) \right\} \\
&\leq R_{\text{source}}(\hat{\boldsymbol{f}}, \hat{h}) - R_{\text{source}}(\boldsymbol{f}^*, h^*) \\
&\leq \sqrt{R_{\text{source}}(\boldsymbol{f}^*, h^*)} \left( 6\sqrt{\frac{b\delta}{nt}} + 146\sqrt{\frac{r^*}{b}} \right) + \frac{102b\delta}{nt} + \frac{217r^*}{b}.
\end{aligned}$$

where the last inequality is due to Theorem 12. $\qquad\square$

By using our hypothesis which rates the task-averaged representation distance to the worst case representation difference we can bound the excess transfer risk.

**Theorem 14.** *Let $\hat{f}_0$ be an empirical risk minimizer of $\hat{R}_{target}(\cdot, \hat{h})$ in Eqn. (3) for any representation $\hat{h}$. Let $\psi(r) \geq b\mathbb{E}\tilde{\mathfrak{R}}_{\mathbf{Z}}(\ell \circ \mathcal{F}_0, r)$ with $r^*$ the fixed point of $\psi(r)$. Then if Assumption 1.A holds, with probability at least $1 - \delta$ :*

$$\begin{aligned}
R_{target}(\hat{f}_0, \hat{h}) &\leq R_{target}(f_0^*, h^*) + \sqrt{R_{target}(f_0^*, h^*)} \left( 9\sqrt{\frac{b\delta}{m}} + 219\sqrt{\frac{r}{b}} \right) \\
&\quad + \frac{171b\delta}{m} + \frac{21967r}{2b} + d_{\mathcal{F},\mathcal{F}_0}(\hat{h}; h^*).
\end{aligned}$$

*Proof of Theorem 14.* Let $\tilde{f}_0 = \arg\min_{f \in \mathcal{F}} R_{\text{target}}(f, \hat{h})$. By adding and subtracting $d_{\mathcal{F},\mathcal{F}_0}(\hat{h}; h^*)$ then applying Theorem 12 to $R_{\text{target}}(\hat{f}_0, \hat{h})$, we have convenient cancellation of $R_{\text{target}}(\tilde{f}_0, \hat{h})$. For notational succinctness let

$$A_t = 6\sqrt{\frac{b\delta}{m}} + 146\sqrt{\frac{r^*}{b}} \quad \text{and} \quad B_t = \frac{102b\delta}{m} + \frac{217r^*}{b}.$$

Therefore we have:

$$\begin{aligned}
R_{\text{target}}(\hat{f}_0, \hat{h}) &= R_{\text{target}}(\hat{f}_0, \hat{h}) - d_{\mathcal{F},\mathcal{F}_0}(\hat{h}; h^*) + d_{\mathcal{F},\mathcal{F}_0}(\hat{h}; h^*) \\
&= R_{\text{target}}(\hat{f}_0, \hat{h}) - \left( R_{\text{target}}(\tilde{f}_0, \hat{h}) - R_{\text{target}}(f_0^*, h^*) \right) + d_{\mathcal{F},\mathcal{F}_0}(\hat{h}; h^*) \\
&\leq R_{\text{target}}(\tilde{f}_0, \hat{h}) + \sqrt{R_{\text{target}}(\tilde{f}_0, \hat{h})} A_t + B_t \\
&\quad - \left( R_{\text{target}}(\tilde{f}_0, \hat{h}) - R_{\text{target}}(f_0^*, h^*) \right) + d_{\mathcal{F},\mathcal{F}_0}(\hat{h}; h^*) \\
&= \sqrt{R_{\text{target}}(\tilde{f}_0, \hat{h})} A_t + B_t + R_{\text{target}}(f_0^*, h^*) + d_{\mathcal{F},\mathcal{F}_0}(\hat{h}; h^*).
\end{aligned}$$

Now note that $R_{\text{target}}(\tilde{f}_0, \hat{h}) = R_{\text{target}}(f_0^*, h^*) + \bar{d}_{\mathcal{F}_0, f_0^*}(\hat{h}, h^*)$. After applying subadditivity of the square root $\sqrt{\cdot}$ and AM-GM we can bound $\bar{d}_{\mathcal{F}_0, f_0^*}(\hat{h}, h^*)$ with Theorem 13. This is concretely

demonstrated with the following chain of inequalities:

$$R_{\text{target}}(\hat{f}_0, \hat{h}) \leq \sqrt{R_{\text{target}}(f_0^*, h^*) + \bar{d}_{\mathcal{F}_0, f_0^*}(\hat{h}, h^*)} A_t + B_t + R_{\text{target}}(f_0^*, h^*) + d_{\mathcal{F}, \mathcal{F}_0}(\hat{h}; h^*)$$

$$\leq \sqrt{R_{\text{target}}(f_0^*, h^*)} A_t + \sqrt{\bar{d}_{\mathcal{F}_0, f_0^*}(\hat{h}, h^*)} A_t + B_t + R_{\text{target}}(f_0^*, h^*) + d_{\mathcal{F}, \mathcal{F}_0}(\hat{h}; h^*)$$

$$\leq \sqrt{R_{\text{target}}(f_0^*, h^*)} A_t + \frac{\bar{d}_{\mathcal{F}_0, f_0^*}(\hat{h}, h^*)}{2} + \frac{A_t^2}{2} + B_t + R_{\text{target}}(f_0^*, h^*) + d_{\mathcal{F}, \mathcal{F}_0}(\hat{h}; h^*)$$

$$\leq \sqrt{R_{\text{target}}(f_0^*, h^*)} A_t + \frac{\sqrt{R_{\text{target}}(f_0^*, h^*)} A_t + B_t}{2}$$

$$+ \frac{A_t^2}{2} + B_t + R_{\text{target}}(f_0^*, h^*) + d_{\mathcal{F}, \mathcal{F}_0}(\hat{h}; h^*)$$

$$\leq \sqrt{R_{\text{target}}(f_0^*, h^*)} \frac{3}{2} A_t + \frac{A_t^2}{2} + \frac{3}{2} B_t + R_{\text{target}}(f_0^*, h^*) + d_{\mathcal{F}, \mathcal{F}_0}(\hat{h}; h^*).$$

We can apply AM-GM to $A_t^2 = \left( 6\sqrt{\frac{b\delta}{m}} + 146\sqrt{\frac{r^*}{b}} \right)$ to get $A^2 \leq \frac{72b\delta}{m} + \frac{42632r}{b}$.

Finally, by collecting like terms, we have that $R_{\text{target}}(\hat{f}_0, \hat{h})$ is bounded by

$$\sqrt{R_{\text{target}}(f_0^*, h^*)} \left( 9\sqrt{\frac{b\delta}{m}} + 219\sqrt{\frac{r}{b}} \right) + \frac{171b\delta}{m} + \frac{21967r}{2b} + R_{\text{target}}(f_0^*, h^*) + d_{\mathcal{F}, \mathcal{F}_0}(\hat{h}; h^*).$$

$\square$

**Theorem 15.** *Let $\hat{h}$ and $\hat{f}_0$ be the learned representation and target predictor, as described in Eqns. (2) and (3). Let $\psi_1(r) \geq b\mathbb{E}\tilde{\mathfrak{R}}_{\mathbf{Z}}(\ell \circ \mathcal{F}^{\otimes t} \circ \mathcal{H}, r)$ and $\psi_2(r) \geq b\mathbb{E}\tilde{\mathfrak{R}}_{\mathbf{Z}}(\ell \circ \mathcal{F}_0, r)$ with $r_1^*$ and $r_2^*$ the fixed points of $\psi_1(r)$ and $\psi_2(r)$, respectively. Then, under Assumption 1.A and that $\boldsymbol{f}^*$ is $(\nu, \epsilon)$-diverse over $\mathcal{F}_0$ w.r.t. $h^*$, with probability at least $1 - 4e^{-\delta}$, the transfer learning risk is upper-bounded by,*

$$R_{target}(\hat{f}_0, \hat{h}) \leq R_{target}(f_0^*, h^*) + \sqrt{R_{target}(f_0^*, h^*)} \left( 9\sqrt{\frac{b\delta}{m}} + 219\sqrt{\frac{r_1^*}{b}} \right) + \frac{171b\delta}{m} + \frac{21967r_1^*}{2b}$$

$$+ \frac{1}{\nu} \left( \sqrt{R_{\text{source}}(\boldsymbol{f}^*, h^*)} \left( 6\sqrt{\frac{b\delta}{nt}} + 146\sqrt{\frac{r_2^*}{b}} \right) + \frac{102b\delta}{nt} + \frac{217r_2^*}{b}. \right) + \varepsilon.$$

*Proof of Theorem 15.* The proof is by first applying Theorem 14, then $(\nu, \varepsilon)$-diversity assumption, and finally Theorem 13. This is demonstrated with the following series of inequalities.

$$R_{\text{target}}(\hat{f}_0, \hat{h}) \leq \sqrt{R_{\text{target}}(f_0^*, h^*)} \left( 9\sqrt{\frac{b\delta}{m}} + 219\sqrt{\frac{r_1^*}{b}} \right) + \frac{171b\delta}{m} + \frac{21967r_1^*}{2b} + R_{\text{target}}(f_0^*, h^*)$$

$$+ d_{\mathcal{F}, \mathcal{F}_0}(\hat{h}; h^*)$$

$$\leq \sqrt{R_{\text{target}}(f_0^*, h^*)} \left( 9\sqrt{\frac{b\delta}{m}} + 219\sqrt{\frac{r_1^*}{b}} \right) + \frac{171b\delta}{m} + \frac{21967r_1^*}{2b} + R_{\text{target}}(f_0^*, h^*)$$

$$+ \frac{1}{\nu} \left( \sqrt{R_{\text{source}}(\boldsymbol{f}^*, h^*)} \left( 6\sqrt{\frac{b\delta}{nt}} + 146\sqrt{\frac{r_2^*}{b}} \right) + \frac{102b\delta}{nt} + \frac{217r_2^*}{b}. \right) + \varepsilon.$$

$\square$

## D.4 Empirical Local Rademacher Complexity Bound

The following lemma allows us to discard the composition of a loss so long as we appropriately scale the cover.

**Lemma 18.** *For a non-negative $H$-smooth loss and any function class $\mathcal{F}$, we have:*

$$\mathcal{N}\left(\mathcal{L}_\ell(\mathcal{F}^{\otimes t}, r), \epsilon, nt\right) \leq \mathcal{N}\left(\mathcal{F}^{\otimes t}, \frac{\epsilon}{\sqrt{12Hrnt}}, nt\right).$$

We follow the same first few steps as in Srebro et al. [2010] but then bound the maximum by summation to recover an empirical $L_2$ cover.

*Proof of Lemma 18.* By Lemma 13 we see that for a non-negative $H$-smooth function $f$, we have that $(f(t) - f(r))^2 \leq 6H(f(t) + f(r))(t-r)^2$. Using this inequality, for any sample $\mathbf{X} = \left\{(x_j^i, y_j^i)\right\}_{\substack{j\in[t], \\ i\in[n]}}$, of $nt$ points, for an $f_\epsilon$, we have,

$$\sup_{\mathbf{X}} \sqrt{\frac{1}{nt} \sum_{i=1}^{n} \sum_{j=1}^{t} \left(\ell\left(f\left(x_j^i\right), y_j^i\right) - \ell\left(f_\varepsilon\left(x_j^i\right), y_j^i\right)\right)^2}$$

$$\leq \sup_{\mathbf{X}} \sqrt{\frac{6H}{nt} \sum_{i=1}^{n} \sum_{j=1}^{t} \left(\ell\left(f\left(x_j^i\right), y_j^i\right) + \ell\left(f_\varepsilon\left(x_j^i\right), y_j^i\right)\right) \left(f\left(x_j^i\right) - f_\varepsilon\left(x_j^i\right)\right)^2}$$

$$\leq \sup_{\mathbf{X}} \sqrt{\frac{6H}{nt} \sum_{i=1}^{n} \sum_{j=1}^{t} \left(\ell\left(f\left(x_j^i\right), y_j^i\right) + \ell\left(f_\varepsilon\left(x_i\right), y_i\right)\right)} \sqrt{\max_{i\in[n]}\left(f\left(x_i\right) - f_\varepsilon\left(x_i\right)\right)^2}$$

$$\leq \sup_{\mathbf{X}} \sqrt{12Hr} \sqrt{\max_{i\in[n],j\in[t]}\left(f\left(x_j^i\right) - f_\varepsilon\left(x_j^i\right)\right)^2}$$

$$\leq \sup_{\mathbf{X}} \sqrt{12Hr} \sqrt{\sum_{i=1}^{n} \sum_{j=1}^{t}\left(f\left(x_j^i\right) - f_\varepsilon\left(x_j^i\right)\right)^2}$$

$$= \sup_{\mathbf{X}} \sqrt{12Hrnt} \sqrt{\frac{1}{nt} \sum_{i=1}^{n} \sum_{j=1}^{t}\left(f\left(x_j^i\right) - f_\varepsilon\left(x_j^i\right)\right)^2}.$$

That is, a cover of $\{\mathbf{f} \in \mathcal{F}^{\otimes t} : P\mathbf{f} \leq r\}$ at radius $\epsilon/\sqrt{12Hrnt}$ is also a cover of $\mathcal{L}_\ell(\mathcal{F}^\otimes, r)$ at radius $\epsilon$, and we can conclude that,

$$\mathcal{N}\left(\mathcal{L}_\ell(\mathcal{F}^{\otimes t}, r), \epsilon, nt\right) \leq \mathcal{N}\left(\mathcal{F}^{\otimes t}, \frac{\epsilon}{\sqrt{12Htrn}}, nt\right),$$

which completes the proof. $\qquad\square$

**Theorem 16** (Smooth non-negative local Rademacher complexity bound). *Under the setting of Theorem 15 along with $\ell$ being $H$-smooth and Assumptions 1.B and 1.C, we have:*

$$\tilde{\mathfrak{R}}_{nt}(\mathcal{L}_\ell(\mathcal{F}^{\otimes t}(\mathcal{H})), r) \leq 640\sqrt{3}G\left(\mathcal{F}^{\otimes t}(\mathcal{H})\right)\sqrt{Hr}\log\left(nt\right)^2 + \frac{80\sqrt{3}D\sqrt{Hr}\log\left(nt\right) + 4\sqrt{br}}{\sqrt{nt}},$$

(29)

*and if $16G\left(\mathcal{F}^{\otimes t}(\mathcal{H})\right) \leq D$ and $\sqrt{b} \geq 640\sqrt{3}\Pi(\mathcal{F}^{\otimes t}(H))$, which holds for large enough samples,*

$$\tilde{\mathfrak{R}}_{nt}(\mathcal{L}_\ell(\mathcal{F}^{\otimes t}(\mathcal{H})), r) \leq 2560\sqrt{3}\sqrt{r}\Pi(\mathcal{F}^{\otimes t}(H))\log\left(\frac{e\sqrt{3}\sqrt{b}}{1920\Pi(\mathcal{F}^{\otimes t}(H))}\right)$$

(30)

*where $G(\mathcal{F}^{\otimes t}(\mathcal{H})) = L\tilde{\mathfrak{G}}_{nt}(\mathcal{H}) + \tilde{\mathfrak{G}}_n(\mathcal{F})$ and $\Pi(\mathcal{F}^{\otimes t}(H)) = \sqrt{H}G\left(\mathcal{F}^{\otimes t}(\mathcal{H})\right)\log\left(\frac{eD}{16G(\mathcal{F}^{\otimes t}(\mathcal{H}))}\right).$*

*Proof of Theorem 16.* Let $\mathbf{Z}$ denote the dataset of $nt$ points. We first compute $\sup_{\boldsymbol{g} \in \mathcal{F}^{\otimes t}(\mathcal{H})} \sup_{\mathbf{Z}} \|f\|_{L^2(\mathbf{Z})}$. Note that,

$$\sup_{\boldsymbol{g} \in \ell(\mathcal{F}^{\otimes t}(\mathcal{H}))} \sup_{\mathbf{Z}} \|\boldsymbol{g}\|_{L^2(\mathbf{Z})} \leq \sup_{\boldsymbol{f} \in \mathcal{F}^{\otimes t}, h \in \mathcal{H}} \sup_{\mathbf{Z}} \|\ell \circ \boldsymbol{f} \circ h\|_{L^2(\mathbf{Z})}$$

$$= \sup_{\boldsymbol{f} \in \mathcal{F}^{\otimes t}, h \in \mathcal{H}} \frac{1}{nt} \sum_{j=1}^{t} \sum_{i=1}^{n} \left(\ell(f_j(h(x_j^i)))\right)^2$$

$$\leq \sup_{\boldsymbol{f} \in \mathcal{F}^{\otimes t}, h \in \mathcal{H}} b\frac{1}{nt} \sum_{j=1}^{t} \sum_{i=n}^{n} \ell(f_j(h(x_j^i))) \leq br.$$

Therefore, by refined Dudley's entropy integral formula, we have

$$\tilde{\mathfrak{R}}_{nt}(\mathcal{L}_\ell(\mathcal{F}^{\otimes t}(\mathcal{H})), r) \leq \inf_{\sqrt{br} \geq \alpha \geq 0} \left\{ 4\alpha + 10 \int_\alpha^{\sqrt{br}} \sqrt{\frac{\log \mathcal{N}(\mathcal{L}_\ell(r), \varepsilon, nt)}{nt}} \, d\varepsilon \right\} \qquad \text{(by Lemma 7)}$$

$$\leq \inf_{\sqrt{br} \geq \alpha \geq 0} \left\{ 4\alpha + 10 \int_\alpha^{\sqrt{br}} \sqrt{\frac{\log \mathcal{N}(\mathcal{F}^{\otimes t}(\mathcal{H}), \frac{\varepsilon}{\sqrt{12Hrnt}}, nt)}{nt}} \, d\varepsilon \right\} \qquad \text{(by Lemma 18)}$$

$$\leq \inf_{\sqrt{br} \geq \alpha \geq 0} \left\{ 4\alpha + 20 \int_\alpha^{\sqrt{br}} \frac{\sqrt{12Hrnt}\tilde{\mathcal{G}}_{nt}(\mathcal{F}^{\otimes t}(\mathcal{H}))}{\varepsilon} \sqrt{\frac{1}{nt}} \, d\varepsilon \right\} \qquad \text{(by Lemma 2)}$$

$$= \inf_{\sqrt{br} \geq \alpha \geq 0} \left\{ 4\alpha + 40\sqrt{3Hr}\tilde{\mathcal{G}}_{nt}(\mathcal{F}^{\otimes t}(\mathcal{H})) \log\left(\frac{\sqrt{br}}{\alpha}\right) \right\}. \qquad (31)$$

Now we apply the Gaussian chain rule Theorem 8 to $\tilde{\mathcal{G}}_{nt}(\mathcal{F}^{\otimes t}(\mathcal{H}))$, giving us:

$$\tilde{\mathfrak{R}}_{nt}(\mathcal{L}_\ell(\mathcal{F}^{\otimes t}(\mathcal{H})), r) \leq \inf_{\sqrt{br} \geq \alpha \geq 0} \left\{ 4\alpha + 40\sqrt{3Hr} \inf_{D \geq \delta \geq 0} \left\{ 4\delta + 64G\left(\mathcal{F}^{\otimes t}(\mathcal{H})\right) \log\left(\frac{D}{\delta}\right) \right\} \log\left(\frac{\sqrt{br}}{\alpha}\right) \right\}.$$

Setting $\delta = \min(16G\left(\mathcal{F}^{\otimes t}(\mathcal{H})\right), D)$ and simplifying by bounding the minimum with $16G\left(\mathcal{F}^{\otimes t}(\mathcal{H})\right)$ we have:

$$\tilde{\mathfrak{R}}_{nt}(\mathcal{L}_\ell(\mathcal{F}^{\otimes t}(\mathcal{H})), r) \leq 2560\sqrt{3}G\left(\mathcal{F}^{\otimes t}(\mathcal{H})\right) \sqrt{Hr} \log\left(\frac{eD}{\min\left(16G\left(\mathcal{F}^{\otimes t}(\mathcal{H})\right), D\right)}\right) \log\left(\frac{\sqrt{br}}{\alpha}\right) + 4\alpha$$

Doing the same for $\delta$ with $\min\left(\sqrt{br}, 640\sqrt{3}G\left(\mathcal{F}^{\otimes t}(\mathcal{H})\right) \sqrt{Hr} \log\left(\frac{eD}{\min(16G(\mathcal{F}^{\otimes t}(\mathcal{H})), D)}\right)\right)$ we have

$$\tilde{\mathfrak{R}}_{nt}(\mathcal{L}_\ell(\mathcal{F}^{\otimes t}(\mathcal{H})), r) \leq 2560\sqrt{3}G\left(\mathcal{F}^{\otimes t}(\mathcal{H})\right) \sqrt{Hr}$$

$$\times \log\left(\frac{eD}{\min\left(16G\left(\mathcal{F}^{\otimes t}(\mathcal{H})\right), D\right)}\right)$$

$$\times \log\left(\frac{e\sqrt{b}}{\min\left(\sqrt{b}, 640\sqrt{3}G\left(\mathcal{F}^{\otimes t}(\mathcal{H})\right) \sqrt{H} \log\left(\frac{eD}{\min(16G(\mathcal{F}^{\otimes t}(\mathcal{H})), D)}\right)\right)}\right).$$

The above expression is optimal with respect to the parameters $\alpha$ and $\delta$ and clearly gives four different possibilities based on the two minima. Since, under a bounded setting, $G(\mathcal{F}^{\otimes t}(\mathcal{H}))$ would typically go down with $nt$, for large enough $nt$ we have that

$$\min(16G\left(\mathcal{F}^{\otimes t}(\mathcal{H})\right), D) = 16G\left(\mathcal{F}^{\otimes t}(\mathcal{H})\right)$$

and

$$\min\left(\sqrt{br}, 640\sqrt{3}G\left(\mathcal{F}^{\otimes t}(\mathcal{H})\right) \sqrt{Hr} \log\left(\frac{eD}{\min\left(16G\left(\mathcal{F}^{\otimes t}(\mathcal{H})\right), D\right)}\right)\right)$$

$$= 640\sqrt{3}G\left(\mathcal{F}^{\otimes t}(\mathcal{H})\right) \sqrt{Hr} \log\left(\frac{eD}{12G\left(\mathcal{F}^{\otimes t}(\mathcal{H})\right)}\right).$$

Under these conditions, we have

$$\tilde{\mathfrak{R}}_{nt}(\tilde{\mathcal{L}}_\ell(\mathcal{F}^{\otimes t}(\mathcal{H})), r) \leq 2560\sqrt{3}\sqrt{r}\Pi(\mathcal{F}^{\otimes t}(H)) \log\left(\frac{e\sqrt{3}\sqrt{b}}{1920\Pi(\mathcal{F}^{\otimes t}(H))}\right),$$

where $\Pi(\mathcal{F}^{\otimes t}(H)) = G\left(\mathcal{F}^{\otimes t}(\mathcal{H})\right)\sqrt{H}\log\left(\frac{eD}{16G(\mathcal{F}^{\otimes t}(\mathcal{H}))}\right)$.

Under the other three possibilities for the minima some or all the logarithmic factors are identically one and lead to a simplified bound.

Alternatively, for a more simple bound which always holds set $\alpha = \sqrt{\frac{br}{nt}}$ and $\delta = \frac{D}{\sqrt{nt}}$ to get

$$\tilde{\mathfrak{R}}_{nt}(\tilde{\mathcal{L}}_\ell(\mathcal{F}^{\otimes t}(\mathcal{H})), r) \leq 640\sqrt{3}G\left(\mathcal{F}^{\otimes t}(\mathcal{H})\right)\sqrt{Hr}\log(nt)^2 + \frac{80\sqrt{3}D\sqrt{Hr}\log(nt) + 4\sqrt{br}}{\sqrt{nt}}.$$

$\square$

**Remark 3** (Standard single function setting.). *Everything in the proof of Theorem 16 up to Equation (31) holds for a non-compositional model and where $t = 1$ which corresponds to the standard setting. In this context, there is no need to apply the Gaussian chain rule, Theorem 8. Let $\mathcal{Q} : \mathcal{X} \to \mathbb{R}$ be a hypothesis class and note that we have $n$ i.i.d. samples.*

*If we set $\alpha = \min\left(\sqrt{br}, 10\sqrt{3}\tilde{\mathfrak{G}}_n(\mathcal{Q})\sqrt{Hr}\right)$ then we have*

$$\tilde{\mathfrak{R}}_n(\mathcal{L}_\ell(\mathcal{Q}), r) \leq 40\sqrt{3}\tilde{\mathfrak{G}}_n(\mathcal{Q})\sqrt{Hr}\log\left(\frac{e\sqrt{b}}{\min\left(\sqrt{b}, 10\sqrt{3}\tilde{\mathfrak{G}}_n(\mathcal{Q})\sqrt{H}\right)}\right),$$

*and if we set $\alpha = \sqrt{\frac{br}{n}}$*

$$\tilde{\mathfrak{R}}_n(\mathcal{L}_\ell(\mathcal{Q}), r) \leq 20\sqrt{3}\tilde{\mathfrak{G}}_n(\mathcal{Q})\sqrt{Hr}\log(n) + \frac{3\sqrt{br}}{\sqrt{n}}.$$

*Using Lemma 5, we also get the following bounds for local Rademacher complexity (as opposed to width),*

$$\mathfrak{R}_n(\mathcal{L}_\ell(\mathcal{Q}), r) \leq 40\sqrt{3}\tilde{\mathfrak{G}}_n(\mathcal{Q})\sqrt{Hr}\log\left(\frac{e\sqrt{b}}{\min\left(\sqrt{b}, 10\sqrt{3}\tilde{\mathfrak{G}}_n(\mathcal{Q})\sqrt{H}\right)}\right) + \sqrt{\frac{br}{n}},$$

$$\mathfrak{R}_n(\mathcal{L}_\ell(\mathcal{Q}), r) \leq 20\sqrt{3}\tilde{\mathfrak{G}}_n(\mathcal{Q})\sqrt{Hr}\log(n) + \frac{5\sqrt{br}}{\sqrt{n}}.$$

### D.5 Theorems for transition from empirical to population

**Lemma 19.** *Let $\mathcal{F}$ be a class of functions that map $\mathcal{X}$ into $[0, b]$ with $b > 0$. Fix $\alpha > 0$. For every $\delta > 0$ and $r$ that satisfy*

$$r \geq 4(\alpha + 1)\mathbb{E}\mathfrak{R}_\mathbf{Z}\{\boldsymbol{f} \in \mathcal{F}^{\otimes t} \mid P\boldsymbol{f} \leq r\} + \frac{b\delta}{nt}\left(\frac{8}{3} + \frac{2}{\alpha}\right)$$

*we have with probability at least $1 - e^{-\delta}$ that*

$$\left\{\boldsymbol{f} \in \mathcal{F}^{\otimes t} \mid P\boldsymbol{f} \leq r\right\} \subseteq \left\{\boldsymbol{f} \in \mathcal{F}^{\otimes t} \mid \hat{P}^n\boldsymbol{f} \leq 2r\right\}.$$

The proof is similar to the proof of Corollary 2.2 within Bartlett et al. [2005].

*Proof of Lemma 19.* For $\boldsymbol{f} \in \{\boldsymbol{f} \in \mathcal{F}^{\otimes t} \mid P\boldsymbol{f} \leq r\}$ we have $\mathrm{Var}[\boldsymbol{f}] \leq P\boldsymbol{f}^2 \leq bP\boldsymbol{f} \leq br$. By Theorem 10, with probability $1 - e^{-\delta}$, every $\boldsymbol{f} \in \{\boldsymbol{f} \in \mathcal{F}^{\otimes t} \mid P\boldsymbol{f} \leq r\}$ satisfies, with $\alpha = \sqrt{2}$,

$$\hat{P}^n \boldsymbol{f} \leq P\boldsymbol{f} + 2(1+\alpha)\mathbb{E}\mathfrak{R}_{\mathbf{Z}}\mathcal{F}_r^{\otimes t} + \sqrt{\frac{2br\delta}{nt}} + b\left(\frac{1}{3} + \frac{1}{\alpha}\right)\frac{\delta}{nt}$$

$$\leq r + 2(1+\alpha)\mathbb{E}\mathfrak{R}_{\mathbf{Z}}\mathcal{F}_r^{\otimes t} + \sqrt{\frac{2br\delta}{nt}} + b\left(\frac{1}{3} + \frac{1}{\alpha}\right)\frac{\delta}{nt}$$

$$\leq r + 2(1+\alpha)\mathbb{E}\mathfrak{R}_{\mathbf{Z}}\mathcal{F}_r^{\otimes t} + \frac{b\delta}{nt} + \frac{r}{2} + b\left(\frac{1}{3} + \frac{1}{\alpha}\right)\frac{\delta}{nt}$$

$$\leq 2r.$$

$\square$

## D.6 Smooth Learning Bounds

**Lemma 20.** *Under the setting of Theorem 16 for any $\delta > 0$, the fixed point $r^*$ is bounded by,*

$$c_1 G\left(\mathcal{F}^{\otimes t}(\mathcal{H})\right)^2 Hb\log(nt)^4 + \frac{2332800D^2 Hb\log(nt)^2}{nt} + \frac{112b^2\delta}{3nt} + \frac{1944b^2}{nt}$$

*where $c_1 < 1.5 \times 10^8$ and $G(\mathcal{F}^{\otimes t}(\mathcal{H})) = L\tilde{\mathfrak{G}}_{nt}(\mathcal{H}) + \tilde{\mathfrak{G}}_n(\mathcal{F})$.*

*Further, if*

$$16G\left(\mathcal{F}^{\otimes t}(\mathcal{H})\right) \leq D \quad and \quad \sqrt{b} \geq 640\sqrt{3}\Pi(\mathcal{F}^{\otimes t}(H)),$$

*where $\Pi(\mathcal{F}^{\otimes t}(H)) = \sqrt{H}G\left(\mathcal{F}^{\otimes t}(\mathcal{H})\right)\log\left(\frac{eD}{16G(\mathcal{F}^{\otimes t}(\mathcal{H}))}\right)$, then for any $\delta > 0$, the fixed the fixed point $r^*$ is bounded by,*

$$c_2 G\left(\mathcal{F}^{\otimes t}(\mathcal{H})\right)^2 Hb\log\left(\frac{De}{16G\left(\mathcal{F}^{\otimes t}(\mathcal{H})\right)}\right)^2 \log\left(\frac{2\sqrt{3}\sqrt{b}}{1920G\left(\mathcal{F}^{\otimes t}(\mathcal{H})\right)\sqrt{H}\log\left(\frac{eD}{16G(\mathcal{F}^{\otimes t}(\mathcal{H}))}\right)}\right)^2$$

$$+ \frac{112b^2\delta}{3nt}$$

*with $c_2 < 8 \times 10^8$.*

*Proof of Lemma 20.* Let

$$\gamma(r) = 4(\alpha+1)\mathbb{E}\mathfrak{R}_{\mathbf{Z}}(b\boldsymbol{f} \in \mathcal{F}^{\otimes t} \mid Pf \leq \frac{r}{b}) + \frac{b^2\delta}{nt}\left(\frac{8}{3} + \frac{2}{\alpha}\right).$$

Now we have

$$b\mathbb{E}\mathfrak{R}_{\mathbf{Z}}\left\{\boldsymbol{f} \in \mathcal{F}^{\otimes t} \mid Pf \leq \frac{r}{b}\right\} \leq 4(1+\alpha)\mathbb{E}\mathfrak{R}_{\mathbf{Z}}\left\{b\boldsymbol{f} \in \mathcal{F}^{\otimes t} \mid Pf \leq \frac{r}{b}\right\} + \frac{b^2\delta}{nt}\left(\frac{8}{3} + \frac{2}{\alpha}\right)$$

$$= \gamma(r). \tag{32}$$

Note that from Lemma 19, we have that, for any $\delta > 0$, with probability at least $1 - e^{-\delta}$,

$$\mathfrak{R}_{\mathbf{Z}}\left\{b\boldsymbol{f} \in \mathcal{F}^{\otimes t} \mid P\boldsymbol{f} \leq \frac{r}{b}\right\} \leq \mathfrak{R}_{\mathbf{Z}}\left\{b\boldsymbol{f} \in \mathcal{F}^{\otimes t} \mid \hat{P}^n\boldsymbol{f} \leq \frac{2r}{b}\right\}.$$

Further, we can apply the concentration of Rademacher complexity, Lemma 11, to bound the expected value of the left-hand side as follows. With probability at least $1 - e^{-\delta}$,

$$\mathbb{E}\mathfrak{R}_{\mathbf{Z}}\left\{b\boldsymbol{f}\in\mathcal{F}^{\otimes t}\mid Pf\leq\frac{r}{b}\right\}\leq 2\mathfrak{R}_{\mathbf{Z}}\left\{b\boldsymbol{f}\in\mathcal{F}^{\otimes t}\mid Pf\leq\frac{r}{b}\right\}+\frac{b^2\delta}{nt}.$$

Hence, with probability $1-2e^{-\delta}$, we get,

$$\mathbb{E}\left[\mathfrak{R}_{\mathbf{Z}}\left\{b\boldsymbol{f}\in\mathcal{F}^{\otimes t}\mid Pf\leq\frac{r}{b}\right\}\right]\leq 2\mathfrak{R}_{\mathbf{Z}}\left\{b\boldsymbol{f}\in\mathcal{F}^{\otimes t}\mid \hat{P}^nf\leq\frac{2r}{b}\right\}+\frac{b^2\delta}{nt}.$$

We now apply Lemma 16 and upper bound the right-hand side by taking supremum over $\mathbf{Z}$. This gives us,

$$\mathbb{E}\left[\mathfrak{R}_{\mathbf{Z}}\left\{b\boldsymbol{f}\in\mathcal{F}^{\otimes t}\mid Pf\leq\frac{r}{b}\right\}\right]\leq 2\mathfrak{R}_{nt}\left\{b\boldsymbol{f}\in\mathcal{F}^{\otimes t}\mid \hat{P}^nf\leq\frac{2r}{b}\right\}+\frac{b^2\delta}{nt}.$$

Plugging the above in Eqn. (32), we get,

$$b\mathbb{E}\mathfrak{R}_{\mathbf{Z}}\left\{\boldsymbol{f}\in\mathcal{F}^{\otimes t}\mid Pf\leq\frac{r}{b}\right\}\leq 8(1+\alpha)\mathbb{E}\mathfrak{R}_{nt}\left\{b\boldsymbol{f}\in\mathcal{F}^{\otimes t}\mid \hat{P}^nf\leq\frac{2r}{b}\right\}$$
$$+\frac{b^2\delta}{nt}\left(\frac{8}{3}+4(1+\alpha)+\frac{2}{\alpha}\right).$$

Suppose that the local Rademacher complexity is bounded by a multiple of $\sqrt{r}$, say $\sqrt{r}E$. Now bounding the local Rademacher complexity with Theorem 16, we have that,

$$b\mathbb{E}\mathfrak{R}_{\mathbf{Z}}\left\{\boldsymbol{f}\in\mathcal{F}^{\otimes t}\mid Pf\leq\frac{r}{b}\right\}\leq 8(1+\alpha)\sqrt{b}\sqrt{2r}E+\frac{b^2\delta}{nt}\left(\frac{8}{3}+4(1+\alpha)+\frac{2}{\alpha}\right)$$
$$=A\sqrt{r}+D$$

where

$$A=8(1+\alpha)\sqrt{b}\sqrt{2}E\qquad D=\frac{b^2\delta}{nt}\left(\frac{8}{3}+4(1+\alpha)+\frac{2}{\alpha}\right).$$

Note that $A\sqrt{r}+D$ is sub-root and larger than $\gamma(r)$. Thus if we solve for a fixed point $r^*$ of this expression we have $A\sqrt{r^*}+D=r^*$. By Lemma 12 we have $r^*\leq A^2+2D$.

Therefore, with $\alpha=1/8$,

$$b\mathfrak{R}_n\left\{\boldsymbol{f}\in\mathcal{F}^{\otimes t}\mid Pf\leq\frac{r}{b}\right\}$$

is bounded by a sub-root function of $r$ with the following fixed point:

$$b72\left(E\right)^2b+\frac{139}{3}\frac{b^2\delta}{nt}.\tag{33}$$

We have seen from Theorem 16 that if $16G\left(\mathcal{F}^{\otimes t}(\mathcal{H})\right)\leq D$ and $\sqrt{b}\geq 640\sqrt{3}\Pi(\mathcal{F}^{\otimes t}(H))$ then we can set

$$E=2560\sqrt{3}\Pi(\mathcal{F}^{\otimes t}(H))\log\left(\frac{e\sqrt{3}\sqrt{b}}{1920\Pi(\mathcal{F}^{\otimes t}(H))}\right).$$

In this setting, using Equation (33), the fixed point is bounded by:

$$796262400G\left(\mathcal{F}^{\otimes t}(\mathcal{H})\right)^2Hb\log\left(\frac{De}{16G\left(\mathcal{F}^{\otimes t}(\mathcal{H})\right)}\right)^2\log\left(\frac{2\sqrt{3}\sqrt{b}}{1920G\left(\mathcal{F}^{\otimes t}(\mathcal{H})\right)\sqrt{H}\log\left(\frac{eD}{16G(\mathcal{F}^{\otimes t}(\mathcal{H}))}\right)}\right)^2+\frac{112b^2\delta}{3nt}$$

Alternatively, we can always set $E$ as follows:

$$E=640\sqrt{3}G\left(\mathcal{F}^{\otimes t}(\mathcal{H})\right)\sqrt{H}\log\left(nt\right)^2+\frac{80\sqrt{3}D\sqrt{H}\log\left(nt\right)+5\sqrt{b}}{\sqrt{nt}}.$$

therefore we also have that the fixed point is bounded as:

$$149299200G\left(\mathcal{F}^{\otimes t}(\mathcal{H})\right)^2Hb\log\left(nt\right)^4+\frac{2332800D^2Hb\log\left(nt\right)^2}{nt}+\frac{112b^2\delta}{3nt}+\frac{1944b^2}{nt}.$$

$\square$

## D.7  Smooth MTL and MTL via MTRL

When we add the additional assumption that the loss function is smooth we have the following extensions to Theorems 12 and 15

**Theorem 17.** *Let $(\hat{\boldsymbol{f}}, \hat{h})$ be an empirical risk minimizer as given in Eqn. (2). Let $\psi(r) \geq b\mathfrak{R}_n(\mathcal{F}^{\otimes t}(\mathcal{H}), r)$ with $r^*$ the fixed point of $\psi(r)$. Then, under Assumption 1 along with $\ell$ being $H$-smooth, with probability $1 - 2e^{-\delta}$,*

$$R_{\text{source}}(\hat{\boldsymbol{f}}, \hat{h}) \leq R_{\text{source}}(\boldsymbol{f}^*, h^*) + \sqrt{R_{\text{source}}(\boldsymbol{f}^*, h^*)} \left( 6\sqrt{\frac{b\delta}{nt}} + 146\sqrt{\frac{r^*}{b}} \right) + \frac{102b\delta}{nt} + \frac{217r^*}{b}.$$
(34)

*where*

$$\frac{r^*}{b} \leq c \left( G\left(\mathcal{F}^{\otimes t}(\mathcal{H})\right)^2 H \log{(nt)}^4 + \frac{D^2 H b \log{(nt)}^2}{nt} + \frac{b\delta}{nt} + \frac{b^2}{nt} \right)$$

*with $c < 1.5 \times 10^8$ and $G(\mathcal{F}^{\otimes t}(\mathcal{H})) = L\tilde{\mathfrak{G}}_{nt}(\mathcal{H}) + \tilde{\mathfrak{G}}_n(\mathcal{F})$.*

*Proof of Theorem 17.* Use the fixed point bound from Lemma 20 that always holds and substitute it into Theorem 12. □

**Theorem 18.** *Let $\hat{h}$ and $\hat{f}_0$ be the learned representation and target predictor, as described in Eqns. (2) and (3). Let $\psi_1(r) \geq b\mathfrak{R}_n(\ell \circ \mathcal{F}^{\otimes t} \circ \mathcal{H}, r)$ and $\psi_2(r) \geq b\mathfrak{R}_n(\ell \circ \mathcal{F}_0, r)$ with $r_1^*$ and $r_2^*$ the fixed points of $\psi_1(r)$ and $\psi_2(r)$, respectively. Then, under Assumption 1 along with $\ell$ being $H$-smooth. and that $\boldsymbol{f}^*$ is $(\nu, \epsilon)$-diverse over $\mathcal{F}_0$ w.r.t. $h^*$, with probability at least $1 - 2e^{-\delta}$, the transfer learning risk is upper-bounded by,*

$$R_{target}(\hat{f}_0, \hat{h}) \leq R_{target}(f_0^*, h^*) + \sqrt{R_{target}(f_0^*, h^*)} \left( 9\sqrt{\frac{b\delta}{m}} + 219\sqrt{\frac{r_1^*}{b}} \right) + \frac{171b\delta}{m} + \frac{21967r_1^*}{2b}$$
(35)

$$+ \frac{1}{\nu} \left( \sqrt{R_{\text{source}}(\boldsymbol{f}^*, h^*)} \left( 6\sqrt{\frac{b\delta}{nt}} + 146\sqrt{\frac{r_2^*}{b}} \right) + \frac{102b\delta}{nt} + \frac{217r_2^*}{b}. \right) + \varepsilon, \quad (36)$$

*where*

$$\frac{r_2^*}{b} \leq c \left( G\left(\mathcal{F}^{\otimes t}(\mathcal{H})\right)^2 H \log{(nt)}^4 + \frac{D^2 H b \log{(nt)}^2}{nt} + \frac{b\delta}{nt} + \frac{b^2}{nt} \right)$$

*where $c < 1.5 \times 10^8$ and*

$$\frac{r_1^*}{b} \leq 97200\tilde{\mathfrak{G}}_m^2(\mathcal{F}_0 \circ \hat{h}) H \log{(m)}^2 + \frac{112b\delta}{3m} + \frac{1296b}{m}.$$

*Proof of Theorem 18.* Use the the fixed point bound from Lemma 20 that always holds and substitute it into Theorem 15. □

## D.8  Local Rademacher complexity chain rule

**Theorem 19.** *Suppose the loss function $\ell$ is $L_\ell$-Lipschitz. Define the restricted representation and predictor classes as follows,*

$$\ell \circ \mathcal{F}_{\mathbf{X}}(r) := \left\{ \ell \circ \boldsymbol{f} \in \ell \circ \mathcal{F}^{\otimes t} : \exists h \in \mathcal{H} : V(\ell \circ \boldsymbol{f} \circ h) \leq r \right\}$$

$$\mathcal{H}_{\mathbf{X}}(r) := \left\{ h \in \mathcal{H} : \exists \boldsymbol{f} \in \mathcal{F}^{\otimes t} : V(\ell \circ \boldsymbol{f} \circ h) \leq r \right\},$$

*where $V$ is the functional in the local Rademacher complexity description. Under Assumptions 1.B and 1.C and that the worst-case Gaussian width of the above is bounded by the sub-root functions $\psi_{\mathcal{F}}$ and $\psi_{\mathcal{H}}$, respectively, there exists an absolute constant $c$ such that*

$$\tilde{\mathfrak{G}}_n(\mathcal{L}_\ell(\mathcal{F}^{\otimes t}(\mathcal{H}), r)) \leq c \Big( \left(LL_\ell\psi_{\mathcal{F}}(r) + \psi_{\mathcal{H}}(r)\right) \log{(nt)} + \frac{D}{(nt)^2} \Big).$$

*Proof of Theorem 19.* Define

$$\ell \circ \mathcal{F}_{\mathbf{X}}^{\otimes t}(\mathcal{H})(r) = \{(\ell \circ \boldsymbol{f}, h) : V(\ell \circ \boldsymbol{f} \circ h) \le r\}.$$

Observe that for any $(\ell \circ \boldsymbol{f}, h) \in \mathcal{F}_{\mathbf{X}}^{\otimes t}(\mathcal{H})$, we have $\boldsymbol{f} \in \mathcal{F}_{\mathbf{X}}(r)$ and $h \in \mathcal{H}_{\mathbf{X}}(r)$. Hence,

$$\ell \circ \mathcal{F}_{\mathbf{X}}^{\otimes t}(\mathcal{H})(r) \subseteq \mathcal{F}_{\mathbf{X}}(r) \circ \mathcal{H}_{\mathbf{X}}(r)$$

Further, by assumption, the composed function $\ell \circ \boldsymbol{f}$ is $L_\ell L$-Lipschitz. We can now apply the (standard) chain rule of Tripuraneni et al. [2021] which gives us,

$$\mathfrak{G}_n(\mathcal{L}_\ell(\mathcal{F}_{\mathbf{X}}^{\otimes t}(\mathcal{H})(r))) \le \mathfrak{G}_{nt}(\ell \circ \mathcal{F}_{\mathbf{X}}(r) \circ \mathcal{H}_{\mathbf{X}}(r))$$

$$\le c\left((LL_\ell \psi_{\mathcal{F}}(r) + \psi_{\mathcal{H}}(r))\log{(nt)} + \frac{D}{(nt)^2}\right)$$

which completes the proof. $\qquad\square$

# E    Detailed comparison with prior works

In this section, we provide a more detailed comparison with prior works.

## E.1    Comparison with Srebro et al. [2010]

Firstly, we identify some erroneous or missing though fixable steps in the proof of Srebro et al. [2010].

1. **Dudley's integral formula.** The work of Srebro et al. [2010] seems to interchange the concepts of Rademacher width and complexity with the same notation $\mathfrak{R}_n(\mathcal{F})$. The original Dudley's formula and its truncated version proved by Srebro et al. [2010] is for Rademacher (or Gaussian) "width", yet in their work, it is used to bound Rademacher complexity. It is possible to translate from width to complexity, but this will lead to an additional term of $\sqrt{\frac{br}{n}}$ as detailed below. This additional term only changes their result by constant factors.

   **Lemma 21.** *Consider a class of functions $\mathcal{F}$ with range in $[0, b]$. Then given input $\mathbf{Z}$ of $n$ points, $\mathfrak{R}_{\mathbf{Z}}\left\{f \in \mathcal{F} : \hat{P}^n f \le r\right\} \le 2\tilde{\mathfrak{R}}_{\mathbf{Z}}\left\{f \in \mathcal{F} : \hat{P}^n f \le r\right\} + \sqrt{\frac{br}{n}}$.*

   *Proof.* The proof follows by an application of Lemma 5. In particular, given input $\mathbf{Z}$ of $n$ points, for any $f \in \left\{f \in \mathcal{F} : \hat{P}^n f \le r\right\}$, we have that

   $$\frac{1}{\sqrt{n}}\|f\|_{L^2(\mathbf{Z})} = \sqrt{\frac{1}{n^2}\sum_{i=1}^n f(z_i)^2} \le \sqrt{\frac{b}{n}\frac{1}{n}\sum_{i=1}^n f(z_i)} \le \sqrt{\frac{br}{n}}$$

   Plugging this in Lemma 5 gives the claimed bound. $\qquad\square$

2. **Centering.** Within Srebro et al. [2010] they consider loss with the "bounded difference" property: $\forall \hat{y}, \hat{y}, y'$ we have that $|\ell(\hat{y}, y) - \ell(\hat{y}', y)| \le b$. To bound the local Rademacher complexity of their loss class which is empirically constrained with $\hat{P}^n(\ell \circ f) \le r$, they use Dudley's integral. Note the upper limit of integration in Dudley's integral is $\sqrt{\hat{P}^n(\ell \circ f)^2}$ and it is claimed that $\sqrt{\hat{P}^n(\ell \circ f)^2} \le \sqrt{br}$ because $\hat{P}^n(\ell \circ f)^2 \le bP(\ell \circ f) \le br$. Yet, this reasoning only follows for $b$-bounded losses not under this weaker condition of bounded difference. Yet, it is possible to center the process and perform a comparable analysis. To resolve this issue let $\tilde{f} = \arg\inf_{f \in \mathcal{F}} \hat{P}^n(\ell \circ f)$ and consider $(\ell \circ f) - (\ell \circ \tilde{f})$. This process is now $b$-bounded. Centering the process in this way does not affect the downstream results because Gaussian/Rademacher width is shift agnostic due to symmetry.

3. **Missing condition on** $n$. This omission is related to the subsequent discussion about comparison with Srebro et al. [2010]. In the proof of Lemma 2.2, after Eqn. 23, the limits of integration are required to satisfy $\sqrt{12Hr}\mathfrak{R}_n(\mathcal{F}) \le \sqrt{br}$. Using the fact that in the setup, in general, $\mathfrak{R}_n(\mathcal{F}) = \Theta\left(\frac{B}{\sqrt{n}}\right)$, the above is equivalent to $n = \Omega\left(\frac{HB^2}{b}\right)$, This condition on $n$ is missing from their main statement. Further, as we detail in Appendix E.1.1, this term, in general, can be unbounded.

4. **Fat-Shattering inequality**. Using the notation from Srebro et al. [2010]. Note that $\text{fat}_x$ is a decreasing function in $x$. Also $x\log(\frac{n}{x})$ is a decreasing function in $x$ for $x \ge 1$. Therefore $\text{fat}_x \log(\frac{n}{\text{fat}_x})$ is an increasing function in $x$. So therefore, for $\varepsilon \in [\gamma, \theta]$ we have $\text{fat}_\gamma \log(\frac{n}{\text{fat}_\gamma}) \le \text{fat}_\varepsilon \log(\frac{n}{\text{fat}_\varepsilon}) \le \text{fat}_\theta \log(\frac{n}{\text{fat}_\theta})$. This in inequality is in the wrong direction on page 27 of Srebro et al. [2010].

Disregarding the above issues, the bound obtained on Srebro et al. [2010] on the local Rademacher complexity is,

$$\tilde{\mathfrak{R}}_n(\mathcal{L}_\ell(\mathcal{F}, r)) = \mathcal{O}\left(\tilde{\mathfrak{R}}_n(\mathcal{F})\sqrt{Hr}\left(\log(n)\right)^{3/2}\right). \tag{37}$$

In contrast ours is,

$$\tilde{\mathfrak{R}}_n(\mathcal{L}_\ell(\mathcal{F}, r)) \le c\tilde{\mathfrak{G}}_n(\mathcal{F})\sqrt{Hr}\log\left(\frac{e\sqrt{b}}{\tilde{\mathfrak{G}}_n(\mathcal{F})\sqrt{H}}\right). \tag{38}$$

**Worst-case improvement for large enough samples.** We first perform a general comparison. Accordingly, we bound the above as,

$$\tilde{\mathfrak{R}}_n(\mathcal{L}_\ell(\mathcal{F}, r)) \le c\tilde{\mathfrak{G}}_n(\mathcal{F})\sqrt{Hr}\log\left(\frac{e\sqrt{b}}{\tilde{\mathfrak{G}}_n(\mathcal{F})\sqrt{H}}\right)$$

$$\le c\tilde{\mathfrak{R}}_n(\mathcal{F})\sqrt{Hr}\sqrt{\log(n)}\log\left(\frac{e\sqrt{b}}{\tilde{\mathfrak{R}}_n(\mathcal{F})\sqrt{H}\sqrt{\log(n)}}\right)$$

$$\le c'\tilde{\mathfrak{R}}_n(\mathcal{F})\sqrt{Hr}\sqrt{\log(n)}\log\left(\frac{eb}{H\log(n)}\frac{n}{B^2}\right).$$

The above is of the same form as the bound of Srebro et al. [2010] in Eqn. (37), and it is easy to see that ours is better whenever $n = e^{\Omega\left(\frac{b}{HB^2}\right)}$.

**Worst-case improvement for constant samples.** Unfortunately, the term $\frac{b}{HB^2}$ can be unbounded, from above, in general, as detailed in Appendix E.1.1. However, a reasonable and standard assumption removes this issue. If we assume that loss at zero is bounded as follows, $\ell(0) \le HB^2$, then we show in Lemma 22, that $\frac{b}{HB^2} = \Omega(1)$, thereby, making the requirement to be a constant number of samples. A uniform bound on $\ell(0)$ features in the characterization of sample complexity of learning with non-negative smooth losses as well as the regime described by the above bound on $\ell(0)$ has been considered in prior works [Arora et al., 2022, Shamir, 2015].

**Improvement for certain hypothesis class.** In the above chain of inequalities, we use the bound that $\tilde{\mathfrak{G}}_n(\mathcal{F}) \le \mathfrak{R}_n(\mathcal{F})\sqrt{\log n}$ which is tight in the worst case. However, there are natural situations where the two are of the same order. A prominent example is linear predictors with data bounded in (any) norm $\|\cdot\|$ and hypothesis class bounded in the via a (regularization) function $R$ which is strongly convex with respect to the dual norm $\|\cdot\|_*$ [Kakade et al., 2008].

### E.1.1 Understanding the $\frac{HB^2}{b}$ term

**No non-trivial lower bound.** Firstly, we see that the term is not non-trivially lower bounded, in general. Consider the function $\ell(f(x)) = \frac{1}{2}\left(q - f(x)\right)^2$. This function is 1-smooth and considers $B = 1$ (which bounds the size of $f(x)$), however, $b$ here is controlled by the size of $q$ and thus could be unbounded. This means that the is a non-trivial lower bound on $\frac{HB^2}{b}$ in general. The above example also works if relax the definition of $b$ to be "bounded difference", $\sup_{f(x), f'(x') \in \mathcal{F}}\left(\ell(f(x)) - \ell(f'(x'))\right)$ as opposed to a uniform absolute bound.

**A lower bound under $\ell(0)$ bound.** We give a lower bound under the assumption of bound on $\ell(0)$.

**Lemma 22.** *Let $\mathcal{F}$ be a class of real-valued functions and let the loss function $\ell : \mathbb{R} \to \mathbb{R}$ be a non-negative $H$-smooth function with $\ell(0) \leq L_0$, and $\sup_{z \in Range(\mathcal{F})} \ell(z) \leq b$. Then,*

$$\frac{HB^2}{b} \geq \frac{1}{1 + \frac{2L_0}{HB^2}}$$

*Proof.* Note that

$$
\begin{aligned}
b = \sup_z \ell(z) &= \ell(z) - \ell(0) + \ell(0) \\
&\leq \langle \ell'(0), z - z' \rangle + \frac{H}{2} |z|^2 + L_0 \\
&\leq \frac{|\ell'(0)|^2}{2H} + \frac{H|z|^2}{2} + \frac{HB^2}{2} + L_0 \\
&\leq \frac{\ell(0)}{2} + HB^2 + L_0 \\
&= 2L_0 + HB^2
\end{aligned}
$$

where the first inequality uses smoothness, the second AM-GM inequality, and the third uses the self-bounding property of non-negative smooth losses (Lemma 2.1 in Srebro et al. [2010]). $\qquad\square$

**No non-trivial upper bound.** Consider the function $\ell(z) = H(1 + \sin(z))$; this is non-negative, $H$-smooth, and $b = 2H$. However, we can define the size of the domain $B$ as arbitrary, without affecting the smoothness and range boundedness. Thus, $\frac{HB^2}{b}$ is unbounded from above, in general. Note that even assuming an upper bound on $\ell(0)$ doesn't help here.

### E.2 Comparison with Denevi et al. [2019], Khodak et al. [2019]

MTRL can be seen as a more specific case of meta-learning in Denevi et al. [2019], Khodak et al. [2019], which has the type of representation learning we study frequently called as "feature learning." It is possible to give guarantees for excess transfer risk in this more general setting. For example, the work of Denevi et al. [2019], under a generative model assumption, is restricted to linear predictors and convex losses. In their Thm. 5, they get a rate of $O\left(\frac{1}{\sqrt{n}} + \frac{1}{\sqrt{t}}\right)$, on excess transfer risk. However, for the feature learning setting, as they point out, this rate contains terms with hidden dependence on $n$ and $t$. Their guarantee for feature learning, with linear representations (which is more restrictive than ours) Corollary 7 , gets a rate of $O\left(\frac{1}{\sqrt{n}} + \frac{1}{t^{1/4}}\right)$.

Another work of Khodak et al. [2019] mainly considers two settings of convex and strongly convex losses, respectively. Interestingly, fast rates in terms of the number of tasks can be obtained. Thm. 5.1 in the work, obtains the following rates, $O\left(\frac{1}{\sqrt{n}} + \frac{1}{\sqrt{nt}}\right)$ for convex Lipschitz losses and $O\left(\frac{1}{n} + \frac{1}{\sqrt{nt}}\right)$ for strongly-convex Lipschitz losses.

