# OpenReview forum: "Optimistic Rates for Multi-Task Representation Learning"
_NeurIPS.cc/2023/Conference — NeurIPS 2023 poster_

### Official Review · Reviewer_JPW9 · 2023-06-23

**Soundness:** 3 good
**Presentation:** 2 fair
**Contribution:** 3 good
**Rating:** 6
**Confidence:** 3

**Summary:**

This work studies multitask representation learning for some (general) function hypothesis classes. The estimators correspond to the composition of a "complex" function $h$ (corresponding to the shared representation) and a specific task regressor $f_i$ from a simple hypothesis class. This work aims at bounding the excess transfer risk of the ERM estimator. It extends the bound of Tripuraneni et. al (2020) to optimistic rates. More precisely, the square root bounds of Tripuraneni et. al (2020) are improved to linear bounds when 0 risk solutions exist on every task (e.g. no label noise). To do so, the authors do not need to assume the loss to be Lipschitz, but instead need to assume it is smooth (Lipschitz derivative).

-------

I acknowledge having read the author's rebuttal. The authors answered some of my concerns as explained in the comments below.

**Strengths:**

Getting optimistic rates for multitask learning is a nice improvement wrt previous bounds on transfer excess risk.
The obtained bound is quite general. Notably, the setup and results are well put in the context of the literature, easily relating this result to previous one.

**Weaknesses:**

My main concern is about the current writing of the paper. First of all, I think it deserves a few careful proofreading, as many typos or grammatical errors can be found. Here are a few examples:
 - l. 108: "we can reuse of"
 - l. 118-119:  "is bound", "is bound by"
 - l. 161: I think there is a j subindex missing in the last sum in both the definition of $Pf$ and $P_n f$
 - l. 192: "In order for there to be any value to learning" -> weird phrasing
 - l. 198: I guess it should be $f^*$ and $\hat{h}$ instead of $f$ and $h'$

Although these typos are not significant in many sentences, they bring confusion when they appear in definitions (lines 161 and 198). More generally, I find the mathematical content to be quite technical and hard to follow. Some claims are confusing if not inexact.
For example the setting described line 171 *does not* imply that $f_j^{\star}\circ h^{\star}$ is the optimal predictor. The same claim indeed also holds for $\frac{1}{2}f_j^{\star}\circ h^{\star}$. I think the authors should here introduce the setting similarly to Tripuraneni et al (2020), saying that $f_j^*$ and $h^*$ are in the hypothesis class.

Also, some notations are heavy/confusing, making the technical content (which is already hard to follow by definition) very hard to get. As an example, the variable $q$ in the equation line 204 represents both an integer and functions. In Theorem 1, it is not clear what $\psi$ exactly is. I find the whole $r_1^*$ and $r_2^*$ quite cumbersome: I would have preferred to directly state Theorem 1 with the bounds on these quantities provided in Theorem 2. More generally, the whole Section 3 is hard to understand. I think that illustrating the obtained results in the particular case of the linear representational model would help the reader.

As a last concern, the current seems like an incremental improvement of Tripuraneni et al (2020), with tools leading to optimistic rates, that are mostly inherited from Srebro et al (2010). Even though the authors claim in the introduction that Srebro et al's tools cannot be directly used to MTL, the difference seems to be tenuous and only be due to bounding the Rademacher complexity.
Also, the authors might provide more motivation regarding optimistic rates.

**Questions:**

See weaknesses section above.

---

> ### Author Rebuttal · Authors · 2023-08-09
>
> Thank you for the feedback and appreciating our contributions. Regarding the typos pointed out and suggestions, we appreciate the careful reading and have made corrections in the revision.
>
> ---
> ## About the setting with the optimal predictor
>
> We thank the reviewer for the suggestion. Indeed, from the assumptions, the optimal predictor is only guaranteed to depend on $x$ via $f_j^*\circ h^*$. We have revised the sentence as suggested by the reviewer.
>
> ---
> ## Regarding $q$ notation on line 204
>
> Note that $q$ is an integer and $q_j(\cdot)$ is a function. However, we will modify the notation emphasize this difference.
>
> ---
> ## About $\psi$ within Thm. 1
>
> We will clarify that it is *some* subroot function of $r$ which bounds the local Rademacher complexity.
>
> ---
> ## For $r_1^*$ and $r_2^*$ notation
>
> Our main result, Theorem 1, actually *only* uses Assumption 1.1., boundedness and non-negativity of the loss function $\ell$. We will change the writing to indicate this. This gives a bound in terms of fixed points of the local Rademacher complexities.
> Finally, under the additional assumption of smoothness of the loss,  Lipschitzness of the predictor class $\mathcal{F}$ and the boundedness of $\mathcal{F}\circ \mathcal{H}$ we can give a more interpretable bound on the fixed point in terms of their individual complexities via the Gaussian chain rule.
> Hence, stating the result in terms of $r_1^*$ and $r_2^*$ is required for generality.
>
> ---
> ## About including a linear representation example
>
> That is a great suggestion; we will indeed include this in the final version.
>
> ---
> ## Regarding our contribution w.r.t. prior works
>
> We see our work as foundational, extending our understanding of multi-task representation learning. For a more complete discussion recall our Section 1.1 - Our techniques, dedicated to `"provid[ing] an overview of techniques and challenges overcome in the context of prior art." However, perhaps this is best emphasized visually, in working towards proving our main MTRL result - see Fig. 1., the proof graph - we have provided many necessary improvements and generalizations to build towards our final MTRL rates.
>
> The main technical contributions are, (a). extending core concentration inequalities tools to the general MTRL setting, (b). bounding the local Rademacher complexity.
> * **Concentration inequalities.**
> Most of the existing tools and techniques in learning theory focuses on the single task setting. In order to show our results we need concentration inequalities which apply to the MTL setting, however these results do not trivially extend to the MTL setting. Indeed, as [YLK+18] observed, the difficulties in deriving MTL results are foundational, going back to a Bennett like inequality for the suprema of empirical processes (our analog is Thm. 7). From here we developed foundational analogs, e.g. Thm. 6, of the single task local Rademacher complexity results applicable to a MTL setting. We believe this theorems are of independent interest.
> * **Optimistic rates for non-negative smooth losses.**
> Part of the proof in [SST10] first bounds the covering number by its fat-shattering dimension and then bounds the fat-shattering dimension by the Rademacher complexity. This works well in the standard single task setting. Yet, there are no analogs we know of for the fat-shattering dimension of the multi-task function class. Herein lies the weakness of applying this approach, whereas our approach uses the Gaussian complexity which is a much more general notion of complexity. Besides the generality of our proof technique, which achieves better rates even in the single task setting, our contribution is a more simple proof.
> * **Mistakes.**
> Finally, in the process of developing the tools which are needed for the MTL setting we have identified various errors within the literature. First, while seminal and foundational, the proof within [SST10] has some minor flaws, in an effort to correct the literature we included those within Appx F.1. Concretely, a fat-shattering inequality is used in the wrong direction, there is an assumption between parameters which is not specified, there is a missing term when converting between Rademacher complexity and width, finally the process is not centered which is required in order for the second moment within the upper limit of integration of Dudley's integral to be bounded. Second, we failed to generalize Lemma 17 to bounded and possibly negative functions see footnote 5 on page 27. Finally, while not a mistake, we clarify the literature w.r.t. a comment made within [YLK+18] about achieving the same constants within a single task setting, see lines 344-356.

---

> > ### Comment · Reviewer_JPW9 · 2023-08-14
> > **Author rebuttal**
> >
> > I thank the authors for their detailed answer. In the light of their answer, I decide to raise my score as the technical contribution seems far from incremental given the authors' answer. I thus think that this work is strong from a technical point of view.
> >
> > Yet, I still believe it requires a lot of polishing in terms of writing, and would recommend the authors to carefully improve this aspect in the revised version.

---

### Official Review · Reviewer_9335 · 2023-06-24

**Soundness:** 4 excellent
**Presentation:** 3 good
**Contribution:** 4 excellent
**Rating:** 8
**Confidence:** 4

**Summary:**

This paper shows novel statistical rates for generalizing to a target task via multi-task representation learning (MTRL) that attain the optimistic 1/nt + 1/m rate, where n is the number of samples per source task, t is the number of source tasks, and m is the number of samples per target tasks, when the optimal source and target risks achievable by the representation and predictor function classes are small. This is an improvement over the previous state-of-the-art rate of 1/rt(nt) + 1/rt(m), and matches the analogous optimistic rate in the single-task setting. Key to the results are novel technical contributions extending local Rademacher complexity analysis to the multi-task setting.

**Strengths:**

1. The results are a very significant contribution in my opinion -- the established rates significantly improve over previous state-of-the-art in the near-realizable setting. Indeed, the near-realizable setting is important to consider. All assumptions are reasonable and consistent with prior work. Broadly, multi-task representation learning is an important research area.

2. The analysis is rigorous, there are no mistakes in the proofs to my knowledge. From my understanding, substantial technical innovation is required to achieve the results by extending the local Rademacher complexity framework to the multi-task setting.

3. The paper is very well-written.

**Weaknesses:**

1. The Related Works section should also compare with [XT21].

**Questions:**

N/A

**Limitations:**

Yes

---

> ### Author Rebuttal · Authors · 2023-08-09
>
> Thank you for the feedback and appreciating our contributions.
> Regarding [XT21], thank you for pointing us to this work --  we will add a discussion as suggested, in the related work section.

---

> > ### Comment · Reviewer_9335 · 2023-08-18
> >
> > Thank you to the authors for their response. I am maintaining my score.

---

### Official Review · Reviewer_BPxZ · 2023-06-29

**Soundness:** 3 good
**Presentation:** 2 fair
**Contribution:** 2 fair
**Rating:** 5
**Confidence:** 2

**Summary:**

This paper aims to examine the optimistic rates for multi-task representation learning. The authors illustrate that the rate may be faster than the standard rate, depending on the complexity of the learning tasks. The analysis comprises multiple theoretical contributions.

**Strengths:**

1. The theoretical analysis sounds solid from my perspective. However, I am not an expert in this area and haven't gone through all the supplementary material. I would like to refer to other reviewers in the discussion period.

2. The authors provide detailed comparisons with other theoretical works and expand on the key contributions, helping to understand the critical points of this work better. However, it is still hard to understand the details for readers not in this field.

**Weaknesses:**

1. The assumption 1 regarding the boundness of the loss function and its gradients is too restrictive, particularly since the feature domain is not bound.

2. Although the paper presents new findings in this field, it lacks a thorough explanation of the significance of these results.

3. No conclusion section.


**Questions:**

1. Can you explain the distinction between multitask learning and multitask representation learning? Typically, MTL involves acquiring shared representation layers, which serve as the objective for MTRL.

2. What is the significance of optimistic rates in MTRL, and how does it manifest in real-world situations?

3. Could you offer an intuitive definition of task diversity as it pertains to definition 2?



**Limitations:**

1. Assumptions are too strong.

---

> ### Author Rebuttal · Authors · 2023-08-09
>
> Thank you for your questions, suggestions, and appreciating our contributions.
>
> ---
> ## Regarding the assumptions on the loss function
>
> Our main result, Theorem 1, actually only uses Assumption 1.1., boundedness and non-negativity of the loss $\ell$. We will change the writing to indicate this.
> For the subsequent results, we assume that the gradient is Lipschitz (i.e. loss is smooth), but this does not preclude the gradient from being unbounded.
> We note that these assumptions are standard and borrowed from prior works in (single-task) learning theory such as [SST10].
>
> ---
> ## About a conclusion section
>
> We will add a conclusion section in the revision.
>
> ---
> ## Regarding the distinction between MTL and MTRL
>
> MTL is more general than MTRL. MTL is about learning several tasks simultaneously. This is not limited to procedures which learn a common representation for all tasks, which is the case for MTRL.
> For an example of work which is MTL but not MTRL see [MP04].
>
> [MP04] - Micchelli, C., \& Pontil, M. (2004). Kernels for Multi--task Learning. Advances in neural information processing systems, 17.
>
> ---
> ## Significance of optimistic rates in MTRL
>
> The pursuit of optimistic rates is motivated by practice. In many settings, while we can prove only a rate of $1/\sqrt{n}$, it is observed in practice that the error converges at a faster rate. This is typically for the tasks that we can learn with good accuracy, which is often the case. This is what we observe in transfer learning and what we hope to understand through optimistic bounds here.
>
> The optimistic rates for MTRL that we provide show that with smooth losses, standard two-stage ERM can automatically adapt to the problem instance (both for the source tasks and the target task). As a result, we can interpolate from the standard rate of $\mathcal{O}(\frac{1}{\sqrt{nt}} + \frac{1}{\sqrt{m}})$ to the fast rate $\mathcal{O}(\frac{1}{nt} + \frac{1}{m})$.
>
> Also, note that in practice, we typically use a fairly complex class for learning the representations (e.g., multilayered neural networks) and a simpler one for the predictor (e.g., linear functions). Therefore, the price we pay for $\epsilon$-excess risk for the target task with the MTRL is $\frac{C(\mathcal{F})}{\epsilon}$ as compared to $\frac{C(\mathcal{F} \circ \mathcal{H})}{\epsilon}$; this can yield significant gains as the former is much smaller. This result further emphasizes the provable benefits of pooling data from multiple tasks for transfer learning.
>
> ---
> ##  Intuition for task diversity assumption
>
> Intuitively the task diversity assumption assumes that the ratio of the excess risk of the target task and the excess risk of the source task is well-behaved, i.e. the ratio is $O(1)$.
> In other words, for any representation, excess risk w.r.t. the best predictors for the target task is upper bounded by the excess risk of the source tasks w.r.t. the best predictors for the source tasks.
> For example, when in the linear case this is when the source tasks span $\mathbb{R}^d$ and therefore are able to "learn" a target task in all directions (e.g. see [DHK+20]).
> We have a high-level discussion on this assumption within Appendix B - Task diversity digression section.
>
> ---
> ## Regarding our assumptions
>
> All our assumptions are standard in learning theory -- see for instance the foundational work of [SST10], in the single-task smooth loss setting, and the work of [TJJ20], in the multi-task Lipschitz loss setting.

---

### Official Review · Reviewer_Sb8D · 2023-07-06

**Soundness:** 3 good
**Presentation:** 3 good
**Contribution:** 2 fair
**Rating:** 6
**Confidence:** 4

**Summary:**

The authors consider the transfer learning and the multi-task learning setting in a Representation Learning context: multiple source tasks are used to learn a good common representation to facilitate the learning process of a target task (transfer learning) or of the same source tasks (multi-task learning). Under regularity assumptions on the loss function and task similarity, the authors provide optimistic statistical rates for the transfer learning and the multi-task learning setting demonstrating the benefit of representation learning. These optimistic bounds interpolate between the standard -1/2 rate and the fast -1 rate, depending on the difficulty (i.e. the realizability) of the learning tasks. In order to reach such a result, the authors also provide the following intermediate contributions: they give a local Rademacher complexity theorem in the representation learning setting (for both the multi-task learning and transfer learning scenarios) and a chain rule for local Rademacher complexity for composite function classes which allows to decouple the (local) complexities of representation and predictor classes.

**Strengths:**

The authors address a topic that is interesting in a formal and rigorous way.

The paper is written in a quite clear way.

**Weaknesses:**

Some bounds given by the authors, especially those related to the Rademacher complexities, and also Sec. 4 and Sec. 5 should be simplified more in my opinion in order to make them more readable.

The authors did not provide computational experiments testing the performance of the proposed method in the main body.

The authors adapt optimistic rates present in literature for the single-task setting in order to get their optimistic bounds for representation learning. I wonder if the theoretical contribution is enough for the venue. I would like to better understand which are the main technical difficulties the authors had to face to adapt the optimistic bounds from the single-task to the multiple-task setting.

Some basic gradient based representation-learning references are missing, such as [1-2] below.

References

[1] Denevi et al. "Online-within-online meta-learning."

[2] Khodak et al. "Adaptive Gradient-Based Meta-Learning Methods."



**Questions:**

In Ass. 1, the Lipschitz and the boundedness assumptions are necessary? Smoothness is for sure necessary to get faster rates.

The authors should recall in my opinion that m in the first equation represent the number of samples of the target task.

Could you please make a detailed comparison between your optimistic bounds and the non-optimistic bounds in the references [1-2] I mentioned above by only keeping the leading terms w.r.t. the samples/tasks number and the complexity measures?

What you call 'transfer learning' looks more similar to meta-learning. In transfer learning usually you only have one source task and one target task and you do not investigate the source task training, but only the transfer knowledge from source to target.

The authors developed their analysis under a quite general notion of task similarity introduced in previous literature. However such similarity assumption seems to be not well motivated for the representation learning setting in which the natural task similarity assumption is that the target estimators of the tasks all lies in the range of the representation. This natural link is well explained for instance in the reference [1] I mentioned above.

The paper 'Optimistic Rates for Learning with a Smooth Loss' does not use Rademacher complexity measures in order to give optimistic rates. Did you based the proofs on that? If yes, why are you instead using Rademacher complexities?

**Limitations:**

I do not see any potential negative societal impact related to this work.

---

> ### Author Rebuttal · Authors · 2023-08-09
>
> Thank you for the feedback and appreciating our work.
>
> ---
> ## Our theoretical contribution and technical difficulties
> We see our work as foundational, extending our understanding of multi-task representation learning. For a more complete discussion recall our Section 1.1 - Our techniques, dedicated to `"provid[ing] an overview of techniques and challenges overcome in the context of prior art." However, perhaps this is best emphasized visually, in working towards proving our main MTRL result - see Fig. 1., the proof graph - we have provided many necessary improvements and generalizations to build towards our final MTRL rates.
>
> The main technical contributions are, (a). extending core concentration inequalities tools to the general MTRL setting, (b). bounding the local Rademacher complexity.
> * **Concentration inequalities.**
> Most of the existing tools and techniques in learning theory focuses on the single task setting. In order to show our results we need concentration inequalities which apply to the MTL setting, however these results do not trivially extend to the MTL setting. Indeed, as [YLK+18] observed, the difficulties in deriving MTL results are foundational, going back to a Bennett like inequality for the suprema of empirical processes (our analog is Thm. 7). From here we developed foundational analogs, e.g. Thm. 6, of the single task local Rademacher complexity results applicable to a MTL setting. We believe this theorems are of independent interest.
> * **Optimistic rates for non-negative smooth losses.**
> Part of the proof in [SST10] first bounds the covering number by its fat-shattering dimension and then bounds the fat-shattering dimension by the Rademacher complexity. This works well in the standard single task setting. Yet, there are no analogs we know of for the fat-shattering dimension of the multi-task function class. Herein lies the weakness of applying this approach, whereas our approach uses the Gaussian complexity which is a much more general notion of complexity. Besides the generality of our proof technique, which achieves better rates even in the single task setting, our contribution is a more simple proof.
> * **Mistakes.**
> Finally, in the process of developing the tools which are needed for the MTL setting we have identified various errors within the literature. First, while seminal and foundational, the proof within [SST10] has some minor flaws, in an effort to correct the literature we included those within Appx F.1. Concretely, a fat-shattering inequality is used in the wrong direction, there is an assumption between parameters which is not specified, there is a missing term when converting between Rademacher complexity and width, finally the process is not centered which is required in order for the second moment within the upper limit of integration of Dudley's integral to be bounded. Second, we failed to generalize Lemma 17 to bounded and possibly negative functions see footnote 5 on page 27. Finally, while not a mistake, we clarify the literature w.r.t. a comment made within [YLK+18] about achieving the same constants within a single task setting, see lines 344-356.
>
> ---
> ## Regarding Assumption 1 Lipschitz and the boundedness
>
> Our main result, Thm. 1, actually *only* uses Assumption 1.1., boundedness and non-negativity of the loss. We will change the writing to indicate this. This gives a bound in terms of fixed points of the local Rademacher complexities. Under the additional assumption of smoothness, we can further bound the fixed point. Finally, under the additional assumption of the Lipschitzness of the predictor class $\mathcal{F}$ and the boundedness of $\mathcal{F}\circ\mathcal{H}$ we can give a more interpretable bound in terms of their individual complexities via the Gaussian chain rule. Concluding, we would like to emphasize that these are standard assumptions, including the non-negativity and boundeness of loss, in learning theory, for instance, see [BBM02, SST10, TJJ20].
>
> ---
> ## Regarding references [1-2]
>
> Thank you for these references. While these indeed bear similarities to our setting, there are crucial differences, which makes them incomparable. These include, a generative model of tasks in the mentioned papers, which we don't have, as well as the assumption of convexity therein (which we understand is made for computational reasons). We will add a discussion to this effect in the revised version.
>
> ---
> ## Regarding transfer learning vs. meta-learning
>
> We are interested in the problem of transfer learning by learning a common representation from multiple source tasks. The most related work is [TJJ20] so we decided to remain within their discourse and reuse their terminologies. We note that the setting of multiple sources tasks has appeared even in many earlier works on transfer learning, for instance, [Bax00]. However, there are indeed many similarities between our setting and the ones mentioned by the reviewer.
>
> ---
> ## Regarding the task similarity assumption
>
> The task similarity assumption has been studied in prior works, for instance [TJJ20]. Our main focus is to understand whether under this standard assumption, we can achieve fast rates, such as those in single task settings. However, in the special case of a linear representation class, the assumption recovers the task diversity assumption within many prior works, such as [DHK+20, TJJ21] and is similar to the one mentioned by the reviewer.
>
> ---
> ## Gaussian vs Rademacher complexity
>
> We assume you mean Gaussian complexity? A side-by-side comparison between our proof and the one within [SST10] shows that although they start similarly, with Dudley's theorem, they soon diverge substantially. The biggest reason is that there exists results in terms Gaussian complexity for which there are no known Rademacher complexity analogs. Nevertheless, since Gaussian and Rademacher complexities are related, so it is possible to state all the results in terms of Rademacher complexity.

---

> > ### Comment · Reviewer_Sb8D · 2023-08-16
> > **Response to the authors**
> >
> > I thank the authors for the reply. Below my comments.
> >
> > Regarding references [1-2], I would like to see a comparison, at least in the convex setting. This would be useful in my opinion in order to understand the meaning of the theoretical results presented by the authors.
> >
> > Regarding the task similarity assumption, the authors say: '..in the special case of a linear representation class, the assumption recovers the task diversity assumption within many prior works, such as [DHK+20, TJJ21] and is similar to the one mentioned by the reviewer.' I would like to have more technical details explaining the link between this and the standard tasks similarity assumption I explained before.

---

> > > ### Author Response · Authors · 2023-08-18
> > >
> > > **Comparison to prior work.**
> > >
> > > As we said in our rebuttal, strictly speaking, our results are incomparable with those prior works, due to different assumptions. However, we present the rates obtained in [1-2], and "compare" them with ours.
> > >
> > > First, recalling the differences, both works study the problem formulation of meta-learning, which, as they explain, is more general than learning with a (potentially non-linear) feature map, which is what they call "Feature Learning". Our work is limited to this setting of "Feature Learning". Besides these works assume a generative model for tasks whereas we have a task diversity assumption. Further, the works consider convex losses, which allows guarantees for gradient-based methods. Our work does not assume convexity.
> > >
> > > We now present the guarantees in the [1-2]. As requested, we only write the bound as a function of number of tasks $T$ and number of samples per tasks $n$.
> > >
> > > The work [1] is restricted to linear predictors and convex losses. In their Thm. 5, they get a rate of $O(\frac{1}{\sqrt{n}}+\frac{1}{\sqrt{T}})$, on excess transfer risk. However, for the feature learning setting, as they point out, this rate contains terms with hidden dependence on $n$ and $T$. Their guarantee for feature learning, with linear representations (which is more restrictive than ours) Corollary 7, gets a rate of $O(\frac{1}{\sqrt{n}}+\frac{1}{T^{1/4}})$.
> > >
> > > The work [2] mainly considers two settings of convex and strongly-convex losses, respectively.
> > > Thm. 5.1 in the work, obtains the following rates,
> > > - $O(\frac{1}{\sqrt{n}}+\frac{1}{\sqrt{nT}})$ for convex Lipschitz losses
> > > - $O(\frac{1}{n}+\frac{1}{\sqrt{n} T})$ for strongly-convex Lipschitz losses.
> > >
> > > In comparison, our rate is between $O(\frac{1}{nT}+\frac{1}{n})$ and $O(\frac{1}{\sqrt{nT}}+\frac{1}{\sqrt{n}})$ depending on the level of realizability in source and target tasks. This adaptivity to realizability is missing in the bounds in works [1-2], which is one of our key contributions. We remind that we are ignoring the hidden numerators in the rate, which interestingly contain complexity of representation and predictor class terms.
> > >
> > > Our worst-case rate then is $O(\frac{1}{\sqrt{n}})$ which is asymptotically no worse than that in [1] and [2] in the convex Lipschitz setting. The rate in [2] for strongly convex setting, with large number of tasks, $\frac{1}{n}$, is better than our worst-case rate but same as our optimistic rate. However, this is primarily due to strong convexity which enable faster rates.
> > >
> > > **Regarding Task similarity assumption**
> > >
> > > We elaborate the connection between our task diversity assumption, which is taken from [TJJ20], and those in works limited to linear representations such as [DHK+20] and [1]. The connections are established in the prior works, and towards this, we quote the relevant parts from these works.
> > >
> > > Note that [TJJ20] say the following about the task-diversity assumption they introduce.
> > >
> > > > Despite the abstraction in this definition of task diversity, it exactly recovers the notion of task diversity in [TJJ20] and Du et al. [2020], where it is restricted to the special case of linear functions and quadratic loss.
> > >
> > > We now motivate the task diversity assumption in [DHK+20] in context of [1]. Ass. 4.3 within [DHK+20] states that the matrix $W^*=[\{w}_1^*,\ldots,\{w}_T^*] \in \mathbb{R}^{k\times T}$ of optimal lower dimensional predictors, satisfies $\sigma_k^2(W^*)=\Omega(\frac{T}{k})$. They go on to say:
> > >
> > > >[This assumption] is equivalent to saying that $\frac{\sigma_1(W^*)}{\sigma_k(W^*)}=O(1)$. Roughly speaking, this means that $\\{w_t^*\\}_{t \in[T]}$ can cover all directions in $\mathbb{R}^k$.
> > >
> > > In contrast, the assumption in [1] is that with $B\_\rho=\mathbb{E}\_{\mu\sim\rho} v\_\mu v\_\mu^{\top}$, where $v\_\mu \in \mathbb{R}^d$, for any $\theta\in\mathbb{R}^{d\times k}$, we have that $\mathrm{Ran}(B\_\rho)\subseteq\mathrm{Ran}(\theta)$. Note that under a generative model, i.e. assuming that the labels are generated by conditional distribution which depends on the product of representation and some lower-dimensional predictor, this assumption is satisfied. So the assumption in [DHK+20] is stronger than [1].
> > >
> > > The reason for this is stated in [DHK+20],
> > >
> > > > Unfortunately, as pointed out by Maurer et al. (2016), there exists an example that satisfies the i.i.d. task assumption for which $\Omega(1/\sqrt{T})$ is unavoidable. This means that the i.i.d. assumption alone is not sufficient if we want to take advantage of a large amount of samples per task. ... We replace the i.i.d. assumption over tasks with natural structural conditions on the input distributions and linear predictors. These conditions depict that the target task can be in some sense “covered” by the source tasks, which will further give rise to the desirable guarantees.
> > >
> > > Our task diversity assumption (taken from [TJJ20]) similarly allows us to "improve" upon the rates in prior works in the i.i.d. tasks setting for general loss functions.

---

> > > > ### Comment · Reviewer_Sb8D · 2023-08-21
> > > > **Reply**
> > > >
> > > > Thank you for the answer.

---

### Decision · Program_Chairs · 2023-09-21

**Decision:**

Accept (poster)

**Comment:**

This paper shows novel statistical rates for generalizing to a target task via multi-task representation learning (MTRL) that attain the optimistic 1/nt + 1/m rate, where n is the number of samples per source task, t is the number of source tasks, and m is the number of samples per target tasks, when the optimal source and target risks achievable by the representation and predictor function classes are small. This is an improvement over the previous state-of-the-art rate of 1/sqrt(nt) + 1/sqrt(m), and matches the analogous optimistic rate in the single-task setting. Key to the results are novel technical contributions extending local Rademacher complexity analysis to the multi-task setting.

All reviewers believe this paper makes valid contributions to the community. The AC agrees and recommends acceptanc.